# Score-informed Neural Operator for Enhancing Ordering-based Causal Discovery

**Jiyeon Kang**[1,2*], **Songseong Kim**[1,2*], **Chanhui Lee**[1,3], **Doyeong Hwang**[1,2],
**Joanie Hayoun Chung**[2], **Yunkyung Ko**[2], **Sumin Lee**[2], **Sungwoong Kim**[3†], **Sungbin Lim**[1,2†]

[1]LG AI Research
[2]Department of Statistics, Korea University
[3]Department of Artificial Intelligence, Korea University

## Abstract

Ordering-based approaches to causal discovery identify topological orders of causal graphs, providing scalable alternatives to combinatorial search methods. Under the Additive Noise Model (ANM) assumption, recent causal ordering methods based on score matching require an accurate estimation of the *Hessian diagonal* of the log-densities. In this paper, we aim to improve the approximation of the Hessian diagonal of the log-densities, thereby enhancing the performance of ordering-based causal discovery algorithms. Existing approaches that rely on Stein gradient estimators are computationally expensive and memory-intensive, while diffusion-model-based methods remain unstable due to the second-order derivatives of score models. To alleviate these problems, we propose **Sc**ore-**i**nformed **N**eural **O**perator (SciNO), a probabilistic generative model in smooth function spaces designed to stably approximate the Hessian diagonal and to preserve structural information during the score modeling. Empirical results show that SciNO reduces order divergence by 42.7% on synthetic graphs and by 31.5% on real-world datasets on average compared to DiffAN, while maintaining memory efficiency and scalability. Furthermore, we propose a probabilistic control algorithm for causal reasoning with autoregressive models that integrates SciNO's probability estimates with autoregressive model priors, enabling reliable data-driven causal ordering informed by semantic information. Consequently, the proposed method enhances causal reasoning abilities of LLMs without additional fine-tuning or prompt engineering.

## 1   Introduction

Ordering-based causal discovery aims to identify a topological ordering of nodes in a causal graph, typically represented as a Directed Acyclic Graph (DAG), such that the order reflects the underlying cause-effect relationships. The combinatorial search over DAG structures is known to be NP-hard, with complexity increasing sharply as the number of variables grows [6]. In contrast, ordering-based approaches first infer a topological order of the causal graph, and then determine the direction of edges based on this order, which substantially reduces the number of candidate graph structures. By leveraging Large Language Models (LLMs), predicting causal orderings rather than full graph structures, improves consistency and reduces structural errors [42]. Moreover, such causal order suffices to identify valid backdoor adjustment sets, without full graph discovery [42, Proposition 3.2].

Recently, causal ordering methods based on score matching have been proposed [33, 35, 45] under the ANM (Additive Noise Model) assumption. All of these approaches determine the causal order by

---

[*]Equal contribution.
[†]Corresponding Authors. {swkim01, sungbin}@korea.ac.kr.

39th Conference on Neural Information Processing Systems (NeurIPS 2025).

iteratively identifying and removing leaf nodes—i.e., variables that do not influence any others in the causal graph. To accomplish this, they require the estimation of the *Jacobian of the score function* $\mathbf{S}(\mathbf{x}) = \nabla_{\mathbf{x}} \log P(\mathbf{x})$, in particular its diagonal, denoted as the *Hessian diagonal* function $\mathscr{D}(\mathbf{x})$:

$$\mathscr{D}(\mathbf{x}) := \text{diag}\left(\mathbf{H}_{\log P}\right)(\mathbf{x}) = \left(\partial_{x_j} \mathbf{S}_j(\mathbf{x}) : j = 1, \ldots, D\right). \tag{1}$$

Score matching based causal ordering methods sequentially identify leaf nodes from a causal graph by estimating variance or mean of the Jacobian of the score function after each leaf removal. SCORE [33] and CaPS [45] use the second-order Stein gradient estimator [23]. However, kernel-based estimators exhibit cubic computational complexity with respect to the number of samples $N$, and suffer from numerical instability due to the kernel matrix inversion. Since the score function must be re-estimated iteratively at every step, the procedure becomes computationally problematic in large-scale causal graphs as causal discovery algorithms require more data samples as the number of nodes $D$ increases. Contrary to kernel-based approaches, DiffAN [35] estimates the Jacobian of the score through the Denoising Diffusion Model [13]. Instead of retraining the score model, DiffAN approximates residue terms by computing the second-order derivatives of initially trained score model at each ordering step. These residual approximations reduce computational burden and allow scalable causal ordering when the number of data points increases, but their performance degrade in high-dimensional settings.

Fundamentally, to identify leaf nodes via the score information, more reliable derivative estimation of score models is necessary; otherwise errors may accumulate throughout the ordering process, ultimately degrading the causal discovery performance. In particular, we focus on improving the approximation of the Hessian diagonal of the log-densities, which is critical for reliable ordering-based causal discovery. While modeling the score function via diffusion models is suitable from a generative perspective, injecting noise into the data can destroy the functional information inherent in the original score function. To address this challenge, inspired by recent advances in score-based generative modeling in Hilbert spaces [25, 26], we propose SciNO (**Sc**ore-**i**nformed **N**eural **O**perator), a functional diffusion model framework which aims to stably approximate the Hessian diagonal of the data distribution by modeling the score function with neural operators [16, 24, 26]. To stably learn the derivatives of the score function, we introduce two key modifications to the time-conditioned Fourier neural operator architecture used in [26]. First, instead of using positional embeddings [43], we incorporate a Learnable Time Encoding (LTE) module that enables the model to jointly learn spatiotemporal derivatives. Second, signals in Fourier layers are decomposed into their real and imaginary parts in the spectral domain, allowing more expressive representation of functional information. Consequently, SciNO enables stabilizing the estimation of both the score function and its second-order derivatives, thus it significantly improves performances of causal ordering methods, including DiffAN and CaPS, especially in high-dimensional settings. Compared to the original DiffAN, SciNO achieves a 42.7% reduction in order divergence on synthetic datasets and a 31.5% reduction on real-world datasets.

Building on the stable estimation of SciNO and motivated by [42], we introduce a method to control autoregressive generative models, e.g. LLMs and Mamba [12], for causal reasoning tasks. While recent studies leverage prior knowledge of LLMs to improve causal reasoning accuracy [2, 8, 21, 27, 42], these approaches often treat LLM responses as binary decisions, failing to reflect uncertainty in prior knowledge [5] and remaining structurally incompatible with ordering-based approaches that sequentially predict leaf nodes. To address these limitations, we propose a probabilistic control method which leverages score-informed statistics inferred by SciNO, guiding autoregressive models to generate more reliable causal reasoning. Applied to real-world datasets, our method reduces order divergence by 64% on average, achieving up to a 79% reduction over uncontrolled models. Furthermore, our approach can lower the computational complexity of LLM queries from $\mathcal{O}(|\mathcal{V}|^2)$ to $\mathcal{O}(|\mathcal{V}|)$ contrary to pairwise prompting methods [1, 8, 15, 20, 27]. Notably, these improvements are achieved without fine-tuning or intensive prompt engineering. The main contributions of our work are summarized as follows:

- We propose a functional diffusion model framework with neural operators, SciNO, that enables stable estimation of both the score function and its derivatives, which are crucial statistics for score matching based causal ordering algorithms assuming ANMs.

- SciNO enables accurate and scalable causal discovery in high-dimensional settings. We empirically show improved performances over existing score matching based causal ordering methods in a memory-efficient way, enhancing its scalability to real-world problems.

- We also propose a probabilistic control method that enhances causal reasoning ability of autoregressive generative models. Without requiring fine tuning, instruction tuning, or prompt engineering, it is particularly effective on high-dimensional causal graphs.

## 2 Preliminaries

In this section, we briefly introduce preliminaries for understanding score matching based causal ordering methods—SCORE [33], DiffAN [35], and CaPS [45]—which differ in terms of identifiability assumptions, Hessian approximation, and leaf node selection criteria.

**Additive Noise Models and Identifiability**  Given a causal Directed Acyclic Graph (DAG) $\mathcal{G}$, a causal order $\pi : \mathcal{V} \to \mathcal{V}$ is a non-unique permutation of the set of nodes $\mathcal{V} = \{1, \ldots, D\}$ such that for any directed edge from node $i$ to $j$ in $\mathcal{G}$, it holds $\pi(i) < \pi(j)$ [30]. Let $\pi_k \subset \pi_{k+1} \subset \pi$ denote a sequence of causal order corresponding to selected nodes at each step $k = 1, \ldots, |\mathcal{V}| - 1$, and write $-\pi_k := \mathcal{V} \setminus \pi_k$, the indices of remaining nodes. To identify a causal order from observational data $\mathcal{D}$, we impose a functional assumption known as the Additive Noise Models (ANMs) [31]:

$$x_i = f_i(\mathrm{Pa}(x_i)) + \epsilon_i, \quad \epsilon_i \sim p_i. \tag{2}$$

**SCORE [33]**  Assuming (2) with Gaussian noise $p_i = \mathcal{N}(0, \sigma_i^2)$, where each $f_i$ is supposed to be *nonlinear* and twice-continuously differentiable, SCORE proposes an identifiability criterion for inferring a topological order of the causal graph $\mathcal{G}$ by investigating the variance of the diagonal Hessian iteratively. Due to [33, 35, Lemma 1], $\mathrm{Var}\left[\mathscr{D}_j(\mathbf{x}_{-\pi_k})\right] = 0$ holds if and only if $j \in \mathcal{V} \setminus \pi_k$ is a leaf node at step $k + 1$. SCORE proposes an ordering algorithm as follows:

$$\mathrm{leaf} = \underset{j \in \mathcal{V} \setminus \pi_k}{\mathrm{argmin}} \, \mathrm{Var}_{\mathbf{x} \sim \mathcal{D}}\left[\mathscr{D}_j(\mathbf{x}_{-\pi_k})\right]. \tag{3}$$

**CaPS [45]**  Under conditions on variance of noises, CaPS proposes an identifiability criterion, where each $f_i$ can be either *linear* or *nonlinear*, extending its applicability to a wider range of structural functions. Under the ANM assumption [45, Theorem 1], CaPS utilizes the expectation of the Hessian diagonal to identify leaf nodes according to the following criterion:

$$\mathrm{leaf} = \underset{j \in \mathcal{V} \setminus \pi_k}{\mathrm{argmax}} \, \mathbb{E}_{\mathbf{x} \sim \mathcal{D}}\left[\mathscr{D}_j(\mathbf{x}_{-\pi_k})\right]. \tag{4}$$

**DiffAN [35]**  SCORE and CaPS require re-estimating the diagonal Hessian in (3) and (4) for each step, resulting in computational overhead when the number of nodes $|\mathcal{V}| = D$ increases. DiffAN replaces the re-estimation procedure by approximating the *deciduous score* $\mathbf{S}(\mathbf{x}_{-\pi_k})$, referring to the score function over the remaining variables after removing leaf nodes. Due to [35, Theorem 1],

$$\mathbf{S}_j(\mathbf{x}_{-\pi_k}) = \mathbf{S}_j(\mathbf{x}) + \sum_{l \in \pi_k} \left( \partial_{x_j} \mathbf{S}_l(\mathbf{x}) \cdot \frac{\mathbf{S}_l(\mathbf{x})}{\partial_{x_l} \mathbf{S}_l(\mathbf{x})} \right), \quad j \in \mathcal{V} \setminus \pi_k. \tag{5}$$

Based on (5), DiffAN leverages diffusion models to learn the score function $\mathbf{S}(\mathbf{x})$ via score model $\widehat{\mathbf{S}}^\theta(t, \mathbf{x})$ near $t \approx 0$ for approximating the Hessian diagonal $\mathscr{D}(\mathbf{x}_{-\pi_k})$ at each step:

$$\mathscr{D}_j(\mathbf{x}_{-\pi_k}) \approx \partial_{x_j} \widehat{\mathbf{S}}_j^\theta(t, \mathbf{x}) + \sum_{l \in \pi_k} \partial_{x_j} \left( \partial_{x_j} \widehat{\mathbf{S}}_l^\theta(t, \mathbf{x}) \cdot \frac{\widehat{\mathbf{S}}_l^\theta(t, \mathbf{x})}{\partial_{x_l} \widehat{\mathbf{S}}_l^\theta(t, \mathbf{x})} \right), \quad j \in \mathcal{V} \setminus \pi_k. \tag{6}$$

## 3 Score-informed Neural Operator

### 3.1 Score Matching in Function Spaces

While approximation (6) reduces computational cost, numerical instability is implicit in the computing second-order derivatives of score models. Conventional MLPs frequently struggle to accurately estimate derivatives [23], so that the curvature information of score models can be deviated from that of the true score function. To address this limitation, we employ Hilbert Diffusion Model (HDM, [26]), which enables functional diffusion modeling in Hilbert spaces $\mathcal{H}$ rather than applying diffusion models in Euclidean space. Due to Sobolev embedding theorem [17], we consider the Sobolev space $\mathcal{H} = W_2^k$, where $k$ denotes the order of differentiability, and adopt a class of neural operators [16] to learn higher-order derivatives of the score function. See Definition A.1 and Remark A.2 for details.

**Theorem 3.1** (Pointwise Approximation). *Let $k \in \mathbb{N}$ such that $k > 2 + \frac{D}{2}$. For any compact subset $K \subset \Omega \subset \mathcal{X}$ and $\epsilon > 0$, there exists a neural operator $\widehat{\mathbf{S}}^\theta$ such that*

$$\sup_{\mathbf{x} \in K} \sum_{|\alpha| \leq m} \left| \partial_{\mathbf{x}}^\alpha \mathbf{S}(\mathbf{x}) - \partial_{\mathbf{x}}^\alpha \widehat{\mathbf{S}}^\theta(\mathbf{x}) \right| \leq \varepsilon, \quad m = \lfloor k - \frac{D}{2} \rfloor. \tag{7}$$

Details of Theorem 3.1 are discussed in Appendix A.2.2. Since we assume $k > 2 + \frac{D}{2}$, we have $m \geq 2$ so that $\widehat{\mathbf{S}}^\theta$ is at least *twice continuously differentiable* which can approximate the target score function $\mathbf{S}_*$ with respect to the Hölder norm $\|\cdot\|_{C^{k-\frac{D}{2}}}$. Theorems A.1 and A.2 are crucial for the approximation (6) to the Hessian diagonal $\mathscr{D}(\mathbf{x})$ since they supports the numerical stability (35) and guarantees the approximation power (37) for computing the second-order derivatives of score models.

Note that our theory is designed to approximate the target score function $\mathbf{S}_* = \nabla_{\mathbf{x}} \log \mathrm{P}_{\text{data}}$ via neural operators. The connection to the marginal score model follows a fundamental procedure in diffusion models. Approximation to marginal score functions with a time variable via neural operators in Hilbert space, $\widehat{\mathbf{S}}^\theta(t, \cdot) \approx \mathbf{S}(t, \cdot)$ for each $t$, is proved in prior work on HDM [26]. Hence the completeness of Hilbert space implies that the distance between $\widehat{\mathbf{S}}^\theta(\cdot)$ and the sequence of functional diffusion models will close to zero as we train score models accurately. Based on HDM and the above approximation power of neural operators, we introduce the architecture of SciNO (**Sc**ore-**i**nformed **N**eural **O**perator) in the following section, a specially designed neural operator which aims to stably approximate the Hessian diagonal $\mathscr{D}(\mathbf{x})$ to enhance score matching based causal discovery algorithms.

## 3.2 Architecture of SciNO

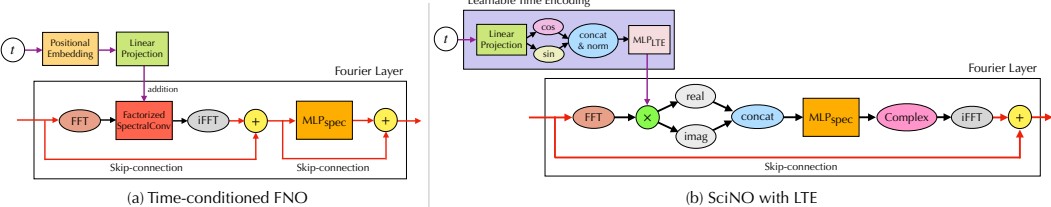

Figure 1: Architectures of time-conditioned FNO [26] and SciNO with Learnable Time Encoding.

Table 1: Comparison of order divergence in Erdös-Rényi synthetic datasets for $D \in \{10, 30, 50, 100\}$ between DiffAN w/ MLP [35], DiffAN w/ SciNO (PE), and DiffAN w/ SciNO (LTE).

| Method\Dataset | ER(d10) | ER(d30) | ER(d50) | ER(d100) |
|---|---|---|---|---|
| DiffAN w/ MLP [35] | $3.2 \pm 1.99$ | $26.7 \pm 8.1$ | $45.5 \pm 7.51$ | $117.0 \pm 12.78$ |
| DiffAN w/ SciNO (PE) | $\mathbf{2.5} \pm 1.36$ | $19.0 \pm 6.91$ | $41.4 \pm 11.33$ | $100.7 \pm 15.62$ |
| DiffAN w/ SciNO (LTE) | $2.7 \pm 1.19$ | $\mathbf{16.9} \pm 6.12$ | $\mathbf{32.8} \pm 6.55$ | $\mathbf{86.6} \pm 12.99$ |

Score-based generative modeling necessitates time-conditioning to model score functions $\widehat{\mathbf{S}}^\theta(t, \mathbf{x})$ over continuous time. Following the approach in DDPM [13], HDM implements time-conditioned Fourier Neural Operator (FNO, [24]) by adding projected tensors from Positional Embedding (PE, [43]) module into each Fourier layer, which transforms the input into the spectral domain and performs spectral convolution to capture global patterns while supporting stable estimation of function derivatives. However, while the time-conditioned FNO performs well in functional generative modeling, it shows slight improvements over the MLP-based DiffAN in causal orderings tasks. This is due to the instability in estimating the derivatives of the score function $\mathbf{S}(\mathbf{x})$ near $t \approx 0$.

To stably learn the derivatives of the score function, we propose two major modifications to the architecture used in [26]. First, signals in Fourier layers are decomposed into their real and imaginary parts in the spectral domain, allowing more expressive representation of functional information. With the modified Fourier layer, SciNO achieves improved causal ordering performance compared to the original DiffAN. However, this performance gain relatively decreases as the number of variables

increases (see SciNO (PE) in Table 1). To scale-up SciNO for high-dimensional graphs, rather than assigning fixed embedding vectors to discrete positions as in PE, we propose a learnable encoding function defined over a continuum, *Learnable Time Encoding* (LTE) module, that enables the model to jointly learn derivatives in both spatial and temporal directions. The detailed architectural specifications for time-conditioned FNO [26] and SciNO are provided in Appendix A.3.

**Learnable Time Encoding in SciNO** We first project the time variable $t$ using $F$-dimensional learnable weights $\mathbf{w}_{\mathrm{proj}} \in \mathbb{R}^F$, to get a projected vector $t\mathbf{w}_{\mathrm{proj}}$. Then, we obtain a normalized signal $\Phi(t) = [\cos(t\mathbf{w}_{\mathrm{proj}}), \sin(t\mathbf{w}_{\mathrm{proj}})]/\sqrt{2F}$, which is passed through an $\mathrm{MLP}_{\mathrm{LTE}} : \mathbb{R}^{2F} \to \mathbb{R}^H$ layer. This embedding is similar to [22], which is designed for multi-dimensional spatial PE. Then, we inject the time encoding via elementwise multiplication into both the real and imaginary parts before the input of $\mathrm{MLP}_{\mathrm{spec}}$ in Fourier layers (see Figure 1). This continuous encoding aims to learn functional information, including spatiotemporal derivatives, and aligns with the objective of fitting higher-order derivatives by enabling a more flexible representation. Consequently, the LTE module contributes to more stable Jacobian estimation and allows SciNO to outperform baselines more clearly as graph dimensionality increases, highlighting its scalability (see SciNO (LTE) in Table 1; additional ablation results are provided in Appendix A.5.1).

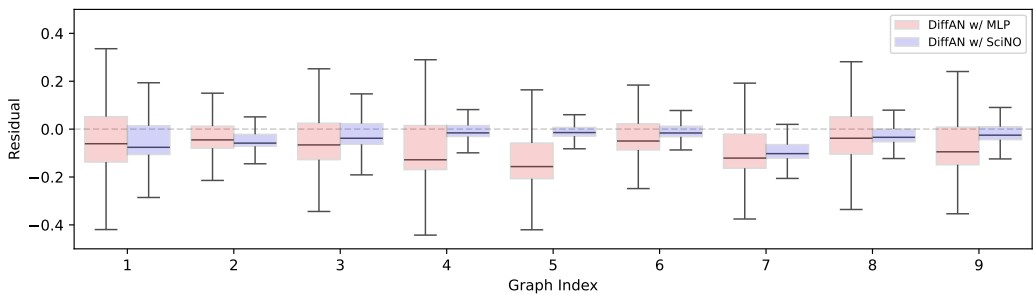

Figure 2: Boxplots of residuals in estimation error of Hessian diagonals corresponding to leaf nodes across 9 synthetic 2D datasets generated from Structural Equation Models (SEMs).

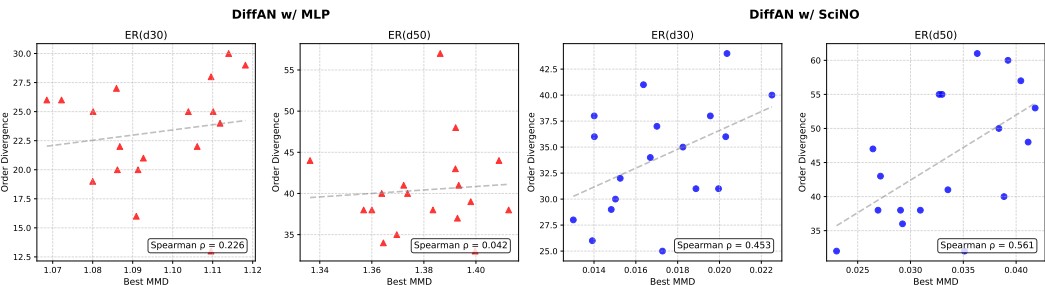

Figure 3: Comparion of correlation between the best MMD and order divergence in Erdös-Rényi synthetic datasets for $D \in \{30, 50\}$ between DiffAN [35] and DiffAN w/ SciNO.

**SciNO elaborates *more explicit* Causal Relationship** Figure 2 shows that DiffAN with SciNO consistently achieves residuals closer to zero when approximating Hessian diagonal $\mathscr{D}(\mathbf{x})$ than the original DiffAN. This result indicates that functional diffusion modeling upon SciNO provides more stable approximation of the score function and its derivatives. We also investigate whether SciNO learns causal relationship explicitly by evaluating the goodness-of-fit between generated samples and data distributions, and analyzing its alignment with causal ordering. Leveraging diffusion models on Erdös-Rényi synthetic datasets, we first generate samples by using trained score models implemented by DiffAN with MLP and SciNO. We then measure the Maximum Mean Discrepancy (MMD, [11]) between the generated samples and the data, and estimate the Spearman rank correlation between the best MMD and order divergence for each model. Figure 3 shows significant positive correlation (ER(d30): 0.453, ER(d50): 0.561) between the goodness-of-fit and causal ordering by SciNO contrary

to MLP. Overall, these findings indicate that SciNO's stable approximation of the score function and its derivatives leads the model to learn an explicit causal relationship.

**SciNO enables *memory efficient* Hessian Approximation**    For scalable causal ordering across both linear and nonlinear ANM settings, one can directly replace the score matching in DiffAN and CaPS by SciNO to approximate the Hessian diagonal $\mathscr{D}(\mathbf{x})$ through (6). Alternatively, one can apply a probing strategy built upon pretrained SciNO during the causal ordering, by freezing the Fourier layers and post-optimize $\text{MLP}_{\text{final}}$ to match the Stein gradient estimator [23]. This probing strategy proves useful when applying SciNO to CaPS, which allows fast and memory-efficient adaptation to the Hessian of marginalized density by avoiding the computation of kernel matrix inverse with whole dataset for each iteration, and does not require retraining of the entire network (Figure 4). While CaPS requires $O(N^3)$ computational complexity depending on the number of samples $N$, the probing requires $O(TB^2N)$ with training epochs $T\ (\ll N)$ and mini-batches of size $B\ (\ll N)$. We also provide the runtime comparison in Figure A.5 for empirical reference.

## 3.3   Probabilistic Control of Autoregressive Causal Ordering

While SciNO is a stable and scalable method for ordering-based causal discovery, its performance can be further enhanced by incorporating semantic information [2, 8, 21, 27, 42]. Motivated by [42] demonstrating the benefits of using LLMs for ordering tasks, we introduce a probabilistic control method that bridges ordering-based causal discovery and LLM-based reasoning.

Our approach reinterprets the sequence generation of autoregressive models, e.g. Large Language Models (LLMs), as a causal ordering process. These models generate a sequence due to the conditional distribution $\mathbb{P}_{\text{AR}}(x_{t+1}|x_{1:t}, \texttt{context})$ over the next position $x_{t+1}$ given a sequence $x_{1:t} = (x_1, \ldots, x_t)$. Here, $\texttt{context}$ refers to contextual information such as documental guidance (e.g. variable names or descriptions) or domain knowledge that helps to infer a causal structure *a priori* among the variables. From a causal ordering perspective, $x_{1:t}$ represents variables whose causal orders are predicted, while $x_{t+1}$ corresponds to a candidate variable for the next leaf node. We construct an evidence (likelihood) based on statistics estimated from observed data using SciNO, and propose a method to integrate the prior and the evidence to infer the posterior probability as follows:

$$\mathbb{P}(x_{t+1}|x_{1:t}, \texttt{stat}, \texttt{context}) \propto \mathbb{P}_{\text{AR}}(x_{t+1}|x_{1:t}, \texttt{context}) \times \mathbb{P}_{\text{SciNO}}(\texttt{stat}|x_{1:t+1}). \quad (8)$$

Here, $\texttt{stat}$ in the evidence term $\mathbb{P}_{\text{SciNO}}(\texttt{stat}|x_{1:t+1})$ denotes a statistic based on $\mathscr{D}(\mathbf{x})$, which can be used to determine the leaf node due to the identifiability criterion (3) or (4), proposed in SCORE [33] and CaPS [45], respectively. In this paper, we focus on the criterion (3) without loss of generality.

We propose two estimators for computing $\mathbb{P}_{\text{SciNO}}(\texttt{stat}|x_{1:t+1})$ based on statistics, average rank and confidence interval of $\text{Var}_{\mathbf{x}\sim\mathcal{D}}[\mathscr{D}_i(\mathbf{x})]$ of each node $i \in \mathcal{V}$. The node set $\mathcal{V}$ represents all variables in the causal graph, where $|\mathcal{V}| = D$ is the number of variables. To compute these statistics, a multiple number of variance samples is essential and can be obtained by applying the MCMC, bootstrapping, or ensemble technique. We apply Deep Ensembles [19] to train multiple independent models with different initializations. Note that due to the probing strategy, one can train multiple models from a single pretrained SciNO, enabling a memory-efficient ensemble strategy. Let $\sigma_i^{(m)}$ denote $\sqrt{\text{Var}_{\mathbf{x}\sim\mathcal{D}}[\mathscr{D}_i(\mathbf{x})]}$ for a given independently trained model $m \in \{1, \ldots, M\}$ and node $i \in \mathcal{V}$.

**Rank-based Control**    Rank-based control uses the statistic of the rank of $\text{Var}_{\mathbf{x}\sim\mathcal{D}}[\mathscr{D}_i(\mathbf{x})]$:

$$\mathbb{P}_{\text{SciNO}}^{(\text{rank})}(\text{Var}_{\mathbf{x}\sim\mathcal{D}}[\mathscr{D}_i(\mathbf{x})] < \text{Var}_{\mathbf{x}\sim\mathcal{D}}[\mathscr{D}_j(\mathbf{x})]|x_{1:t}, x_{t+1} = i), \quad j \in \mathcal{V} \setminus \pi_t. \quad (9)$$

Let $\pi_t$ denote the partial causal order fixed by $x_{1:t}$. Motivated from Plackett–Luce model [32], we approximate (9) by computing the average rank $\bar{r}(i)$ for each node $i \in \mathcal{V}$ across models:

$$\widehat{P}_{m\sim M}^{(\text{rank})}(\sigma_i^{(m)} < \sigma_j^{(m)}|x_{1:t}, x_{t+1} = i) = \frac{\exp\left(-\bar{r}(i)\right)}{\sum_{j\in\mathcal{V}\setminus\pi_t}\exp\left(-\bar{r}(j)\right)}, \quad \bar{r}(i) = \frac{1}{M}\sum_{m=1}^{M} r_i^{(m)}. \quad (10)$$

Here $r_i^{(m)}$ denotes the rank of $\sigma_i^{(m)}$, sorted in ascending order for each model. A lower average rank implies that the node consistently exhibits minimal variance across models. Rank-based estimator (10) provides a scale-invariant and outlier-robust way to compare variances across models.

**Confidence Interval-based Control** Due to the Central Limit Theorem (CLT), the sample mean of $(\sigma_i^{(m)})^2$ approximates a Gaussian distribution, from which we derive the confidence interval $CI(i)$. Let $CI_{lower}(i)$ and $CI_{upper}(i)$ denote the lower and the upper bound of the confidence interval. Confidence interval-based control utilizes the following statistic of the uncertainty in $(\sigma_i^{(m)})^2$ estimation:

$$\mathbb{P}_{SciNO}^{(CI)}(CI_{lower}(i) \leq CI_{upper}(j_*)|x_{1:t}, x_{t+1} = i), \quad j_* = \underset{j \in \mathcal{V} \setminus \pi_t}{\arg\min} \, \text{Var}_{\mathbf{x} \sim \mathcal{D}} \left[ \mathscr{D}_j(\mathbf{x}) \right]. \quad (11)$$

We approximate the probability (11) using the following empirical estimator:

$$\widehat{P}_{m \sim M}^{(CI)}(CI_{lower}(i) \leq CI_{upper}(j_{min}^{(m)})|x_{1:t}, x_{t+1} = i) = \frac{1}{M} \sum_{m=1}^{M} \mathbb{1} \left[ CI_{lower}(i) \leq CI_{upper}(j_{min}^{(m)}) \right]. \quad (12)$$

Here, $j_{min}^{(m)}$ denotes the node $j$ of minimal $\sigma_j^{(m)}$, and $\mathbb{1}[\cdot]$ is the indicator function. Instead of directly comparing variance estimates, estimator (12) can account for the uncertainty across different models.

---

**Algorithm 1:** Probabilistic Control of Autoregressive Causal Ordering

**Input:** Variable list $\mathcal{V} = \mathcal{V}_{context} \cup \mathcal{V}_{\neg context}$
**Output:** Causal order $\pi$

1  Initialize $\pi \leftarrow \emptyset$
2  **while** $\mathcal{V}_{-\pi} \neq \emptyset$ **do**
3     Let $\mathbf{x}_\pi \leftarrow (x_{\pi(i)} : i = 1, \ldots, |\pi|)$ and set $\mathcal{V}_{-\pi} := \mathcal{V} \setminus \pi$
4     **foreach** $v \in \mathcal{V}_{-\pi}$ **do**
5        Compute prior $P_{AR}(v|\mathbf{x}_\pi, \texttt{context})$
6        Compute evidence $\widehat{P}_{SciNO}(v)$ ;                           `// by (10) or (12)`
7        **if** $v \in \mathcal{V}_{context}$ **then**
8           Compute $P(v) \leftarrow P_{AR}(v|\mathbf{x}_\pi, \texttt{context}) \cdot \widehat{P}_{SciNO}(v)$;    `// soft supervision`
9        **else**
10          Compute $P(v) \leftarrow \frac{1}{|\mathcal{V}_{-\pi}|} \widehat{P}_{SciNO}(v)$;          `// hard supervision`
11    Select leaf node $v^* = \underset{v \in \mathcal{V}_{-\pi}}{\arg\max} \, P(v)$ and update $\pi \leftarrow \pi \cup \{v^*\}$
12 **return** $\pi$

---

Algorithm 1 presents our probabilistic control of autoregressive causal ordering, by using the estimators proposed in (10) or (12). Each variable belongs to either $\mathcal{V}_{context}$, which contains variables with contextual information or domain knowledge, or $\mathcal{V}_{\neg context}$ otherwise. The posterior computation differs depending on which set the variable belongs to. For $v \in \mathcal{V}_{context}$, the algorithm applies *soft supervision* which updates the prior prediction of autoregressive model by multiplying the evidence term (10) or (12). If variables in $v \in \mathcal{V}_{\neg context}$, then the algorithm applies *hard supervision* by imposing a uninformative prior, which selects a leaf node based on the evidence term. These supervisions enable the autoregressive model to operate reliably with mixed levels of contextual information.

## 4 Empirical Results

We evaluate whether the proposed method improves existing ordering-based methods, DiffAN [35] and CaPS [45]. Additional baseline comparisons are provided in Appendix A.5.2. Given the inferred order, we apply a pruning based on feature selection [4] to compare the structural accuracies achieved by the resulting causal graphs across different ordering methods. We also validate causal ordering performances by applying a probabilistic control to LLMs and analyze the resulting changes in probability distributions. Evaluation metrics include Order Divergence (OD, [33]) for causal ordering, Structural Hamming Distance (SHD, [29]) and Structural Intervention Distance (SID, [41]) for causal discovery. See Appendix A.4.2 for details on each metric.

### 4.1 Synthetic Data: Erdös-Renyi Random Graphs

**Dataset** We generate causal DAGs using the Erdös–Rényi (ER) model [9]. For a graph with $D$ nodes, we set the expected number of edges to $4D$. We sample node values according to a nonlinear

ANM with Gaussian noise, where the functions are generated by a Gaussian Process with a Radial Basis Function (RBF) kernel with fixed bandwidth 1. To evaluate model's scalability, we vary the number of nodes across 2, 3, 5, 10, 30, 50, and 100. For each case, we generate 10 random graphs and produce 1,000 samples per graph.

**Results**  Table 2 shows that SciNO improves the performance of DiffAN in most metrics, particularly in terms of OD, and remains scalable in high-dimensional settings, notably reducing OD from 117.0 to 86.6 when $D = 100$. The top line graphs in Figure A.4 show the cumulative order divergence, which measures the inaccurate prediction over the leaf nodes. Original DiffAN accumulates OD more rapidly as the ordering progresses, while SciNO maintains more accurate prediction, particularly in high-dimensional settings ($D \in \{30, 50\}$). The bottom of heatmap in Figure A.4 visualizes the ranked variance of the estimated Hessian diagonal for each variable at ordering steps $\{10, 20, 30\}$ on ER graphs with 30 nodes. SciNO provides more accurate leaf node predictions and assigns consistently lower variances to ground-truth leaves, due to the stable estimation of the Hessian diagonal.

Table 2: Comparison of causal discovery metrics(OD/SHD/SID) between DiffAN [35] and DiffAN with SciNO in (a) synthetic datasets and (b) real and semi-synthetic datasets: Physics, Sachs, and BNLearn with *nonlinear* ANM. Each score is recorded over 10 random graphs for synthetic datasets, and over 10 independent runs for real and semi-synthetic datasets.

| Dataset\Metric | DiffAN [35] | | | DiffAN w/ SciNO (Ours) | | |
|---|---|---|---|---|---|---|
| | OD ($\downarrow$) | SHD ($\downarrow$) | SID ($\downarrow$) | OD ($\downarrow$) | SHD ($\downarrow$) | SID ($\downarrow$) |
| ER(d2) | $0.2 \pm 0.4$ | $0.3 \pm 0.64$ | $0.3 \pm 0.64$ | $\mathbf{0.0} \pm 0.0$ | $\mathbf{0.0} \pm 0.0$ | $\mathbf{0.0} \pm 0.0$ |
| ER(d3) | $0.4 \pm 0.49$ | $0.7 \pm 0.9$ | $1.1 \pm 1.58$ | $\mathbf{0.1} \pm 0.3$ | $\mathbf{0.2} \pm 0.6$ | $\mathbf{0.4} \pm 1.2$ |
| ER(d5) | $1.1 \pm 1.22$ | $1.8 \pm 1.78$ | $3.6 \pm 4.32$ | $\mathbf{0.9} \pm 1.14$ | $\mathbf{1.7} \pm 1.95$ | $4.8 \pm 4.94$ |
| ER(d10) | $3.2 \pm 1.99$ | $21.9 \pm 3.47$ | $47.8 \pm 10.20$ | $\mathbf{2.7} \pm 1.19$ | $\mathbf{20.5} \pm 3.5$ | $\mathbf{41.6} \pm 9.31$ |
| ER(d30) | $26.7 \pm 8.1$ | $94.2 \pm 17.57$ | $546.3 \pm 82.80$ | $\mathbf{16.9} \pm 6.12$ | $\mathbf{88.8} \pm 16.5$ | $\mathbf{492.0} \pm 91.64$ |
| ER(d50) | $45.5 \pm 7.51$ | $184.2 \pm 14.63$ | $1690.3 \pm 133.55$ | $\mathbf{32.8} \pm 6.55$ | $\mathbf{180.0} \pm 15.45$ | $\mathbf{1622.0} \pm 110.92$ |
| ER(d100) | $117.0 \pm 12.78$ | $463.3 \pm 27.64$ | $7562.2 \pm 435.94$ | $\mathbf{86.6} \pm 12.99$ | $\mathbf{445.9} \pm 32.84$ | $\mathbf{7259.2} \pm 653.27$ |
| Physics(d7) | $3.3 \pm 0.78$ | $8.6 \pm 2.06$ | $16.8 \pm 4.73$ | $\mathbf{1.9} \pm 0.54$ | $\mathbf{5.8} \pm 1.60$ | $\mathbf{8.2} \pm 4.62$ |
| Sachs(d8) | $5.7 \pm 2.69$ | $22.8 \pm 4.56$ | $26.1 \pm 10.34$ | $\mathbf{5.6} \pm 2.65$ | $23.1 \pm 3.86$ | $\mathbf{23.3} \pm 9.10$ |
| MAGIC-NIAB(d44) | $8.8 \pm 6.85$ | $89.1 \pm 25.85$ | $227.2 \pm 157.64$ | $\mathbf{4.1} \pm 0.70$ | $\mathbf{72.0} \pm 4.52$ | $\mathbf{125.5} \pm 28.08$ |
| ECOLI70(d46) | $21.8 \pm 4.26$ | $111.0 \pm 11.47$ | $684.8 \pm 80.67$ | $\mathbf{12.8} \pm 3.19$ | $\mathbf{90.3} \pm 7.13$ | $\mathbf{500.9} \pm 58.16$ |
| MAGIC-IRRI(d64) | $12.0 \pm 3.87$ | $146.9 \pm 13.46$ | $321.9 \pm 64.64$ | $\mathbf{10.6} \pm 1.69$ | $\mathbf{144.5} \pm 4.43$ | $\mathbf{254.4} \pm 39.66$ |
| ARTH150(d107) | $43.3 \pm 15.77$ | $613.3 \pm 72.49$ | $2456.0 \pm 953.08$ | $\mathbf{21.4} \pm 2.76$ | $\mathbf{515.6} \pm 15.26$ | $\mathbf{1186.7} \pm 104.19$ |

Table 3: Comparison of causal discovery metrics(OD/SHD/SID) between CaPS [45] and CaPS with SciNO in BNLearn datasets with *linear* ANM. Each score is recorded over 10 independent runs.

| Dataset\Metric | CaPS [45] | | | CaPS w/ SciNO (Ours) | | |
|---|---|---|---|---|---|---|
| | OD ($\downarrow$) | SHD ($\downarrow$) | SID ($\downarrow$) | OD ($\downarrow$) | SHD ($\downarrow$) | SID ($\downarrow$) |
| MAGIC-NIAB(d44) | 36 | 160 | 1024 | $37.5 \pm 0.50$ | $165.4 \pm 0.50$ | $1185.9 \pm 5.61$ |
| ECOLI70(d46) | 25 | 134 | 786 | $25.4 \pm 0.80$ | $141.3 \pm 4.78$ | $882.4 \pm 13.02$ |
| MAGIC-IRRI(d64) | 42 | 179 | 1192 | $\mathbf{41.6} \pm 0.80$ | $183.5 \pm 2.97$ | $1280.7 \pm 36.83$ |
| ARTH150(d107) | 49 | 406 | 3051 | $48.2 \pm 0.87$ | $\mathbf{380.0} \pm 4.63$ | $\mathbf{2799.1} \pm 73.26$ |

## 4.2 Real and Semi-Synthetic Data

**Dataset**  We use three datasets to evaluate causal structure learning in real-world graphs: (i) A Physics commonsense-based synthetic dataset comprising 5,000 nonlinear-SEM samples generated over a 7-variable water-evaporation DAG [20]. (ii) The Sachs flow-cytometry dataset [34] comprises 7,466 single-cell measurements of 8 proteins after removing cyclic terminal nodes. (iii) The BNLearn collection of four real-world graphs— MAGIC-NIAB [38] (44 nodes/ 66 edges), ECOLI70 [36] (46 nodes/ 70 edges), MAGIC-IRRI [37] (64 nodes/ 102 edges), ARTH150 [28] (107 nodes/ 150 edges)—each simulated with 10,000 samples under both linear and nonlinear Structural Equation Models (SEMs). Detailed sampling procedures are provided in the Appendix A.4.1.

**Results**  Consistent with the results on ER synthetic graphs, DiffAN with SciNO outperforms the MLP-based model across all real datasets (see Table 2). As discussed in Section 3.2, SciNO can be applied to both methods that assume nonlinear relationships, such as DiffAN [35], and methods based on linear structures, such as CaPS [45]. Table 3 compares the performances of the original

CaPS and CaPS with probed SciNO on BNLearn datasets under linear ANM. SciNO achieves comparable performances across all datasets in terms of OD, with improved results observed on higher-dimensional graphs such as MAGIC-IRRI and ARTH150. As shown in Figure 4, while CaPS fails to scale to large datasets due to memory constraints, probed SciNO exhibits significantly lower memory usage.

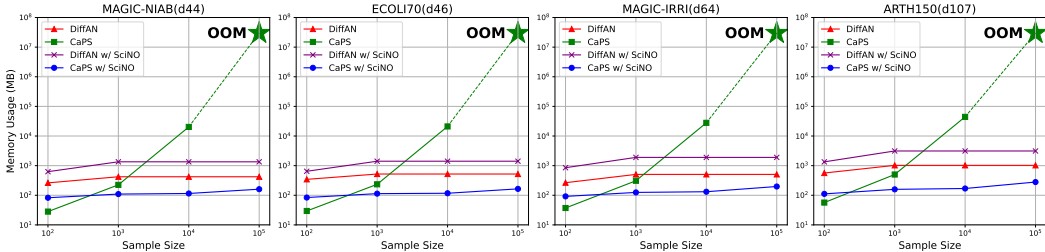

Figure 4: Comparison of GPU memory usage of DiffAN [35], CaPS [45], SciNO during the causal ordering process while scaling up the sample size. SciNO demonstrates memory efficiency while CaPS suffers out-of-memory (OOM) errors when the number of samples is larger than 100,000.

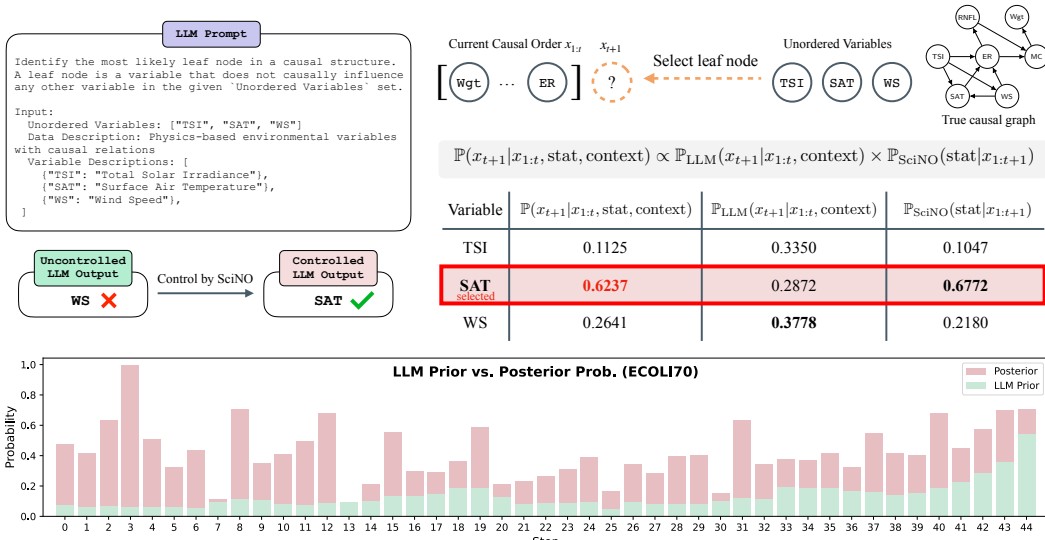

Figure 5: **(Top)** Overview of the LLM control. The LLM is prompted to output the most likely leaf node among unordered variables. Prior probabilities are multiplied by SciNO's data-informed evidence terms to compute posteriors. The variable with the highest posterior probability is selected as the leaf node. **(Bottom)** Updated posterior probabilities ■ obtained by the SciNO-based control versus LLM prior probabilities ■ of ground-truth leaf nodes at each causal reasoning step (ECOLI70).

## 4.3 Controlling LLM for Causal Ordering

**Setting** We prompt LLMs (Llama-3.1-8B-Instruct [10], GPT-4o [14]) to predict the leaf node among unordered variables, then LLMs output the prior probability $\mathbb{P}_{\text{LLM}}(x_{t+1}|x_{1:t}, \texttt{context})$ in an autogressive manner (see Figure 5). Compared to pairwise prompt methods [1, 8, 15, 20, 27], this approach reduces the number of LLM calls from $\mathcal{O}(|\mathcal{V}|^2)$ to $\mathcal{O}(|\mathcal{V}|)$. To mitigate bias due to the variable name length, we apply a length normalization for $\alpha \in \{0.5, 1.0\}$. Full experimental details are provided in Appendix A.6.1.

We evaluate the impact of evidence quality by computing the evidence term $\mathbb{P}(\texttt{stat}|x_{1:t+1})$ with three models: SCORE [33], DiffAN [35], and SciNO. For this computation, we employ the estimators from (10) and (12). While DiffAN and SciNO are implemented with Deep Ensembles of $M = 30$ models, SCORE uses bootstrapping with 1000 replicates due to its incompatibility with Deep Ensembles.

Table 4: Comparison of the OD in BNLearn datasets between GPT-4o, uncontrolled Llama, and controlled Llama via SciNO. Differences are expressed as percent change relative to the uncontrolled result. Each score is recorded over 10 independent runs.

| Method\Dataset | MAGIC-NIAB(d44) | ECOLI70(d46) | MAGIC-IRRI(d64) | ARTH150(d107) |
|---|---|---|---|---|
| GPT-4o | $11.9 \pm 0.94$ | $41.1 \pm 2.30$ | $20.9 \pm 2.30$ | $77.1 \pm 7.46$ |
| Llama-3.1-8b(1-norm) | 9 | 32 | 18 | 80 |
| w/ control(rank) | $3.2 \pm 0.40$ (▼64.4%) | $10.6 \pm 0.49$ (▼66.9%) | **9.9** $\pm 0.54$ (▼45.0%) | $20.4 \pm 0.92$ (▼74.5%) |
| w/ control(CI) | **2.0** $\pm 0.00$ (▼77.8%) | **9.9** $\pm 1.97$ (▼69.1%) | $11.1 \pm 0.83$ (▼38.3%) | **18.9** $\pm 1.14$ (▼76.4%) |
| Llama-3.1-8b(0.5-norm) | 14 | 32 | 17 | 80 |
| w/ control(rank) | $4.1 \pm 0.83$ (▼70.7%) | $10.5 \pm 0.92$ (▼67.2%) | $9.1 \pm 0.30$ (▼46.5%) | $20.4 \pm 0.92$ (▼74.5%) |
| w/ control(CI) | **2.9** $\pm 0.30$ (▼79.3%) | **9.8** $\pm 1.83$ (▼69.4%) | **10.1** $\pm 0.83$ (▼40.6%) | **18.9** $\pm 1.14$ (▼76.4%) |

**Results**  Llama-3.1 and GPT-4o, recognized as high-performance reasoning LLMs, exhibit significant limitations when confronted with high-dimensional causal graphs. Figure 5 shows the LLM prior and updated posterior probabilities of ground-truth leaf nodes at each reasoning step on the ECOLI70 dataset. Prior probabilities remain predominantly below 0.2, indicating poor predictive performance when faced with a large number of variables. In contrast, updated posterior probabilities in overall steps show that LLM control can reduce inaccurate leaf prediction, thereby enhancing causal reasoning. Consequently, Table 4 shows that the SciNO-based control improves performance across all datasets with more than 40 nodes, achieving up to 75% improvement on ARTH150. Note that the control method also outperforms the results reported in Table 2, which indicates that incorporating semantic information contributes in causal ordering.

We benchmarked our SciNO-based control against the same algorithm using evidence from DiffAN and SCORE. The full results are presented in Table A.7. On high-dimensional datasets, the SciNO-based control yields a 64% average OD reduction, substantially higher than the 46% reduction from DiffAN. This indicates that the performance gain is attributable to both the novel control method and reliable evidence provided by SciNO. The results in Table 4 are obtained under the setting where all variables are well-described, i.e., $\mathcal{V}_{\text{context}} = \mathcal{V}$ in Algorithm 1. See Appendix A.6.2 for other settings when $\mathcal{V}_{\text{context}} \neq \mathcal{V}$.

## 5   Conclusion and Limitation

We propose SciNO, a functional diffusion model framework that enables accurate causal ordering through stable approximation of the Hessian diagonal. SciNO consistently enhances existing score matching based causal ordering methods in both accuracy and scalability across diverse synthetic and real-world causal graph benchmarks. We further propose a probabilistic control method for supervising autoregressive generative models to generate more reliable causal reasoning by using score-informed statistics estimated by SciNO. In real-world decision making, the inaccuracy of causal discovery algorithms may cause detrimental consequences. Our method improves the reliability of causal ordering by contributing to more trustworthy reasoning and safer downstream applications. While SciNO demonstrates its effectiveness in various causal ordering tasks, several aspects of its applicability remain to be explored. First, the score matching based causal discovery frameworks mainly assume ANMs, whose identifiability is restricted within additive functional relations. Second, SciNO is currently limited to continuous random variables. Future research will explore extending the framework to accommodate discrete or mixed-type datasets, as well as multimodal or structured inputs such as images, text, or temporal sequences.

## Acknowledgments and Disclosure of Funding

This work was supported by LG AI Research and also supported by the NRF grant funded by the Korea government (MSIT) (No.RS-2024-00410082, 50%) and (No. NRF-2022M3J6A1063595), and also supported by Institute of Information & communications Technology Planning & Evaluation(IITP) grant funded by the Korea government (MSIT)(No.2022-0-00612, 50%), (No. RS-2024-00436857, ITRC, 20%), (No.RS-2025-02304828, Artificial Intelligence Star Fellowship Support Program, 15%) and (No.RS-2019-II190079, Artificial Intelligence Graduate School Program (Korea University)).

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

# A  Supplementary Material

In this supplementary material, we provide the information for understanding SciNO with detailed explanations. The supplementary material is organized as follows: Section A.1 introduces related works in ordering based causal discovery, score matching methods, and LLM-assisted causal discovery. Section A.2 provides theoretical derivations and proofs for preliminaries and the proposed method. Section A.3 explains a detailed description of the architecture of SciNO with layer-by-layer formulations on how to implement it. Section A.4 describes the experimental setup, datasets, evaluation metrics and implementation details of empirical results in Sections 4.1 and 4.2. Section A.5 presents additional ablation studies and comparisons with causal discovery baselines. Section A.6 presents experimental details in Section 4.3 and additional results on controlling LLMs for causal ordering.

## A.1  Related Works

### A.1.1  Ordering-based Causal Discovery

Ordering-based causal discovery methods are typically categorized into score matching based and non-score matching based approaches. SCORE [33], DiffAN [35], and CaPS [45] are score matching based methods that estimate the Hessian of the log density to identify leaf nodes and construct causal order. Non-score matching methods such as CAM [4] and GSPo[3] recover the causal order using functional assumptions and conditional independence tests, respectively. In this work, we focus on the score matching based approaches [33, 35, 45], which estimate gradients of the log-density for causal ordering.

### A.1.2  Score Matching and Score-based Generative Models

A common approach to estimating the score function of a data distribution is to use a gradient estimator based on the Stein identity [23]. Stein kernel-based methods allow score estimation without an explicit density and have been used in SCORE [33] and CaPS [45]. However, these methods require computing the inverse of a kernel matrix, leading to high computational and memory costs. In recent work, Diffusion models [13, 39] have emerged as a key method for score estimation in generative modeling. However, the commonly used Denoising Diffusion Model [13, 39] is defined over finite-dimensional spaces and thus struggle to handle functional data such as derivatives. Functional diffusion models [25, 26] address this issue by defining the diffusion process over a Hilbert space and learning it directly via score-based modeling. [25, 26] utilize the Neural Operators [16, 24] to learn mappings between functions, allowing the model to capture both the functional form and its derivatives.

### A.1.3  Causal Discovery with Large Language Models

Recent studies leverage domain knowledge inferred from LLM as priors or constraints to enhance data-driven causal discovery [2, 8, 20, 21, 27, 42]. Most methods adopt pairwise prompts to construct causal graphs via repeated queries on variable pairs [1, 8, 15, 20, 27]. To address inability of LLMs to distinguish between direct and indirect effects, [42] introduces a triplet prompt that queries relationships among three variables simultaneously, deriving causal order rather than graph structures. Existing methods treat LLMs as a black-box and directly convert generated tokens into binary decisions. This obscures uncertainty and limits the scientific explanatory power of causal inference. [8] partially addresses this by extracting pairwise edge-direction probabilities from the LLM and combining them with mutual information, but this approach remains limited, as it does not directly reflect causal relationships, limiting its explanatory power.

## A.2  Proof of Theorems

### A.2.1  Derivation of the Hessian diagonal approximation

In this section, we derive approximations (5) and (6), which are suggested by DiffAN [35], for the sake of completeness. Under the ANM assumption (2), the score function $\mathbf{S}(\mathbf{x})$ can be derived as:

$$\mathbf{S}_j(\mathbf{x}) = \partial_{x_j} \log P(\mathbf{x}) = \partial_{x_j} \log \prod_{i=1}^{D} P(x_i \mid \mathrm{Pa}(x_i)) = \partial_{x_j} \sum_{i=1}^{D} \log P(x_i \mid \mathrm{Pa}(x_i)), \quad j \in \mathcal{V}, \quad (13)$$

where $D$ denotes the number of nodes $|\mathcal{V}|$. Let $\epsilon_i = x_i - f_i$. By the change of variable, it holds that

$$\partial_{x_j} \sum_{i=1}^{D} \log P(x_i \mid \mathrm{Pa}(x_i)) = \partial_{x_j} \sum_{i=1}^{D} \log p_i(x_i - f_i) \tag{14}$$

$$= \frac{\partial \log p_j(x_j - f_j)}{\partial x_j} - \sum_{i \in \mathrm{Ch}(x_j)} \frac{\partial f_i}{\partial x_j} \frac{\partial \log p_i(x_i - f_i)}{\partial \epsilon_i}. \tag{15}$$

When a leaf node $x_l$ is marginalized out, the resulting score $\mathbf{S}_j(\mathbf{x}_{-l})$ for $x_j \in \mathrm{Pa}(x_l)$ becomes:

$$\mathbf{S}_j(\mathbf{x}_{-l}) = \mathbf{S}_j(\mathbf{x}) + \frac{\partial f_l}{\partial x_j} \frac{\partial \log p_l(x_l - f_l)}{\partial \epsilon_l}. \tag{16}$$

Let $\delta_{j,l}$ denote the residue term added to the score of $x_j$ upon marginalizing out $x_l$:

$$\delta_{j,l} := \frac{\partial f_l}{\partial x_j} \cdot \frac{\partial \log p_l(x_l - f_l)}{\partial \epsilon_l}. \tag{17}$$

Note that the residue term $\delta_{j,l}$ vanishes if $x_j \notin \mathrm{Pa}(x_l)$. For a leaf node $x_l$, it holds that:

$$\mathbf{S}_l(\mathbf{x}) = \frac{\partial \log p_l(x_l - f_l)}{\partial x_l} = \frac{\partial \log p_l(x_l - f_l)}{\partial \epsilon_l} \cdot \frac{\partial(x_l - f_l)}{\partial x_l} = \frac{\partial \log p_l(x_l - f_l)}{\partial \epsilon_l}. \tag{18}$$

The partial derivatives of the score $\mathbf{S}_l(\mathbf{x})$ with respect to $x_j$ for any $j \in \mathcal{V}$ are given by:

$$\partial_{x_j} \mathbf{S}_l(\mathbf{x}) = \frac{\partial^2 \log p_l(x_l - f_l)}{\partial \epsilon_l^2} \cdot \frac{\partial f_l}{\partial x_j}. \tag{19}$$

In particular, when $j = l$, this reduces to:

$$\partial_{x_l} \mathbf{S}_l(\mathbf{x}) = \frac{\partial^2 \log p_l(x_l - f_l)}{\partial \epsilon_l^2}. \tag{20}$$

Thus, $\delta_{j,l}$ can be rewritten as:

$$\delta_{j,l} = \frac{\partial f_l}{\partial x_j} \frac{\partial \log p_l(x_l - f_l)}{\partial \epsilon_l} \tag{21}$$

$$= \frac{\partial^2 \log p_l(x_l - f_l)}{\partial \epsilon_l^2} \cdot \frac{\partial f_l}{\partial x_j} \cdot \frac{\partial \log p_l(x_l - f_l)}{\partial \epsilon_l} \bigg/ \frac{\partial^2 \log p_l(x_l - f_l)}{\partial \epsilon_l^2} \tag{22}$$

$$= \partial_{x_j} \mathbf{S}_l(\mathbf{x}) \cdot \frac{\mathbf{S}_l(\mathbf{x})}{\partial_{x_l} \mathbf{S}_l(\mathbf{x})} \tag{23}$$

Let $\pi_k \subset \pi$ denote the set of selected leaf nodes up to step $k$, and $-\pi_k = \mathcal{V} \setminus \pi_k$ denote the remaining ones. The deciduous score on the remaining variables $\mathbf{S}(\mathbf{x}_{-\pi_k}) \in \mathbb{R}^{D-|\pi_k|}$ are approximated by:

$$\mathbf{S}_j(\mathbf{x}_{-\pi_k}) = \mathbf{S}_j(\mathbf{x}) + \sum_{l \in \pi_k} \left( \partial_{x_j} \mathbf{S}_l(\mathbf{x}) \cdot \frac{\mathbf{S}_l(\mathbf{x})}{\partial_{x_l} \mathbf{S}_l(\mathbf{x})} \right), \quad j \in \mathcal{V} \setminus \pi_k. \tag{24}$$

Hence we obtain (5). Finally, the Hessian diagonal $\mathscr{D}(\mathbf{x}_{-\pi_k}) = \left( \partial_{x_j} \mathbf{S}_j(\mathbf{x}_{-\pi_k}) : j \in \mathcal{V} \setminus \pi_k \right)$ is:

$$\mathscr{D}_j(\mathbf{x}_{-\pi_k}) = \partial_{x_j} \mathbf{S}_j(\mathbf{x}) + \sum_{l \in \pi_k} \partial_{x_j} \left( \partial_{x_j} \mathbf{S}_l(\mathbf{x}) \cdot \frac{\mathbf{S}_l(\mathbf{x})}{\partial_{x_l} \mathbf{S}_l(\mathbf{x})} \right) \tag{25}$$

$$\approx \partial_{x_j} \widehat{\mathbf{S}}_j^\theta(t, \mathbf{x}) + \sum_{l \in \pi_k} \partial_{x_j} \left( \partial_{x_j} \widehat{\mathbf{S}}_l^\theta(t, \mathbf{x}) \cdot \frac{\widehat{\mathbf{S}}_l^\theta(t, \mathbf{x})}{\partial_{x_l} \widehat{\mathbf{S}}_l^\theta(t, \mathbf{x})} \right), \quad j \in \mathcal{V} \setminus \pi_k. \tag{26}$$

Thus, we obtain the approximation (6).

**Remark A.1.** Note that (25) requires the second-order derivatives of $\mathbf{S}(\mathbf{x})$ and so does approximation (6) replacing the score function by the score model $\widehat{\mathbf{S}}^\theta(t, \mathbf{x})$.

### A.2.2 Approximation Power of Neural Operators

**Score-based Generative Modeling in Function Spaces** For readers who are not familiar with functional diffusion modeling, we briefly introduce a time-reversal theory of score-based generative modeling in Hilbert spaces [26].

Consider the following stochastic evolution equations in the Hilbert space $\mathcal{H}$:

$$d\mathbf{X}_t = \mathbf{B}_t(\mathbf{X}_t)dt + \mathbf{G}_t d\mathbf{W}_t, \quad d\widehat{\mathbf{X}}_t = \widehat{\mathbf{B}}_t(\widehat{\mathbf{X}}_t)dt + \widehat{\mathbf{G}}_t d\mathbf{W}_t, \quad t \in [0, T], \tag{27}$$

where $(\mathbf{X}_t, \widehat{\mathbf{X}}_t)$ is a pair of forward and reverse stochastic processes with coefficients $(\mathbf{B}_t, \mathbf{G}_t)_{t \in [0,T]}$ and $(\widehat{\mathbf{B}}_t, \widehat{\mathbf{G}}_t)_{t \in [0,T]}$, and $\mathbf{W}_t$ is an $\mathcal{H}$-valued Wiener process. [26, Theorem 2.1] provides the time-reversal formula which connects pairs $(\mathbf{X}_t, \widehat{\mathbf{X}}_t)$ to have the same marginal distribution:

$$\widehat{\mathbf{B}}_t(u) = -\mathbf{B}_{T-t}(u) + \mathbf{S}_{T-t}(u), \quad \widehat{\mathbf{G}}_t = \mathbf{G}_{T-t}, \quad t \in [0, T]. \tag{28}$$

Here $\mathbf{S}_t$ denotes the score operator for each marginal probability measures $(\mu_t)_{t \in [0,T]}$ of the solution to the forward equation. Therefore, we can generate samples from $\widehat{\mathbf{X}}_T \sim \mathbb{P}_{\text{data}}$ and approximate the score operator of $\mathbb{P}_{\text{data}}$ in function spaces by training the score model $\widehat{\mathbf{S}}^\theta(t, \cdot)$ and running the reverse equation with replacing the score operator $\mathbf{S}(t, \cdot)$ in (28) by the trained score model $\widehat{\mathbf{S}}^\theta(t, \cdot)$ for each $t \in [0, T]$. Therefore, the learning objective of score-based generative modeling in function spaces is equivalent to the score matching between $\mathbf{S}(t, \cdot)$ and $\widehat{\mathbf{S}}^\theta(t, \cdot)$ in the Hilbert space $\mathcal{H}$ [25, 26].

**Proof of Theorem 3.1** To explain the statement in Theorem 3.1 precisely, let us introduce necessary definitions and theorems. We first introduce Sobolev spaces and Sobolev embedding theorem [17], which are main elements for understanding a fundamental theory in neural operators [16].

**Definition A.1** ([17, Chapter 1.3]). Let $W_2^k = W_2^k(\mathcal{X})$ be the Sobolev space with the norm:

$$\|f\|_{W_2^k} := \sum_{|\alpha| \leq k} \|\partial_{\mathbf{x}}^\alpha f\|_{L_2(\mathcal{X})} \tag{29}$$

with the inner-product:

$$\langle f, g \rangle_{W_2^k} := \sum_{|\alpha| \leq k} \langle \partial_{\mathbf{x}}^\alpha f, \partial_{\mathbf{x}}^\alpha g \rangle_{L_2}. \tag{30}$$

**Remark A.2.** Note that the inner-product (30) induces the norm which is equivalent to the Sobolev norm (29). Due to the Sobolev embedding theorem [17, Chapter 10], it holds that

$$W_2^k \subset C^{k - \frac{D}{2}}, \tag{31}$$

where $C^{k - \frac{D}{2}}$ denotes the Hölder space with the following Hölder-norm:

$$\|f\|_{C^{k - \frac{D}{2}}} := \max_{|\alpha| \leq m} \sup_{\mathbf{x} \in \mathcal{X}} \left\| \partial_{\mathbf{x}}^\beta f(\mathbf{x}) \right\| + \max_{|\beta| = m} \left\| \partial_{\mathbf{x}}^\beta f \right\|_{C^{k - \frac{D}{2} - m}}, \quad m = \lfloor k - \frac{D}{2} \rfloor. \tag{32}$$

Sobolev embedding (31) implies that for any $f \in W_2^k$, it holds

$$\|f\|_{C^{k - \frac{D}{2}}} \lesssim \|f\|_{W_2^k}, \tag{33}$$

where the notation $\lesssim$ means the inequality $\leq$ holds up to constant factors. Therefore, we can approximate a function including its derivatives by (33) if we can control the Sobolev norm (29).

Let us set $a := (\mathbf{B}_t, \mathbf{G}_t)_{t \in [0,T]}$ as *smooth* coefficients defined on $[0, T]$ and consider their function class as $\mathcal{A} = C([0, T])$, which determines the probability measure $\mu_T$ by simulating the forward stochastic evolution equation $\mathbf{X}_t$ in (27). For a given $a \in \mathcal{A}$, let us define an operator $\mathcal{G}^\dagger(a) = \mathbf{S}_*$ which maps smooth coefficients $a$ to the target score function $\mathbf{S}_* = \nabla_{\mathbf{x}} \log P_{\text{data}}$ of the data distribution $\mathbb{P}_{\text{data}}$ on the data space $\mathcal{X}$. In this paper, we assume the target score function satisfies $\mathbf{S}_* \in W_2^k(\Omega)$ for any Lipschitz domain $\Omega \subset \mathcal{X}$ where $k > 2 + \frac{D}{2}$. This is a natural assumption for many probability densities which are continuously differentiable on Lipschitz subsets of $\mathcal{X}$.

**Theorem A.1.** *Suppose the map $\mathcal{G}^\dagger : C([0,T]) \to W_2^k(\Omega)$ is continuous for any Lipschitz subset $\Omega \subset \mathcal{X}$ such that $\mathcal{G}^\dagger(a) = \mathbf{S}_*$. Then for any $\varepsilon > 0$ and compact set $\bar{A} \subset \mathcal{A}$, there exists a neural operator $\mathcal{G} : C([0,T]) \to W_2^k(\Omega)$ such that*

$$\sup_{a \in \bar{A}} \left\| \mathcal{G}^\dagger(a) - \mathcal{G}(a) \right\|_{W_2^k(\Omega)} \leq \varepsilon. \tag{34}$$

*Furthermore, the neural operator $\mathcal{G} : C^\infty([0,T]) \to W_2^k(\Omega)$ is discretization-invariant such that*

$$\sup_{a \in \mathcal{A}} \| \mathcal{G}(a) \| \lesssim \sup_{a \in \mathcal{A}} \left\| \mathcal{G}^\dagger(a) \right\|, \tag{35}$$

*where the notation $\lesssim$ means the inequality $\leq$ holds up to constant factors.*

*Proof.* Due to [24, Theorem 8] and Remark A.2, the neural operator $\mathcal{G}$ is discretization-invariant since $W_2^k(\Omega)$ is continuously embedded in $C(\bar{\Omega})$. We apply [24, Theorem 11] to show (34) and (35). Note that Assumption 9 holds since we set $\mathcal{A} = C([0,T])$ and so does Assumption 10 since we set $\mathcal{U} = W_2^k(\Omega)$. Therefore, for a given $\varepsilon > 0$ and a compact set $\bar{A} \subset \mathcal{A}$, by [24, Theorem 11], there exists a neural operator $\mathcal{G}$ such that

$$\sup_{a \in \bar{A}} \left\| \mathcal{G}^\dagger(a) - \mathcal{G}(a) \right\|_{W_2^k(\Omega)} \leq \epsilon. \tag{36}$$

Thus, (34) is proved. Also, (35) holds automatically because $\mathcal{U} = W_2^k(\Omega)$ is a Hilbert space with the inner product (30). The theorem is proved. □

**Remark A.3.** Theorem A.1 implies that there exists a set of $L$-layer neural operators which can approximate the operator $\mathcal{G}^\dagger$ which acts on a set of smooth coeffcients $a = (\mathbf{B}_t, \mathbf{G}_t)_{t \in [0,T]}$. Since $(\mathbf{B}_t, \mathbf{G}_t)_{t \in [0,T]}$ determines marginal densities $(\mu_t : t \in [0,T])$ of stochastic equations (27), the inequality (34) in Theorem A.1 enhances the approximation to *weak* derivatives of the target score function $\mathbf{S}_*$ in the Sobolev space. To obtain the desired approximation result in Hölder space, we utilize the Sobolev embedding theorem (see Remark A.2), upon Theorem A.1.

Now we prove the main result. Let us restate Theorem 3.1 formally.

**Theorem A.2.** *Let $k \in \mathbb{N}$ such that $k > 2 + \frac{D}{2}$. For any compact subset $K \subset \Omega \subset \mathcal{X}$ and $\epsilon > 0$, there exists a neural operator $\widehat{\mathbf{S}}^\theta$ such that*

$$\sup_{\mathbf{x} \in K} \sum_{|\alpha| \leq m} \left| \partial_\mathbf{x}^\alpha \mathbf{S}(\mathbf{x}) - \partial_\mathbf{x}^\alpha \widehat{\mathbf{S}}^\theta(\mathbf{x}) \right| \leq \varepsilon, \quad m = \lfloor k - \frac{D}{2} \rfloor. \tag{37}$$

*Proof.* Let $\varepsilon > 0$ be given and fix a compact set $K \subset \Omega$. Due to Theorem A.1, there exists a neural operator $\mathcal{G}$ which maps a smooth coeffcients $a = (\mathbf{B}_t, \mathbf{G}_t)_{t \in [0,T]} \in \mathcal{A}$ to $W_2^k(\Omega)$-valued function, $\mathcal{G}(a) = \widehat{\mathbf{S}}^\theta : \Omega \to \mathbb{R}^D$, with the approximation power (34). Therefore, by (34), it holds that

$$\left\| \mathbf{S}_* - \widehat{\mathbf{S}}^\theta \right\|_{W_2^k(\Omega)} \leq \varepsilon/C, \tag{38}$$

where the constant $C$ is determined by (31). Having $K \subset \Omega$ in mind, by (31) and (32),

$$\sup_{\mathbf{x} \in K} \sum_{|\alpha| \leq m} \left| \partial_\mathbf{x}^\alpha \mathbf{S}(\mathbf{x}) - \partial_\mathbf{x}^\alpha \widehat{\mathbf{S}}^\theta(\mathbf{x}) \right| \leq \left\| \mathbf{S}_* - \widehat{\mathbf{S}}^\theta \right\|_{C^{k-\frac{D}{2}}(\Omega)} \leq C \left\| \mathbf{S}_* - \widehat{\mathbf{S}}^\theta \right\|_{W_2^k(\Omega)} \leq \varepsilon, \tag{39}$$

hence we obtain (37). The theorem is proved. □

### A.3 Architectural Details

#### A.3.1 Architecture of SciNO

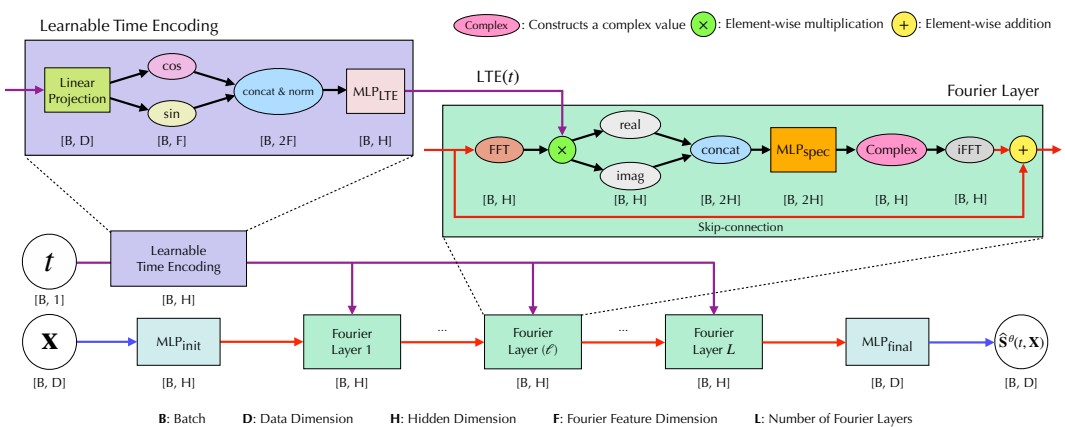

Figure A.1: Architecture of Score-informed Neural Operator

A $D$-dimensional input data $\mathbf{X} \in \mathbb{R}^D$ is passed through the initial $\text{MLP}_{\text{init}} : \mathbb{R}^D \to \mathbb{R}^H$ layer:

$$\mathbf{X}^{(1)} = \text{MLP}_{\text{init}}(\mathbf{X}). \tag{40}$$

To explain the intermediate layers in SciNO, let $\ell \in \{1, \dots, L\}$ denote the index of Fourier layers. Then we transform $\mathbf{X}^{(\ell)}$ into the spectral domain to get the following $\mathbb{C}^H$-dimensional tensor:

$$\xi^{(\ell)} = \text{FFT}(\mathbf{X}^{(\ell)}) \odot \text{LTE}(t), \quad t \in [0, 1] \tag{41}$$

where FFT means the Fast Fourier Transform, $\odot$ denotes the element-wise multiplication, and $\text{LTE} : [0, 1] \to \mathbb{R}^H$ is the *Learnable Time Encoding* module. The real $\Re(\xi^{(\ell)})$ and imaginary $\Im(\xi^{(\ell)})$ parts are then split to construct an $\mathbb{R}^{2H}$-valued tensor $\chi^{(\ell)} = \left[\Re(\xi^{(\ell)}), \Im(\xi^{(\ell)})\right]$. Then we plug the tensor into an $\text{MLP}_{\text{spec}}^{(\ell)} : \mathbb{R}^{2H} \to \mathbb{R}^{2H}$ layer in the spectral domain:

$$\mathbf{Z}^{(\ell)} = \text{MLP}_{\text{spec}}^{(\ell)}(\chi^{(\ell)}). \tag{42}$$

Then we combine $\mathbf{Z}^{(\ell)} = [\mathbf{Z}_{\text{real}}^{(\ell)}, \mathbf{Z}_{\text{imag}}^{(\ell)}]$ to get a $\mathbb{C}^H$-valued tensor $\zeta^{(\ell)} = \mathbf{Z}_{\text{real}}^{(\ell)} + i\mathbf{Z}_{\text{imag}}^{(\ell)}$. Next, we transform the signal $\zeta^{(\ell)}$ from the spectral domain to the spatial domain with skip connection, which helps alleviate gradient vanishing during training:

$$\mathbf{X}^{(\ell+1)} = \mathbf{X}^{(\ell)} + \text{iFFT}(\zeta^{(\ell)}), \tag{43}$$

where $\text{iFFT}(\cdot)$ denotes the inverse Fourier transformation. By repeating the above procedure for $\ell \in \{1, \dots, L\}$, we get the output of the Fourier layer $\mathbf{X}^{(L+1)}$. Then $\mathbf{X}^{(L+1)}$ is passed through the final $\text{MLP}_{\text{final}} : \mathbb{R}^H \to \mathbb{R}^D$ layer for final processing:

$$\widehat{\mathbf{S}}^\theta(t, \mathbf{X}) = \text{MLP}_{\text{final}}(\mathbf{X}^{(L+1)}). \tag{44}$$

#### A.3.2 MLP Modules in SciNO

The MLP modules used in SciNO are organized as follows.

**Initial MLP Layer.** The input data $\mathbf{X} \in \mathbb{R}^D$ is first projected into $\mathbb{R}^H$ through $\text{MLP}_{\text{init}}$:

$$\text{MLP}_{\text{init}}(\mathbf{X}) = \text{Dropout}_{0.2}\left(\text{LayerNorm}\left(\text{LeakyReLU}\left(W_{\text{init}}\mathbf{X}\right)\right)\right) \in \mathbb{R}^H, \tag{45}$$

where $W_{\text{init}} \in \mathbb{R}^{H \times D}$.

**Spectral MLP Layer.** $\text{MLP}_{\text{spec}}$ is designed to effectively capture differential information in the frequency domain. The input $\chi^{(\ell)} \in \mathbb{R}^{2H}$ is processed as follows:

$$\text{MLP}^{(\ell)}_{\text{spec}}(\chi^{(\ell)}) = \text{BatchNorm}\left(\text{LeakyReLU}\left(W^{(\ell)}_{\text{spec}}\chi^{(\ell)} + b^{(\ell)}_{\text{spec}}\right)\right) \in \mathbb{R}^{2H}. \tag{46}$$

The layer is parameterized by $W^{(\ell)}_{\text{spec}}$ and $b^{(\ell)}_{\text{spec}}$, with shapes $\mathbb{R}^{2H \times 2H}$ and $\mathbb{R}^{2H}$.

**Final MLP Layer.** The output $\mathbf{X}^{(L+1)} \in \mathbb{R}^H$ from the final Fourier layer is mapped to $\mathbb{R}^D$ via $\text{MLP}_{\text{final}}$:

$$\mathbf{h}_1 = \text{LeakyReLU}(W^{(1)}_{\text{final}}\mathbf{X}^{(L+1)} + b^{(1)}_{\text{final}}), \tag{47}$$

$$\mathbf{h}_2 = \text{LeakyReLU}(W^{(2)}_{\text{final}}\mathbf{h}_1 + b^{(2)}_{\text{final}}), \tag{48}$$

$$\text{MLP}_{\text{final}}(\mathbf{X}^{(L+1)}) = W^{(3)}_{\text{final}}\mathbf{h}_2 + b^{(3)}_{\text{final}}. \tag{49}$$

Here, $W^{(1)}_{\text{final}} \in \mathbb{R}^{H \times H}$, $b^{(1)}_{\text{final}} \in \mathbb{R}^H$, $W^{(2)}_{\text{final}} \in \mathbb{R}^{S \times H}$, $b^{(2)}_{\text{final}} \in \mathbb{R}^S$, $W^{(3)}_{\text{final}} \in \mathbb{R}^{D \times S}$, and $b^{(3)}_{\text{final}} \in \mathbb{R}^D$ denote the weights and biases of the $\text{MLP}_{\text{final}}$. The dimension $S$ denotes an intermediate hidden dimension between $D$ and $H$.

**Time Encoding MLP Layer.** Given the Fourier feature $\Phi(t) \in \mathbb{R}^{2F}$, the time encoding $\text{LTE}(t)$ is computed via $\text{MLP}_{\text{LTE}}$:

$$\text{MLP}_{\text{LTE}}(\Phi(t)) = W^{(2)}_{\text{LTE}}\text{GeLU}\left(W^{(1)}_{\text{LTE}}\Phi(t) + b^{(1)}_{\text{LTE}}\right) + b^{(2)}_{\text{LTE}} \in \mathbb{R}^H, \tag{50}$$

where $W^{(1)}_{\text{LTE}} \in \mathbb{R}^{M \times 2F}$, $b^{(1)}_{\text{LTE}} \in \mathbb{R}^M$, $W^{(2)}_{\text{LTE}} \in \mathbb{R}^{H \times M}$, and $b^{(2)}_{\text{LTE}} \in \mathbb{R}^H$. Here, $M$ denotes the hidden dimension of the intermediate layer in the Time Encoding MLP.

### A.3.3 Architecture of Time-Conditioned FNO

We describe the architecture of the time-conditioned Fourier Neural Operator (FNO) [26] for tabular data, following the same setting as SciNO for consistent comparison.

The time-conditioned FNO takes an input $\mathbf{X} \in \mathbb{R}^D$ and first maps it to a $H$-dimensional hidden representation via a lifting layer:

$$\mathbf{X}^{(1)} = \text{Linear}(\mathbf{X}). \tag{51}$$

A time encoding $\text{TE} \in \mathbb{R}^H$ is obtained from the scalar diffusion time $t$ using sinusoidal positional encoding [43] defined as a mapping $\text{PE} : [0, 1] \to \mathbb{R}^H$ and an $\text{MLP}_{\text{PE}} : \mathbb{R}^H \to \mathbb{R}^H$ layer:

$$\text{TE} = \text{MLP}_{\text{PE}}(\text{PE}(t)), \quad t \in [0, 1]. \tag{52}$$

This $H$-dimensional encoding is added to the lifted input:

$$\mathbf{X}^{(1)} \leftarrow \mathbf{X}^{(1)} + \text{TE}. \tag{53}$$

For each Fourier layer $\ell \in \{1, \dots, L\}$, we transform the hidden representation $\mathbf{X}^{(\ell)}$ into the spectral domain via FFT. The time encoding, processed by MLP, is then injected into the Fourier coefficients through element-wise addition to get the $\mathbb{C}^H$-dimensional tensor:

$$\xi^{(\ell)} = \text{FFT}(\mathbf{X}^{(\ell)}) + \text{MLP}^{(\ell)}_{\text{TE}}(\text{TE}). \tag{54}$$

Then we apply spectral weights $W^{(\ell)}_{\text{spec}} \in \mathbb{C}^{H \times H}$:

$$\widetilde{\xi}^{(\ell)} = W^{(\ell)}_{\text{spec}}\xi^{(\ell)}. \tag{55}$$

The transformed $\mathbb{C}^H$-dimensional signal is brought back to the $\mathbb{R}^H$-dimensional spatial domain via inverse FFT:

$$\mathbf{Z}^{(\ell)} = \text{iFFT}(\widetilde{\xi}^{(\ell)}). \tag{56}$$

A residual connection is applied:

$$\tilde{\mathbf{X}}^{(\ell)} = \mathbf{X}^{(\ell)} + \mathbf{Z}^{(\ell)}. \tag{57}$$

Then, an $\text{MLP}_{\text{spatial}}^{(\ell)} : \mathbb{R}^H \to \mathbb{R}^H$ layer is applied with a residual connection followed by a GeLU activation:

$$\mathbf{X}^{(\ell+1)} = \text{GeLU}(\tilde{\mathbf{X}}^{(\ell)} + \text{MLP}_{\text{spatial}}^{(\ell)}(\tilde{\mathbf{X}}^{(\ell)})). \tag{58}$$

After processing through $L$ layers, the final hidden representation $\mathbf{X}^{(L+1)}$ is projected back to the original $D$-dimension via an $\text{MLP}_{\text{final}} : \mathbb{R}^H \to \mathbb{R}^D$ layer:

$$\widehat{\mathbf{S}}^\theta(t, \mathbf{X}) = \text{MLP}_{\text{final}}(\mathbf{X}^{(L+1)}). \tag{59}$$

## A.4 Experimental Setting

### A.4.1 Datasets

**Physics** We generate a Physics commonsense-based synthetic dataset using the physics-based causal DAG introduced in [20], which models causal relations among 7 variables related to water evaporation. Each edge weight is sampled from the following uniform distributions:

$$e \sim \mathcal{U}(-1, -0.1) \cup \mathcal{U}(0.1, 1). \tag{60}$$

Using the sampled adjacency matrix $A$, we generate 5,000 samples via the following nonlinear Structural Equation Model (SEM):

$$x = 2\sin(A^\top(x + 0.5 \cdot \mathbf{1})) + A^\top(x + 0.5 \cdot \mathbf{1}) + z, \quad z \sim \mathcal{N}(0, 1). \tag{61}$$

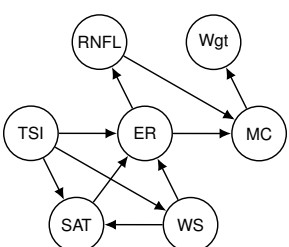

Figure A.2: Physics commonsense-based synthetic graph with 7 nodes: Rainfall (RNFL), Total Solar Irradiance (TSI), Surface Air Temperature (SAT), Wind Speed (WS), Evaporation Rate (ER), Moisture Content (MC), and Object Weight (Wgt).

**Sachs** The Sachs dataset [34] contains 7,466 single-cell measurements of 11 proteins and phospho-proteins involved in human immune signaling, collected under various stimulation conditions using flow cytometry. To ensure a DAG structure for evaluating causal order, we exclude terminal nodes (`praf`, `plcg`, and `PIP2`) involved in cycles and use the remaining 8-node graph for experiments.

**BNLearn** We use four real-world DAGs from the BNLearn repository— MAGIC-NIAB [38] (44 nodes/ 66 edges), ECOLI70 [36] (46 nodes/ 70 edges), MAGIC-IRRI [37] (64 nodes/ 102 edges), and ARTH150 [28] (107 nodes/ 150 edges)—and generate data based on their causal structures. For the linear case, samples are generated using the graph structures and linear functional relationships provided by BNLearn. For the nonlinear case, we generate data using nonlinear SEMs defined over the same graphs, where each functional relationship is parameterized by Multilayer Perceptron (MLP). We generate 10,000 samples per graph in both linear and nonlinear settings for evaluation.

### A.4.2 Metrics

**Order Divergence.** Given a causal order $\pi$ and a true adjacency matrix $\mathcal{G}$, the Order Divergence (OD, [33]) is defined as:

$$D_{\text{top}}(\pi, \mathcal{G}) = \sum_{i=1}^{D} \sum_{j:\pi_i > \pi_j} \mathcal{G}_{ij}. \tag{62}$$

Order divergence counts the number of edges in $\mathcal{G}$ that are inconsistent with the order $\pi$. A smaller value indicates a closer alignment between $\pi$ and $\mathcal{G}$.

**SHD.** Structural Hamming Distance (SHD, [41]) measures the structural discrepancy between a predicted graph $\widehat{\mathcal{G}}$ and the ground-truth graph $\mathcal{G}$. It is defined as the total number of edge insertions, deletions, and reversals required to convert $\widehat{\mathcal{G}}$ into $\mathcal{G}$:

$$\text{SHD}(\mathcal{G}, \widehat{\mathcal{G}}) = \# \left\{ (i,j) \in \mathcal{V}^2 \,\middle|\, \mathcal{G} \text{ and } \widehat{\mathcal{G}} \text{ do not have the same type of edge between } i \text{ and } j \right\}. \tag{63}$$

A lower SHD indicates that the predicted graph $\widehat{\mathcal{G}}$ is closer to the true causal graph $\mathcal{G}$ in terms of structural similarity.

**SID.** Structural Intervention Distance (SID, [29]) quantifies how similarly a predicted graph $\widehat{\mathcal{G}}$ and the ground-truth graph $\mathcal{G}$ infer causal relationships under intervention:

$$\text{SID}(\mathcal{G}, \widehat{\mathcal{G}}) = \# \left\{ (i,j),\, i \neq j \,\middle|\, \begin{array}{l} \text{Pa}^{\widehat{\mathcal{G}}}(X_i) \text{ not a valid adjustment set for} \\ \text{the intervention } X_j | \text{do}(X_i) \text{ in graph } \mathcal{G} \end{array} \right\}. \tag{64}$$

A lower SID indicates that the predicted graph $\widehat{\mathcal{G}}$ is more causally consistent with the true causal graph $\mathcal{G}$ under interventions, making SID a suitable metric for evaluating the reliability of causal inference.

### A.4.3 Implementation Details

For synthetic datasets, we set the number of Fourier layers to $L = 10$, and for real-world datasets, we use $L = 1$. The Fourier feature dimension in the LTE module (50) is fixed at $F = 32$. We set the hidden dimensions of the model as $H = \max(1024, 5D)$, and $S = \max(128, 3D)$ in (45), (46), and (47), where $D$ denotes the number of nodes $|\mathcal{V}|$. All experiments are conducted on a single NVIDIA A6000 GPU. Data, checkpoints, and all training configurations are publicly available[3].

---

[3] https://github.com/LGAI-Research/SciNO

## A.5  Additional Experiments

### A.5.1  Ablation Studies

We conduct a series of ablation studies to evaluate the impact of key architectural choices.

**PE vs. LTE**    In Table A.1, we provide additional evidence comparing standard PE and LTE on datasets beyond the ER graphs. Consistent with Table 1, LTE yields lower OD, especially as the number of variables increases, confirming its effectiveness in diverse graph settings.

Table A.1: Comparison of OD in real and semi-synthetic datasets between PE and LTE in SciNO. Each score is recorded over 10 independent runs.

| Method\Dataset | Physics(d7) | Sachs(d8) | MAGIC-NIAB(d44) | ECOLI70(d46) | MAGIC-IRRI(d64) | ARTH150(d107) |
|---|---|---|---|---|---|---|
| DiffAN w/ SciNO (PE) | $2.7 \pm 0.46$ | $\mathbf{4.4} \pm 1.50$ | $\mathbf{4.0} \pm 1.55$ | $30.6 \pm 2.42$ | $16.6 \pm 1.12$ | $33.9 \pm 3.94$ |
| DiffAN w/ SciNO (LTE) | $\mathbf{1.9} \pm 0.54$ | $5.6 \pm 2.65$ | $4.1 \pm 0.70$ | $\mathbf{12.8} \pm 3.19$ | $\mathbf{10.6} \pm 1.69$ | $\mathbf{21.4} \pm 2.76$ |

**Additive vs. Multiplicative LTE**    To validate our design choice of using a multiplicative form of LTE, we compare it against the more common additive variant in Table A.2. Across ER graph settings, the multiplicative variant consistently achieves lower OD, with the performance gap becoming more pronounced as the number of variables increases.

Table A.2: Comparison of OD in ER datasets between additive and multiplicative LTE in SciNO. Each score is recorded over 10 random graphs.

| Method\Dataset | ER(d10) | ER(d30) | ER(d50) | ER(d100) |
|---|---|---|---|---|
| DiffAN w/ SciNO (Additive LTE) | $2.8 \pm 1.74$ | $17.67 \pm 5.64$ | $38.86 \pm 11.42$ | $99.2 \pm 14.27$ |
| DiffAN w/ SciNO (Multiplicative LTE) | $\mathbf{2.7} \pm 1.19$ | $\mathbf{16.9} \pm 6.12$ | $\mathbf{32.8} \pm 6.55$ | $\mathbf{86.6} \pm 12.99$ |

**DiffAN with LTE**    We applied LTE to DiffAN to assess whether LTE alone explains the performance gain. As shown in Table A.3, simply injecting LTE into DiffAN does not yield performance improvement. This suggests that LTE alone is insufficient, and that the performance gain results from combining LTE with our architectural design.

Table A.3: Comparison of OD on synthetic and real datasets between DiffAN with LTE and SciNO. Each score is recorded over 10 random graphs for synthetic datasets, and over 10 independent runs for real and semi-synthetic datasets.

| Dataset\Method | DiffAN w/ MLP (LTE) | DiffAN w/ SciNO (LTE) |
|---|---|---|
| ER(d2) | $0.2 \pm 0.4$ | $\mathbf{0.0} \pm 0.0$ |
| ER(d3) | $0.1 \pm 0.3$ | $\mathbf{0.1} \pm 0.3$ |
| ER(d5) | $\mathbf{0.8} \pm 0.75$ | $0.9 \pm 1.14$ |
| ER(d10) | $3.9 \pm 1.51$ | $\mathbf{2.7} \pm 1.19$ |
| ER(d30) | $37.1 \pm 7.33$ | $\mathbf{16.9} \pm 6.12$ |
| ER(d50) | $62.0 \pm 11.96$ | $\mathbf{32.8} \pm 6.55$ |
| ER(d100) | $120.3 \pm 17.12$ | $\mathbf{86.6} \pm 12.99$ |
| Physics(d7) | $5.0 \pm 1.1$ | $\mathbf{2.0} \pm 0.0$ |
| Sachs(d8) | $7.2 \pm 2.71$ | $\mathbf{5.8} \pm 2.93$ |
| MAGIC-NIAB(d44) | $10.6 \pm 12.72$ | $\mathbf{3.8} \pm 0.75$ |
| ECOLI70(d46) | $20.4 \pm 1.85$ | $\mathbf{14.4} \pm 3.61$ |
| MAGIC-IRRI(d64) | $25.0 \pm 23.06$ | $\mathbf{10.0} \pm 0.89$ |
| ARTH150(d107) | $82.0 \pm 23.28$ | $\mathbf{20.8} \pm 2.32$ |

### A.5.2 Comparison with Causal Discovery Baselines

**ANM-based Baselines**  We compare SciNO against ANM-based baselines: CAM [4] and SCORE [33]. As shown in Table A.4, SciNO achieves consistently lower SHD than CAM on the high-dimensional ER datasets and superior performance on the semi-synthetic benchmarks, except for ARTH150. Compared to SCORE, SciNO yields similar performance on low-dimensional graphs and demonstrates a significant advantage on high-dimensional settings, achieving higher accuracy than second-order Stein gradient methods. Overall, SciNO demonstrates reliable performance across both synthetic and real datasets, particularly in high-dimensional regimes.

Table A.4: Comparison of causal discovery metrics(OD/SHD/SID) between CAM [4], SCORE [33], and SciNO in (a) synthetic datasets and (b) real and semi-synthetic datasets: Physics, Sachs, and BNLearn with *nonlinear* ANM. Each score is recorded over 10 random graphs for synthetic datasets, and over 10 independent runs for real and semi-synthetic datasets.

| Dataset\Metric | CAM [4] | | SCORE [33] | | | DiffAN w/ SciNO (Ours) | | |
|---|---|---|---|---|---|---|---|---|
| | SHD ($\downarrow$) | SID ($\downarrow$) | OD ($\downarrow$) | SHD ($\downarrow$) | SID ($\downarrow$) | OD ($\downarrow$) | SHD ($\downarrow$) | SID ($\downarrow$) |
| ER(d2) | **0.0** ± 0.0 | **0.0** ± 0.0 | 0.1 ± 0.3 | 0.2 ± 0.6 | 0.2 ± 0.5 | **0.0** ± 0.0 | **0.0** ± 0.0 | **0.0** ± 0.0 |
| ER(d3) | **0.0** ± 0.0 | **0.0** ± 0.0 | **0.0** ± 0.0 | **0.0** ± 0.0 | **0.0** ± 0.0 | 0.1 ± 0.3 | 0.2 ± 0.6 | 0.4 ± 1.2 |
| ER(d5) | **0.0** ± 0.0 | **0.0** ± 0.0 | **0.0** ± 0.0 | 0.1 ± 0.3 | 0.1 ± 0.3 | 0.9 ± 1.14 | 1.7 ± 1.95 | 4.8 ± 4.94 |
| ER(d10) | 16.6 ± 4.45 | **30.9** ± 11.96 | 4.2 ± 2.6 | 21.5 ± 5.24 | 41.2 ± 13.24 | **2.7** ± 1.19 | 20.5 ± 3.5 | 41.6 ± 9.31 |
| ER(d30) | 129.0 ± 14.99 | 450.2 ± 99.2 | 24.8 ± 11.97 | 92.8 ± 16.27 | 510.8 ± 90.77 | **16.9** ± 6.12 | **88.8** ± 16.5 | 492.0 ± 91.64 |
| ER(d50) | 238.5 ± 17.29 | **1503.6** ± 201.35 | 49.3 ± 14.37 | 180.8 ± 15.99 | 1527.1 ± 179.61 | **32.8** ± 6.55 | **180.0** ± 15.45 | 1622.0 ± 110.92 |
| ER(d100) | 537.2 ± 45.37 | **6289.9** ± 533.87 | 120.0 ± 18.49 | 455.3 ± 26.34 | 7175.0 ± 435.0 | **86.6** ± 12.99 | **445.9** ± 32.84 | 7259.2 ± 653.27 |
| Physics(d7) | 12 | 11 | **1** | **2** | 12 | 1.9 ± 0.54 | 5.8 ± 1.60 | **8.2** ± 4.62 |
| Sachs(d8) | 37 | 41 | 6 | **16** | 30 | **5.6** ± 2.65 | 23.1 ± 3.86 | **23.3** ± 9.10 |
| MAGIC-NIAB(d44) | 167 | 384 | 8 | **61** | 189 | **4.1** ± 0.70 | 72.0 ± 4.52 | **125.5** ± 28.08 |
| ECOLI70(d46) | 184 | 828 | **7** | **80** | 771 | 12.8 ± 3.19 | 90.3 ± 7.13 | **500.9** ± 58.16 |
| MAGIC-IRRI(d64) | 261 | 791 | 17 | 175 | 529 | **10.6** ± 1.69 | 144.5 ± 4.43 | **254.4** ± 39.66 |
| ARTH150(d107) | **437** | 2133 | 38 | 460 | 2709 | **21.4** ± 2.76 | 515.6 ± 15.26 | **1186.7** ± 104.19 |

**Non-ANM Baselines**  We further compare SciNO with classical Causal Discovery algorithms that do not rely on ANM assumptions, the constraint-based PC [40] and the score-based GES [7]. To ensure a fair comparison, we convert the predicted graphs of all methods into CPDAGs and report results using Structural Hamming Distance for CPDAGs (SHD-C). As shown in Table A.5, SciNO consistently outperforms both PC and GES across ER datasets of various sizes. This highlights the advantages of our approach against these classical methods: it achieves superior accuracy while also being more scalable, overcoming the high computational complexity. Beyond these statistical algorithms, we also consider gradient-based approaches such as DAG-GNN [46] and GraN-DAG [18], which parameterize causal mechanisms via deep generative models and gradient-based optimization. The results are summarized in Table A.6. While DAG-GNN and GraN-DAG perform competitively on small graphs, their accuracy deteriorates as dimensionality increases. In contrast, SciNO maintains consistently low error even for $D = 50$, clearly demonstrating its scalability.

Table A.5: Comparison of SHD-C between PC [40], GES [7] and SciNO in ER datasets. Each score is recorded over 10 random graphs.

| Dataset\Method | PC [40] | GES [7] | DiffAN w/ SciNO (Ours) |
|---|---|---|---|
| ER(d2) | 0.1 ± 0.3 | 0.2 ± 0.4 | **0.0** ± 0.0 |
| ER(d3) | 0.6 ± 0.92 | 1.3 ± 1.01 | **0.0** ± 0.0 |
| ER(d5) | 9.03 ± 0.64 | 8.4 ± 1.43 | **1.3** ± 2.41 |
| ER(d10) | 34.6 ± 2.93 | 35.2 ± 2.96 | **23.8** ± 5.49 |
| ER(d30) | 108.5 ± 15.69 | 105.7 ± 15.3 | **88.3** ± 18.11 |
| ER(d50) | 195.9 ± 12.08 | 180.2 ± 12.45 | **174.3** ± 14.48 |

Table A.6: Comparison of causal discovery metrics (SHD/SID) between DAG-GNN [46], GraNDAG [18], and SciNO across ER synthetic datasets with varying node sizes. Results are averaged over 10 random graphs.

| Dataset\Metric | DAG-GNN [46] | | GraN-DAG [18] | | DiffAN w/ SciNO (Ours) | |
|---|---|---|---|---|---|---|
| | SHD ($\downarrow$) | SID ($\downarrow$) | SHD ($\downarrow$) | SID ($\downarrow$) | SHD ($\downarrow$) | SID ($\downarrow$) |
| ER(d2) | $0.9 \pm 0.3$ | $0.9 \pm 0.3$ | $\mathbf{0.0} \pm 0.0$ | $\mathbf{0.0} \pm 0.0$ | $\mathbf{0.0} \pm 0.0$ | $\mathbf{0.0} \pm 0.0$ |
| ER(d3) | $2.1 \pm 0.94$ | $2.9 \pm 1.37$ | $0.3 \pm 0.9$ | $\mathbf{0.4} \pm 1.2$ | $\mathbf{0.2} \pm 0.6$ | $\mathbf{0.4} \pm 1.2$ |
| ER(d5) | $9.3 \pm 0.78$ | $14.0 \pm 2.1$ | $\mathbf{0.0} \pm 0.0$ | $\mathbf{0.0} \pm 0.0$ | $1.7 \pm 1.95$ | $4.8 \pm 4.94$ |
| ER(d10) | $34.4 \pm 21.62$ | $71.3 \pm 7.26$ | $20.7 \pm 8.9$ | $30.4 \pm 19.86$ | $\mathbf{20.5} \pm 3.5$ | $41.6 \pm 9.31$ |
| ER(d30) | $108.0 \pm 16.37$ | $651.9 \pm 60.35$ | $140.9 \pm 14.54$ | $614.1 \pm 31.44$ | $\mathbf{88.8} \pm 16.5$ | $\mathbf{492.0} \pm 91.64$ |
| ER(d50) | $192.8 \pm 10.81$ | $1900.9 \pm 206.39$ | $232.5 \pm 16.76$ | $1802.1 \pm 40.28$ | $\mathbf{180.0} \pm 15.45$ | $\mathbf{1622.0} \pm 110.92$ |
| ER(d100) | $396.0 \pm 22.12$ | $7586.4 \pm 585.98$ | $\mathbf{232.5} \pm 31.81$ | $7623.4 \pm 455.73$ | $445.9 \pm 32.84$ | $\mathbf{7259.2} \pm 653.27$ |

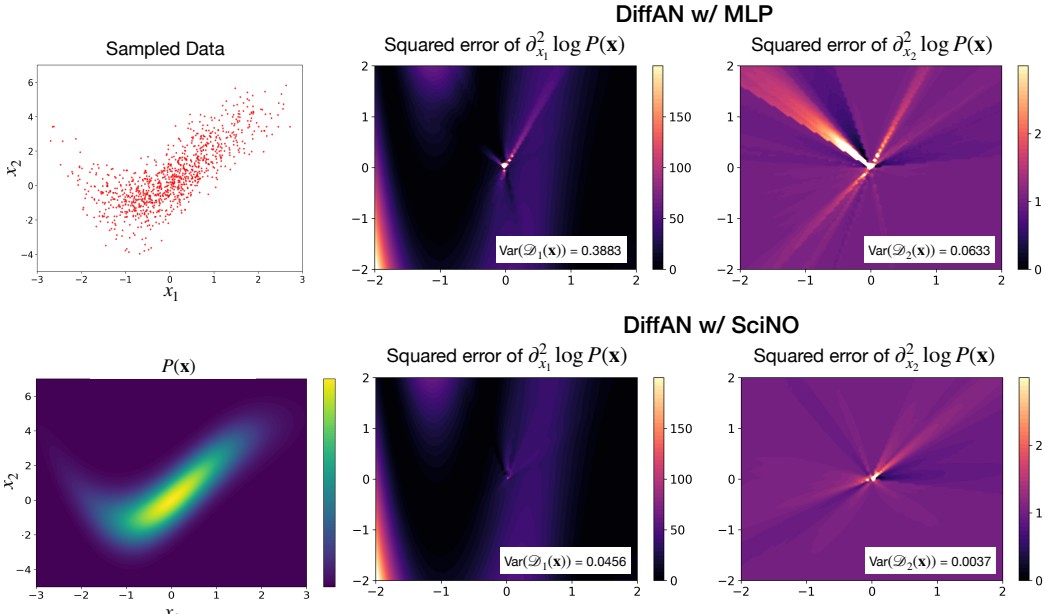

Figure A.3: Comparison of Hessian diagonal approximation between DiffAN with MLP (top row) and DiffAN with SciNO (bottom row). The left column shows sampled data and its true density. The center and right columns illustrate the squared approximation error of the second derivatives of $\log P(\mathbf{x})$ with respect to $x_1$ and $x_2$, computed over a meshgrid.

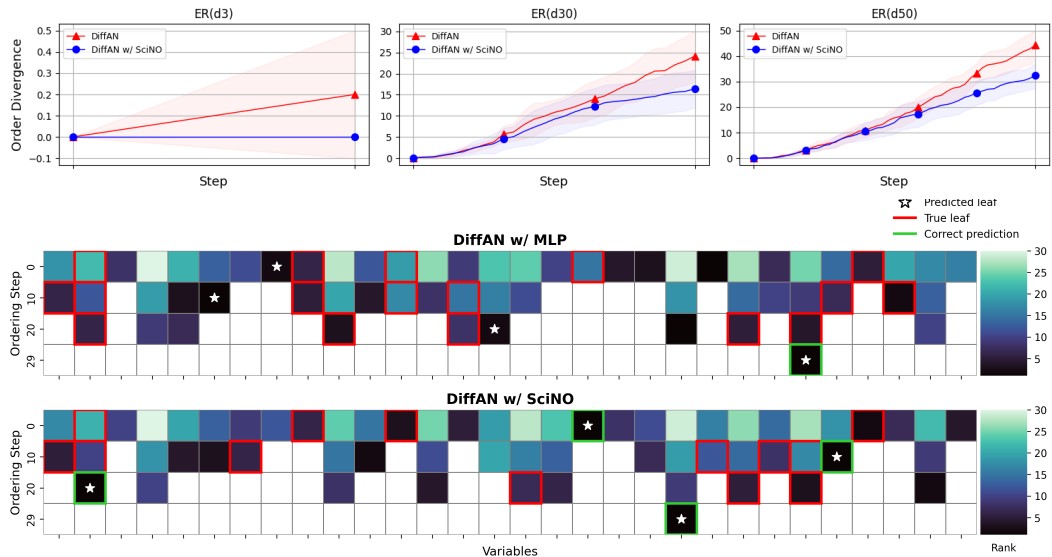

Figure A.4: **(Top)** Order divergence in ER datasets per each step. The solid lines represent the average order divergence, and the shaded regions indicate the 95% confidence intervals across 10 random graphs with $D \in \{3, 30, 50\}$. **(Bottom)** Heatmaps of variance of the Hessian diagonal estimated by DiffAN with MLP and SciNO, respectively. Darker colors indicate lower estimated variance. Red boxes □ denote ground-truth causal leaves, and white stars ☆ indicate the variables selected by models. Green boxes □ denote cases where the model predicts leaf node correctly.

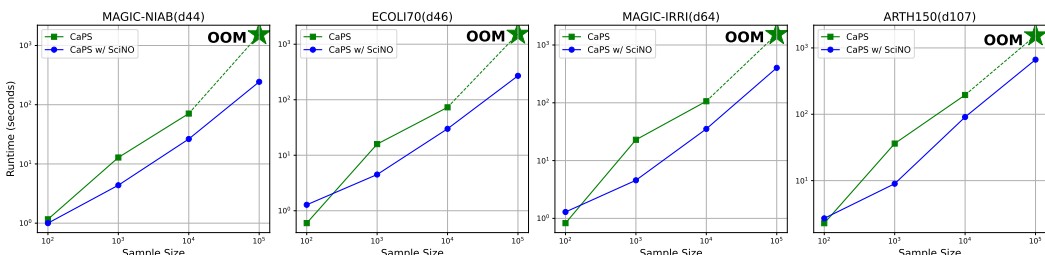

Figure A.5: Runtime in seconds of CaPS [45] and CaPS with SciNO across four real datasets (MAGIC-NIAB, ECOLI70, MAGIC-IRRI, and ARTH150) for different sample sizes. Since SciNO involves iterative training, we report its runtime per epoch. CaPS fails with OOM at 100,000 samples.

## A.6 Controlling LLM for Causal Ordering

### A.6.1 Experimental Details in Section 4.3

**Implementation Details in LLM Control**  The LLM prompt is designed to select a leaf node, a variable among the unordered variables that does not causally influence any other variable. Each prompt consists of an `Unordered Variables` list, a `Data Description` summarizing the overall data context, and `Variable Descriptions` providing natural language descriptions of each variable (as illustrated in Figure A.7). Simple input-output examples are provided within the prompt to clarify the task objective and promote consistent output behavior. To avoid formatting inconsistencies such as stray whitespace or mismatched capitalization, each variable name is prefixed with `node_` and presented in the format `node_variable_name`.

For each variable name, the LLM generates token-level conditional probabilities for the sequence of tokens $t_1, t_2, \ldots, t_n$ that constitute $v$, where $v = (t_1, \ldots, t_n)$. The probability of generating the full variable name $v$ is computed using the chain rule:

$$P_{\text{LLM}}(v \mid \texttt{context}) = \prod_{i=1}^{n} P_{\text{LLM}}(t_i \mid t_1, \ldots, t_{i-1}, \texttt{context}). \tag{65}$$

Here, `context` refers to contextual information, including variable descriptions or domain knowledge. In practice, we construct the full input sequence by appending each variable name to the prompt in the form of `node_variable_name`, and then feed the entire sequence to the model in a single forward pass. To compute the probability of each variable name, we consider only the tokens following the common prefix `node`, since tokenization often splits strings like `node_v` into `node` and `_v`. This allows us to isolate the informative portion of the name and compute joint probabilities efficiently, without repeatedly modifying the prompt.

**Length Normalization** Since variable names typically consist of multiple tokens with varying length, using raw probabilities introduces a bias against longer names. To address this, we apply length normalization by exponentiating the average of the log-likelihoods of the tokens composing the variable name [44]. To further mitigate over-penalization of longer names, we normalize the value by dividing it by length$^\alpha$ ($0 < \alpha \leq 1$). In this paper, we use $\alpha \in \{0.5, 1.0\}$.

$$P_{\alpha\text{-norm}}(v \mid \texttt{context}) = \exp\left(\frac{1}{n^\alpha} \sum_{i=1}^{n} \log P_{\text{LLM}}(t_i \mid t_1, \ldots, t_{i-1}, \texttt{context})\right). \quad (66)$$

**Leaf node selection through LLM control.** Llama-3 provides token-level probabilities directly, enabling us to compute the leaf node probabilities using (66). The leaf node is selected as $v^* = \arg\max_{v \in \mathcal{V} \setminus \pi_k} P_{\alpha\text{-norm}}(v)$, i.e., the variable with the highest length-normalized probability, where $\pi_k$ denotes the set of selected leaf nodes up to step $k$. For GPT-4o, we treat the variables mentioned in the final response as leaf nodes, since the model's probability distribution cannot be externally controlled during generation. We set the temperature to its default value of 1.

**Results** Figure A.6 presents the results under the setting where all variables are well-described, i.e., $\mathcal{V}_{\text{context}} = \mathcal{V}$. The top plot compares the LLM prior probabilities with the posterior probabilities—updated via control—of the ground-truth leaf nodes at each step of the causal reasoning process. A substantial increase in the posterior probability is consistently observed across high-dimensional datasets such as ECOLI70, MAGIC-NIAB, MAGIC-IRRI, and ARTH150. The effect is especially pronounced in the early stages of the causal ordering process, where the true causal structure must be inferred from a large pool of candidate nodes. Furthermore, the bottom plot illustrates several cases in which the posterior successfully identifies the true leaf node, even when SciNO alone fails to do so. These results indicate that the LLM prior can be complemented through integration with SciNO's data-informed evidence, while the semantic knowledge embedded in the LLM can also compensate for incomplete or uncertain statistical inference.

Table A.7 presents the complete benchmark results of the LLM control method using evidence from SCORE [33], DiffAN [35], and SciNO. SciNO shows clear superiority on high-dimensional datasets, achieving an average 64% OD reduction, compared to 46% from DiffAN. While SCORE-based control yields improvements on some datasets, it fails to surpass its data-only baseline on ECOLI70 and ARTH150, as shown in Table A.4. Overall, these findings indicate that the synergy between LLM priors and data-driven evidence is most robust with stable, diffusion-based methods like SciNO.

### A.6.2 Additional Experiments on LLM Control

In many domains such as biomedicine, healthcare, and finance, variable names are deliberately abstracted and metadata is often scarce and uncertain. To account for this practical challenge, we conduct an additional experiment in which all variable names are masked and the proportion of variables accompanied by descriptions is systematically varied.

**Experimental Setting** We consider an experiment to simulate practical conditions involving a mixture of $\mathcal{V}_{\text{context}}$ and $\mathcal{V}_{\neg\text{context}} \neq \emptyset$, where $\mathcal{V}_{\text{context}}$ refers to variables that contain contextual or domain-specific information, whereas $\mathcal{V}_{\neg\text{context}}$ includes variables without such information. Variable names are replaced with randomly generated four-letter alphabetic strings (e.g., 'zntr', 'lgvp', 'aqsm'), and descriptions are provided for 10%, 40%, or 70% of the variables. Each experiment is repeated three times, with randomization applied to both the selection of context-given variables and their input order. We use semi-synthetic datasets from BNLearn: MAGIC-NIAB (7/44 variables have descriptions), ECOLI70 (41/46), MAGIC-IRRI (10/64), and ARTH150 (105/107). Due to the limited availability of variable descriptions in MAGIC-NIAB and MAGIC-IRRI, we restrict experiments on these two datasets to the 10% description setting.

Table A.7: Comparison of OD in real and semi-synthetic datasets between GPT-4o, uncontrolled Llama, and controlled Llama via SCORE [33], DiffAN [35] and SciNO. Differences are expressed as percent change relative to the uncontrolled result. Each score is recorded over 10 independent runs.

| Method\Dataset | | Physics(d7) | Sachs(d8) | MAGIC-NIAB(d44) | ECOLI70(d46) | MAGIC-IRRI(d64) | ARTH150(d107) |
|---|---|---|---|---|---|---|---|
| GPT-4o | | $1.1 \pm 0.54$ | $4.4 \pm 1.11$ | $11.9 \pm 0.94$ | $41.1 \pm 2.30$ | $20.9 \pm 2.30$ | $77.1 \pm 7.46$ |
| SCORE [33] | Llama-3.1-8b(1-norm) | 3 | 3 | 9 | 32 | 18 | 80 |
| | w/ control(rank) | $0.0 \pm 0.00$ | $9.0 \pm 0.00$ | $\mathbf{1.0} \pm 0.00$ | $\mathbf{14.7} \pm 0.46$ | $11.6 \pm 0.49$ | $\mathbf{42.9} \pm 1.04$ |
| | w/ control(CI) | $0.0 \pm 0.00$ | $6.0 \pm 0.00$ | $1.8 \pm 0.40$ | $18.3 \pm 0.46$ | $\mathbf{11.1} \pm 0.30$ | $45.4 \pm 6.22$ |
| | Llama-3.1-8b(0.5-norm) | 1 | 3 | 14 | 32 | 17 | 80 |
| | w/ control(rank) | $0.0 \pm 0.00$ | $9.0 \pm 0.00$ | $\mathbf{2.0} \pm 0.63$ | $\mathbf{14.9} \pm 0.30$ | $9.9 \pm 0.30$ | $\mathbf{42.9} \pm 1.04$ |
| | w/ control(CI) | $0.0 \pm 0.00$ | $7.0 \pm 0.00$ | $\mathbf{2.0} \pm 0.00$ | $18.1 \pm 0.30$ | $\mathbf{8.3} \pm 1.62$ | $45.4 \pm 6.22$ |
| DiffAN [35] | Llama-3.1-8b(1-norm) | 3 | 3 | 9 | 32 | 18 | 80 |
| | w/ control(rank) | $3.9 \pm 0.30$ | $4.0 \pm 1.00$ | $\mathbf{1.9} \pm 0.70$ | $21.1 \pm 2.02$ | $\mathbf{7.4} \pm 0.66$ | $\mathbf{21.5} \pm 2.54$ |
| | w/ control(CI) | $3.6 \pm 0.49$ | $3.7 \pm 0.46$ | $5.3 \pm 1.10$ | $\mathbf{18.3} \pm 1.42$ | $14.3 \pm 1.27$ | $67.9 \pm 9.80$ |
| | Llama-3.1-8b(0.5-norm) | 1 | 3 | 14 | 32 | 17 | 80 |
| | w/ control(rank) | $4.0 \pm 0.00$ | $4.4 \pm 1.20$ | $\mathbf{1.7} \pm 0.64$ | $20.8 \pm 1.89$ | $\mathbf{7.5} \pm 0.81$ | $\mathbf{21.5} \pm 2.54$ |
| | w/ control(CI) | $4.0 \pm 0.00$ | $5.0 \pm 0.00$ | $8.2 \pm 1.72$ | $\mathbf{18.2} \pm 1.33$ | $13.3 \pm 1.85$ | $67.9 \pm 9.80$ |
| SciNO (Ours) | Llama-3.1-8b(1-norm) | 3 | 3 | 9 | 32 | 18 | 80 |
| | w/ control(rank) | $1.3 \pm 0.64$ | $3.5 \pm 1.28$ | $3.2 \pm 0.40$ | $10.6 \pm 0.49$ | $\mathbf{9.9} \pm 0.54$ | $20.4 \pm 0.92$ |
| | w/ control(CI) | $1.3 \pm 0.64$ | $3.7 \pm 0.64$ | $\mathbf{2.0} \pm 0.00$ | $\mathbf{9.9} \pm 1.97$ | $11.1 \pm 0.83$ | $\mathbf{18.9} \pm 1.14$ |
| | Llama-3.1-8b(0.5-norm) | 1 | 3 | 14 | 32 | 17 | 80 |
| | w/ control(rank) | $1.3 \pm 0.64$ | $3.8 \pm 0.98$ | $4.1 \pm 0.83$ | $10.5 \pm 0.92$ | $\mathbf{9.1} \pm 0.30$ | $20.4 \pm 0.92$ |
| | w/ control(CI) | $1.3 \pm 0.64$ | $4.8 \pm 0.40$ | $\mathbf{2.9} \pm 0.30$ | $\mathbf{9.8} \pm 1.83$ | $10.1 \pm 0.83$ | $\mathbf{18.9} \pm 1.14$ |

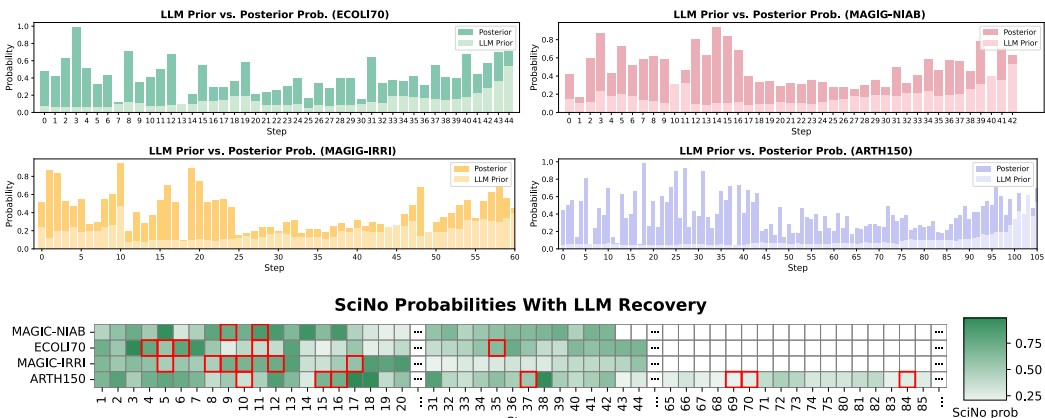

Figure A.6: **(Top)** Comparison of LLM prior and posterior probabilities for the true leaf nodes across four datasets (ECOLI70, MAGIC-NIAB, MAGIC-IRRI, and ARTH150). Posterior probabilities are computed by multiplying the LLM prior with SciNO computed probabilities. **(Bottom)** Heatmap of SciNO probabilities for the SciNO-selected nodes at each ordering step across datasets. Red boxes □ denote steps where SciNO alone fails to select the correct leaf node, but the correct node is successfully recovered via integration with LLM priors.

**Control with *Hard* and *Soft* Supervision** Under the above experimental conditions, we follow the causal ordering procedure described in Algorithm 1. The control mechanism is formalized as:

$$\mathbb{P}(x_{t+1}|x_{1:t}, \texttt{stat}, \texttt{context}) \propto \mathbb{P}_{\text{AR}}(x_{t+1}|x_{1:t}, \texttt{context}) \times \mathbb{P}_{\text{SciNO}}(\texttt{stat}|x_{1:t+1}). \quad (67)$$

Let $\pi_t$ denote the partial causal order determined by $x_{1:t}$, the sequence of nodes selected in the previous steps. The evidence term $\mathbb{P}_{\text{SciNO}}(\texttt{stat}|x_{1:t+1})$ is estimated using two approaches introduced in Section 3.3. The first estimator relies on the average rank $\bar{r}(i)$ for each node $i \in \mathcal{V} \setminus \pi_t$ across models:

$$\widehat{P}_{m\sim M}^{(\texttt{rank})}(\sigma_i^{(m)} < \sigma_j^{(m)}|x_{1:t}, x_{t+1} = i) = \frac{\exp\left(-\bar{r}(i)\right)}{\sum_{j \in \mathcal{V} \setminus \pi_t} \exp\left(-\bar{r}(j)\right)}, \quad \bar{r}(i) = \frac{1}{M}\sum_{m=1}^{M} r_i^{(m)}. \quad (68)$$

Here, $r_i^{(m)}$ is the rank of node $i$ in model $m$, based on the variance of its Hessian diagonal. The second estimator is based on confidence intervals:

$$\widehat{P}_{m\sim M}^{(\texttt{CI})}(\text{CI}_{\text{lower}}(i) \leq \text{CI}_{\text{upper}}(j_{\min}^{(m)})|x_{1:t}, x_{t+1} = i) = \frac{1}{M}\sum_{m=1}^{M} \mathbb{1}\left[\text{CI}_{\text{lower}}(i) \leq \text{CI}_{\text{upper}}(j_{\min}^{(m)})\right]. \quad (69)$$

where $j_{\min}^{(m)}$ is the node with the smallest variance of the Hessian diagonal in model $m$.

Depending on whether variable descriptions are available, we apply two supervision strategies. For variables without descriptions $\mathcal{V}_{\neg\text{context}}$, we use *hard supervision*, relying exclusively on the evidence term estimated by (10) or (12):

$$P(v) \propto \frac{1}{|\mathcal{V} \setminus \pi_t|} \widehat{P}_{\text{SciNO}}(\texttt{stat}|\pi_t, x_{t+1} = v). \tag{70}$$

For variables with descriptions $\mathcal{V}_{\text{context}}$, we apply *soft supervision* by multiplying the LLM-derived prior (65) to the evidence term (10) or (12):

$$P(v) \propto P_{\alpha\text{-norm}}(x_{t+1} = v|\pi_t, \texttt{context}) \cdot \widehat{P}_{\text{SciNO}}(\texttt{stat}|\pi_t, x_{t+1} = v). \tag{71}$$

To adapt the soft supervision signal, one can use a temperature parameter $\tau \geq 0$:

$$\widehat{P}_{\tau}(\texttt{stat}|\pi_t, x_{t+1} = v) = \left( \widehat{P}_{\text{SciNO}}(\texttt{stat}|\pi_t, x_{t+1} = v) \right)^{\tau}, \tag{72}$$

$$\widehat{P}_{\tau-\text{soft}}(\texttt{stat}|\pi_t, v) = \text{softmax}(\widehat{P}_{\tau}(\texttt{stat}|\pi_t, x_{t+1} = v)). \tag{73}$$

We choose $\tau \in \{0, 1, 2, 3, 4, 5\}$ for experiments. When $\tau > 5$, the evidence term grows exponentially and overwhelms the prior, leading to predictions that are nearly indistinguishable from using the data statistic alone. Applying the above $\tau$-temperature to the soft supervision, the posterior predictive term is computed by:

$$P(v) \propto P_{\alpha\text{-norm}}(v|\pi_t, \texttt{context}) \cdot \widehat{P}_{\tau-\text{soft}}(\texttt{stat}|\pi_t, x_{t+1} = v), \quad v \in \mathcal{V} \setminus \pi_t. \tag{74}$$

Then we select the variable with the highest posterior probability as the next leaf node:

$$v^* = \underset{v \in \mathcal{V} \setminus \pi_t}{\text{argmax}} \; P(v). \tag{75}$$

**Results** Table A.8 reports the order divergence under the setting where $\mathcal{V}_{\text{context}} \neq \mathcal{V}$, across varying levels of description proportions and $\tau$ values. Compared to the results in Table 4, performance without control reveals that the LLM's ability to infer causality deteriorates as descriptive content decreases and variable names are masked. Across all datasets, increasing $\tau$ from 0 to 1 consistently improves performance, even under low description conditions (e.g., 10%), indicating robustness in the combination of LLM prior and SciNO evidence. When the description is limited, performance tends to plateau after $\tau = 1$, suggesting that the contribution from LLM priors has been fully exploited, leaving SciNO as the dominant signal. Conversely, with a higher description ratio, the effect of soft supervision becomes more pronounced as $\tau$ varies, allowing the model to better leverage both semantic and structural signals. Interestingly, a higher $\tau$ does not always lead to better results. This highlights the importance of finding an optimal balance between the LLM's semantic knowledge and the evidence from SciNO. We compare various combinations of description proportions and $\tau$ values. Among them, the configuration using 70% descriptions with $\tau = 5$ achieves the highest performance on the ARTH150 dataset. It outperforms all other tested combinations, including those with lower description ratios, different $\tau$ values, and even the use of SciNO alone. This suggests that when richer semantic information is available, the integration with SciNO becomes especially effective. In conclusion, these results demonstrate that even under limited description conditions, combining LLM semantic knowledge with data-driven causal inference can lead to improved performance.

Table A.8: Comparison of OD across semi-synthetic datasets under the control with hard and soft supervision setting with varying description proportions (70%, 40%, 10%) and temperature parameters $\tau \in \{0, 1, 2, 3, 4, 5\}$. The lowest OD for each dataset and description proportion is highlighted in bold. Each score is recorded over 3 independent runs.

| Dataset | Proportion | Method | w/o control | $\tau = 0$ | $\tau = 1$ | $\tau = 2$ | $\tau = 3$ | $\tau = 4$ | $\tau = 5$ |
|---|---|---|---|---|---|---|---|---|---|
| **MAGIC-NIAB** | 10% | Llama-3.1-8b(1-norm) w/ control(rank) | 24.7 ± 9.3 | 5.3 ± 0.6 | **3.3 ± 0.6** | **3.3 ± 0.6** | **3.3 ± 0.6** | 3.7 ± 0.6 | 3.7 ± 0.6 |
| | | Llama-3.1-8b(0.5-norm) w/ control(rank) | 22.3 ± 7.6 | 4.7 ± 0.6 | **3.3 ± 0.6** | **3.3 ± 0.6** | **3.3 ± 0.6** | 3.7 ± 0.6 | 3.7 ± 0.6 |
| | | Llama-3.1-8b(1-norm) w/ control(CI) | 24.7 ± 9.3 | 5.0 ± 1.0 | 4.0 ± 0.0 | 4.0 ± 0.0 | 4.0 ± 0.0 | 4.0 ± 0.6 | 4.0 ± 0.0 |
| | | Llama-3.1-8b(0.5-norm) w/ control(CI) | 22.3 ± 7.6 | 5.0 ± 1.0 | 4.0 ± 0.0 | 4.0 ± 0.0 | 4.0 ± 0.0 | 4.3 ± 0.6 | 4.3 ± 0.6 |
| **ECOLI70** | 70% | Llama-3.1-8b(1-norm) w/ control(rank) | 30.3 ± 2.1 | 35.7 ± 2.9 | 8.7 ± 0.6 | 8.3 ± 0.6 | **8.0 ± 0.0** | **8.0 ± 0.0** | 8.3 ± 0.6 |
| | | Llama-3.1-8b(0.5-norm) w/ control(rank) | 28.7 ± 2.9 | 39.0 ± 4.0 | 9.0 ± 0.0 | 8.3 ± 0.6 | 8.3 ± 0.6 | 8.3 ± 0.6 | 8.7 ± 0.6 |
| | | Llama-3.1-8b(1-norm) w/ control(CI) | 30.3 ± 2.1 | 24.0 ± 2.0 | 11.3 ± 1.5 | 11.7 ± 2.3 | 12.7 ± 1.5 | 12.0 ± 1.0 | 11.7 ± 1.2 |
| | | Llama-3.1-8b(0.5-norm) w/ control(CI) | 28.7 ± 2.9 | 27.3 ± 4.2 | 11.0 ± 2.7 | 12.7 ± 1.5 | 12.0 ± 1.0 | 12.3 ± 3.8 | 10.0 ± 2.0 |
| | 40% | Llama-3.1-8b(1-norm) w/ control(rank) | 33.0 ± 1.7 | 21.7 ± 8.5 | 8.3 ± 0.6 | **8.0 ± 0.0** | **8.0 ± 0.0** | **8.0 ± 0.0** | **8.0 ± 0.0** |
| | | Llama-3.1-8b(0.5-norm) w/ control(rank) | 36.7 ± 9.7 | 20.7 ± 7.5 | 9.3 ± 0.6 | 8.3 ± 0.6 | 8.3 ± 0.6 | 8.3 ± 0.6 | 8.3 ± 0.6 |
| | | Llama-3.1-8b(1-norm) w/ control(CI) | 33.0 ± 1.7 | 17.0 ± 4.4 | 12.3 ± 4.0 | 11.7 ± 0.6 | 11.7 ± 0.6 | 11.7 ± 0.6 | 11.7 ± 0.6 |
| | | Llama-3.1-8b(0.5-norm) w/ control(CI) | 36.7 ± 9.7 | 16.3 ± 3.2 | 12.7 ± 4.0 | 11.0 ± 1.7 | 11.0 ± 1.7 | 11.0 ± 1.7 | 11.0 ± 1.7 |
| | 10% | Llama-3.1-8b(1-norm) w/ control(rank) | 34.3 ± 3.8 | 13.0 ± 3.6 | **8.0 ± 0.0** | **8.0 ± 0.0** | **8.0 ± 0.0** | **8.0 ± 0.0** | **8.0 ± 0.0** |
| | | Llama-3.1-8b(0.5-norm) w/ control(rank) | 33.3 ± 2.5 | 14.0 ± 2.0 | **8.0 ± 0.0** | **8.0 ± 0.0** | **8.0 ± 0.0** | **8.0 ± 0.0** | **8.0 ± 0.0** |
| | | Llama-3.1-8b(1-norm) w/ control(CI) | 34.3 ± 3.8 | 13.0 ± 3.6 | 11.3 ± 0.6 | 11.0 ± 1.0 | 10.7 ± 0.6 | 10.7 ± 1.5 | 10.7 ± 1.5 |
| | | Llama-3.1-8b(0.5-norm) w/ control(CI) | 33.3 ± 2.5 | 12.7 ± 3.8 | 11.0 ± 0.0 | 10.7 ± 0.6 | 10.7 ± 1.5 | 10.7 ± 1.5 | 11.0 ± 1.0 |
| **MAGIC-IRRI** | 10% | Llama-3.1-8b(1-norm) w/ control(rank) | 29.7 ± 1.5 | 11.3 ± 1.2 | 10.3 ± 0.6 | 9.7 ± 0.6 | **9.3 ± 0.6** | 9.7 ± 0.6 | 9.7 ± 0.6 |
| | | Llama-3.1-8b(0.5-norm) w/ control(rank) | 25.3 ± 2.9 | 9.7 ± 2.1 | 9.7 ± 1.2 | 9.7 ± 0.6 | 9.7 ± 0.6 | **9.3 ± 0.6** | 9.7 ± 0.6 |
| | | Llama-3.1-8b(1-norm) w/ control(CI) | 29.7 ± 1.5 | 10.0 ± 1.0 | 9.7 ± 0.6 | 9.7 ± 0.6 | 9.7 ± 0.6 | 9.7 ± 0.6 | 9.7 ± 0.6 |
| | | Llama-3.1-8b(0.5-norm) w/ control(CI) | 25.3 ± 2.9 | 10.3 ± 1.5 | 9.7 ± 0.6 | 9.7 ± 0.6 | 9.7 ± 0.6 | 9.7 ± 0.6 | 9.7 ± 0.6 |
| **ARTH150** | 70% | Llama-3.1-8b(1-norm) w/ control(rank) | 73.0 ± 3.5 | 80.7 ± 1.2 | 20.3 ± 1.2 | 20.7 ± 0.6 | 21.3 ± 1.2 | 21.0 ± 1.0 | 21.0 ± 1.0 |
| | | Llama-3.1-8b(0.5-norm) w/ control(rank) | 64.3 ± 3.5 | 73.3 ± 8.6 | 19.7 ± 0.6 | 19.7 ± 0.6 | 20.0 ± 0.0 | 20.3 ± 0.6 | 20.3 ± 0.6 |
| | | Llama-3.1-8b(1-norm) w/ control(CI) | 73.0 ± 3.5 | 77.3 ± 5.5 | 20.0 ± 1.7 | 19.7 ± 1.2 | 19.0 ± 1.7 | 20.0 ± 2.0 | 19.3 ± 2.1 |
| | | Llama-3.1-8b(0.5-norm) w/ control(CI) | 64.3 ± 3.5 | 71.3 ± 8.1 | 17.7 ± 0.6 | 17.0 ± 1.0 | 16.7 ± 2.1 | **16.3 ± 1.2** | **16.3 ± 1.2** |
| | 40% | Llama-3.1-8b(1-norm) w/ control(rank) | 76.0 ± 10.4 | 81.3 ± 12.9 | 19.7 ± 1.2 | 20.0 ± 1.0 | 20.0 ± 0.0 | 20.0 ± 0.0 | 20.0 ± 0.0 |
| | | Llama-3.1-8b(0.5-norm) w/ control(rank) | 87.7 ± 1.2 | 77.7 ± 9.0 | **19.0 ± 0.0** | 19.3 ± 0.6 | 20.0 ± 0.0 | 20.0 ± 0.0 | 20.0 ± 0.0 |
| | | Llama-3.1-8b(1-norm) w/ control(CI) | 76.0 ± 10.4 | 57.0 ± 25.5 | 22.3 ± 1.2 | 20.7 ± 1.2 | 20.7 ± 1.2 | 20.0 ± 1.7 | 20.0 ± 2.0 |
| | | Llama-3.1-8b(0.5-norm) w/ control(CI) | 87.7 ± 1.2 | 59.3 ± 18.5 | 19.3 ± 1.5 | 21.7 ± 0.6 | 20.7 ± 1.2 | 20.7 ± 1.2 | 19.7 ± 1.5 |
| | 10% | Llama-3.1-8b(1-norm) w/ control(rank) | 76.0 ± 7.6 | 40.7 ± 17.4 | 19.7 ± 0.6 | 20.0 ± 1.0 | 20.0 ± 1.0 | 20.7 ± 1.2 | 20.7 ± 1.2 |
| | | Llama-3.1-8b(0.5-norm) w/ control(rank) | 88.7 ± 6.4 | 37.0 ± 13.8 | 19.7 ± 0.6 | 19.7 ± 0.6 | 20.3 ± 1.5 | 20.7 ± 1.2 | 20.7 ± 1.2 |
| | | Llama-3.1-8b(1-norm) w/ control(CI) | 76.0 ± 7.6 | 24.7 ± 8.1 | 19.3 ± 1.5 | 19.3 ± 1.5 | 19.7 ± 2.1 | **19.0 ± 1.0** | **19.0 ± 1.0** |
| | | Llama-3.1-8b(0.5-norm) w/ control(CI) | 88.7 ± 6.4 | 25.0 ± 7.0 | 19.3 ± 1.5 | 19.3 ± 1.5 | 19.7 ± 2.1 | **19.0 ± 1.0** | **19.0 ± 1.0** |

# B   Miscellaneous

You are an AI assistant tasked with identifying the most likely leaf node in a causal structure.
A leaf node is a variable that does not cause any other variables in the unordered variables set.
Your goal is to determine the best leaf node among **Unordered Variables** using the given information.

Selection Criteria:
- A leaf node does not act as a cause for any other variable in **Unordered Variables**.
- If multiple candidates exist, select the one that is influenced by others but does not influence any other
  variable in **Unordered Variables**.

Important Formatting Rules:
- Respond **only** with the variable name of the selected leaf node.
- Do **not** include any punctuation, reasoning, quotes, or formatting.
- **Leaf Node** must be **exactly one** variable name as plain text, matching one from the **Unordered Variables**
  list.
- Do **not** include any additional text before or after the variable name.

Example 1:
Input :
**Unordered Variables**: ["node_CloudCover", "node_Humidity", "node_Pressure", "node_Temperature"]
**Data Description:** The dataset contains weather data recorded hourly with multiple atmospheric variables.
**Variable Descriptions:**
[
"node_CloudCover": "The fraction of the sky covered by clouds.",
"node_Humidity": "The amount of water vapor in the air.",
"node_Pressure": "The atmospheric pressure at a given location.",
"node_Temperature": "The measure of how hot or cold the air is."
]
Output : node_Temperature

Input :
**Unordered Variables:** {unordered_variables}
**Data Description:** {data_description}
**Variable Descriptions:** {variable_description}

Output :

Figure A.7: Prompting for Leaf Node Selection

**Physics** ( $\alpha$(=alpha)=0.5, Control: rank-based )

**=== Causal Ordering Step  t=1 ===**

**LLM Prior:** {'Weight of object': 0.303, 'Moisture Content of object': 0.205, 'Rainfall': 0.144, ..., 'Total Solar Irradiance': 0.071, 'Surface Air Temperature': 0.068}
**SciNO Likelihood:** {'Weight of object': 0.635, 'Rainfall': 0.233, 'Moisture Content of object': 0.086, ..., 'Wind Speed': 0.003, 'Total Solar Irradiance': 0.002}
**Updated Posterior:** {'Weight of object': 0.776, 'Rainfall': 0.136, 'Moisture Content of object': 0.071, ..., 'Wind Speed': 0.001, 'Total Solar Irradiance': 0.001}

**Current Causal Order $\pi_1$:** [Weight of object]

**=== Causal Ordering Step t=2 ===**

**LLM Prior:** {'Moisture Content of object': 0.288, 'Rainfall': 0.220, 'Rate of Evaporation': 0.219, ..., 'Total Solar Irradiance': 0.086, 'Surface Air Temperature': 0.086}
**SciNO Likelihood:** {'Rainfall': 0.644, 'Moisture Content of object': 0.214, 'Surface Air Temperature': 0.084, ..., 'Wind Speed': 0.010, 'Total Solar Irradiance': 0.005}
**Updated Posterior:** {'Rainfall': 0.641, 'Moisture Content of object': 0.278, 'Rate of Evaporation': 0.040, ..., 'Wind Speed': 0.005, 'Total Solar Irradiance': 0.002}

**Current Causal Order $\pi_2$:** [Weight of object, Rainfall]

(.....)

**=== Causal Ordering Step t=5 ===**

**LLM Prior:** {'Wind Speed': 0.464, 'Total Solar Irradiance': 0.300, 'Surface Air Temperature': 0.235}
**SciNO Likelihood:** {'Surface Air Temperature': 0.677, 'Wind Speed': 0.218, 'Total Solar Irradiance': 0.104}
**Updated Posterior:** {'Surface Air Temperature': 0.545, 'Wind Speed': 0.346, 'Total Solar Irradiance': 0.107}

**Current Causal Order $\pi_5$:** [Weight of object, Rainfall, ... ,  Surface Air Temperature]

**=== Causal Ordering Step t=6 ===**

**LLM Prior:** {'Wind Speed': 0.584, 'Total Solar Irradiance': 0.415}
**SciNO Likelihood:** {'Wind Speed': 0.704, 'Total Solar Irradiance': 0.295}
**Updated Posterior:** {'Wind Speed': 0.769, 'Total Solar Irradiance': 0.230}

**Current Causal Order $\pi_6$:** [Weight of object, Rainfall, .... ,  Wind Speed]

**=== Final Results ===**

**Complete Causal Order:** [Total Solar Irradiance, Wind Speed, ..., Rainfall, Weight of object]
**Order Divergence:** 1

Figure A.8: LLM control log on the Physics dataset.

**Sachs** ( $\alpha$(=alpha)=1.0, Control: CI-based )

**=== Causal Ordering Step  t=1 ===**

**LLM Prior:** {'pakts473': 0.195, 'p44/42': 0.166, 'pjnk': 0.163, ..., 'PIP3': 0.077, 'P38': 0.071}
**SciNO Likelihood:** {'pmek': 0.125, 'PIP3': 0.125, 'p44/42': 0.125, ..., 'P38': 0.125, 'pjnk': 0.12}
**Updated Posterior:** {'pakts473': 0.195, 'p44/42': 0.166, 'pjnk': 0.163, ..., 'PIP3': 0.077, 'P38': 0.071}

**Current Causal Order $\pi_1$:** [pakts473]

**=== Causal Ordering Step t=2 ===**

**LLM Prior:** {'p44/42': 0.213, 'PKC': 0.210, 'pjnk': 0.204, ..., 'PIP3': 0.089, 'P38': 0.074}
**SciNO Likelihood:** {'pmek': 0.142, 'PIP3': 0.142, 'p44/42': 0.143, ..., 'P38': 0.143, 'pjnk': 0.142}
**Updated Posterior:** {'p44/42': 0.213, 'PKC': 0.210, 'pjnk': 0.204, ..., 'PIP3': 0.089, 'P38': 0.074}
**Current Causal Order $\pi_2$:** [pakts473, p44/42]

(.....)

**=== Causal Ordering Step t=6 ===**

**LLM Prior:** {'P38': 0.496, 'PKA': 0.286, 'PIP3': 0.217}
**SciNO Likelihood:** {'PIP3': 0.416, 'P38': 0.416, 'PKA': 0.166}
**Updated Posterior:** {'P38': 0.599, 'PIP3': 0.262, 'PKA': 0.138}

**Current Causal Order $\pi_6$:** [pakts473, p44/42, ..., P38]

**=== Causal Ordering Step t=7 ===**

**LLM Prior:** {'PIP3': 0.605, 'PKA': 0.394}
**SciNO Likelihood:** {'PIP3': 0.500, 'PKA': 0.500}
**Updated Posterior:** {'PIP3': 0.605, 'PKA': 0.394}

**Current Causal Order $\pi_7$:** [pakts473, p44/42, ..., P38, PIP3]

**=== Final Results ===**

**Complete Causal Order:** [PKA, PIP3, ... , p44/42, pakts473]
**Order Divergence:** 3

Figure A.9: LLM control log on the Sachs dataset.

**MAGIC-NIAB** ( $\alpha$(=alpha)=0.5, Control: rank-based )

**=== Causal Ordering Step  t=1 ===**

**LLM Prior:** {'YR.GLASS': 0.176, 'YLD': 0.142, 'HT': 0.118, …, 'G1750': 0.019, 'G1276': 0.016}
**SciNO Likelihood:** {'YR.FIELD': 0.328, 'FT': 0.296, 'YLD': 0.297, …, 'G1294': 0.000, 'G1276': 0.000}
**Updated Posterior:** {'YLD': 0.424, 'FT': 0.311, 'YR.FIELD': 0.216, …, 'G1276': 0.000, 'G1294': 0.000}

**Current Causal Order $\pi_1$:** [YLD]

**=== Causal Ordering Step t=2 ===**

**LLM Prior:** {'FT': 0.161, 'HT': 0.149, 'G43': 0.108, …, 'G1750': 0.022, 'G800': 0.015}
**SciNO Likelihood:** {'YR.FIELD': 0.443, 'FT': 0.414, 'FUS': 0.102, …, 'G1276': 0.000, 'G1263': 0.000}
**Updated Posterior:** {'FT': 0.712, 'YR.FIELD': 0.171, 'FUS': 0.111, …, 'G1294': 0.000, 'G1263': 0.000}

**Current Causal Order $\pi_2$:** [YLD, FT]

(…..)

**=== Causal Ordering Step t=42 ===**

**LLM Prior:** {'G418': 0.359, 'G1750': 0.347, 'G1217': 0.292}
**SciNO Likelihood:** {'G418': 0.685, 'G1750': 0.193, 'G1217': 0.121}
**Updated Posterior:** {'G418': 0.706, 'G1750': 0.192, 'G1217': 0.101}

**Current Causal Order $\pi_{42}$:** [YLD, FT, …, G418]

**=== Causal Ordering Step t=43 ===**

**LLM Prior:** {'G1750': 0.527, 'G1217': 0.472}
**SciNO Likelihood:** {'G1217': 0.660, 'G1750': 0.339}
**Updated Posterior:** {'G1217': 0.635, 'G1750': 0.364}

**Current Causal Order $\pi_{43}$:** [YLD, FT, …, G418, G1217]

**=== Final Results ===**

**Complete Causal Order:** [G1750, G1217, …, G38, FT, YLD]
**Order Divergence: 2**

Figure A.10: LLM control log on the MAGIC-NIAB dataset.

**ECOLI70** ( α(=alpha)=0.5, Control: rank-based )

**=== Causal Ordering Step t=1 ===**

**LLM Prior:** {'atpG': 0.072, 'asnA': 0.054, 'atpD': 0.054, ..., 'b1191': 0.030, 'yedE': 0.027}
**SciNO Likelihood:** {'ygbD': 0.742, 'tnaA': 0.193, 'nmpC': 0.030, ..., 'sucD': 0.000, 'yecO': 0.000}
**Updated Posterior:** {'ygbD': 0.476, 'tnaA': 0.440, 'atpG': 0.058, ..., 'yecO': 0.000, 'sucD': 0.000}

**Current Causal Order $\pi_1$:** [ygbD]

**=== Causal Ordering Step t=2 ===**

**LLM Prior:** {'atpG': 0.060, 'yheI': 0.048, 'asnA': 0.046, ..., 'b1583': 0.033, 'b1191': 0.032}
**SciNO Likelihood:** {'cspA': 0.720, 'tnaA': 0.274, 'pspA': 0.003, ..., 'dnaG': 0.000, 'yecO': 0.000}
**Updated Posterior:** {'cspA': 0.579, 'tnaA': 0.418, 'pspA': 0.001, ..., 'dnaG': 0.000, 'icdA': 0.000}

**Current Causal Order $\pi_2$:** [ygbD, cspA]

(.....)

**=== Causal Ordering Step t=44 ===**

**LLM Prior:** {'b1191': 0.359, 'cspG': 0.352, 'eutG': 0.287}
**SciNO Likelihood:** {'b1191': 0.669, 'eutG': 0.214, 'cspG': 0.115}
**Updated Posterior:** {'b1191': 0.701, 'eutG': 0.179, 'cspG': 0.118}

**Current Causal Order $\pi_{44}$ :** [ygbD, cspA, ..., b1191]

**=== Causal Ordering Step t=45 ===**

**LLM Prior:** {'eutG': 0.540, 'cspG': 0.459}
**SciNO Likelihood:** {'eutG': 0.673, 'cspG': 0.326}
**Updated Posterior:** {'eutG': 0.708, 'cspG': 0.291}

**Current Causal Order $\pi_{45}$:** [ygbD, cspA, ..., b1191, eutG]

**=== Final Results ===**

**Complete Causal Order:** [cspG, eutG, ..., cspA, ygbD]
**Order Divergence:** 10

Figure A.11: LLM control log on the ECOLI70 dataset.

**MAGIC-IRRI** ( $\alpha$(=alpha)=0.5, Control: rank-based )

=== Causal Ordering Step t=1 ===

**LLM Prior:** {'YLD': 0.238, 'HT': 0.070, 'BROWN': 0.069, ..., 'G3092': 0.023, 'AMY': 0.021}
**SciNO Likelihood:** {'CHALK': 0.664, 'GW': 0.158, 'YLD': 0.134, ..., 'FT': 0.000, 'AMY': 0.000}
**Updated Posterior:** {'YLD': 0.518, 'CHALK': 0.445, 'BROWN': 0.026, ..., 'AMY': 0.000, 'FT': 0.000}

**Current Causal Order $\pi_1$:** [YLD]

=== Causal Ordering Step t=2 ===

**LLM Prior:** {'HT': 0.119, 'CHALK': 0.099, 'BROWN': 0.093, ..., 'G3927': 0.030, 'G3098': 0.024}
**SciNO Likelihood:** {'CHALK': 0.652, 'GW': 0.232, 'BROWN': 0.068, ..., 'AMY': 0.000, 'GTEMP': 0.000}
**Updated Posterior:** {'CHALK': 0.866, 'BROWN': 0.084, 'GW': 0.045, ..., 'FT': 0.000, 'HT': 0.000}

**Current Causal Order $\pi_2$:** [YLD, CHALK]

(.....)

=== Causal Ordering Step t=62 ===

**LLM Prior:** {'G3925': 0.379, 'G3823': 0.360, 'G4156': 0.260}
**SciNO Likelihood:** {'G4156': 0.439, 'G3925': 0.284, 'G3823': 0.275}
**Updated Posterior:** {'G4156': 0.355, 'G3925': 0.335, 'G3823': 0.308}

**Current Causal Order $\pi_{62}$:** [YLD, CHALK, ..., G4156]

=== Causal Ordering Step t=63 ===

**LLM Prior:** {'G3823': 0.690, 'G3925': 0.309}
**SciNO Likelihood:** {'G3925': 0.689, 'G3823': 0.310}
**Updated Posterior:** {'G3823': 0.500, 'G3925': 0.499}

**Current Causal Order $\pi_{63}$:** [YLD, CHALK, ..., G4156, G3823]

=== Final Results ===

**Complete Causal Order:** [G3925, G3823, ..., CHALK, YLD]
**Order Divergence:** 9

Figure A.12: LLM control log on the MAGIC-IRRI dataset.

**ARTH150** ( $\alpha$(=alpha)=0.5, Control: CI-based )

**=== Causal Ordering Step t=1 ===**

**LLM Prior:** {'519': 0.066, '78': 0.044, '93': 0.037, ..., '4': 0.025, '96': 0.024}
**SciNO Likelihood:** {'496': 0.500, '677': 0.500, '4': 0.000, ..., '47': 0.000, '61': 0.000}
**Updated Posterior:** {'496': 0.662, '677': 0.337, '4': 0.000, ..., '47': 0.000, '61': 0.000}

**Current Causal Order** $\pi_1$: [496]

**=== Causal Ordering Step t=2 ===**

**LLM Prior:** {'519': 0.073, '78': 0.047, '93': 0.040, ..., '783': 0.026, '539': 0.025}
**SciNO Likelihood:** {'677': 1.000, '4': 0.000, '8': 0.000, ..., '61': 0.000, '63': 0.000}
**Updated Posterior:** {'677': 1.000, '4': 0.000, '8': 0.000, ..., '61': 0.000, '63': 0.000}

**Current Causal Order** $\pi_2$**:** [496, 677]

(.....)

**=== Causal Ordering Step t=105 ===**

**LLM Prior:** {'738': 0.378, '539': 0.311, '783': 0.309}
**SciNO Likelihood:** {'539': 0.333, '738': 0.333, '783': 0.333}
**Updated Posterior:** {'738': 0.378, '539': 0.311, '783': 0.309}

**Current Causal Order** $\pi_{105}$ **:** [496, 677, ..., 738]

**=== Causal Ordering Step t=106 ===**

**LLM Prior:** {'783': 0.533, '539': 0.466}
**SciNO Likelihood:** {'539': 0.967, '783': 0.032}
**Updated Posterior:** {'539': 0.963, '783': 0.036}

**Current Causal Order** $\pi_{106}$ : [496, 677, ..., 738, 539]

**=== Final Results ===**

**Complete Causal Order:** [783, 539, ..., 677, 496]
**Order Divergence:** 18

Figure A.13: LLM control log on the ARTH150 dataset.

