# OpenReview forum: "Score-informed Neural Operator for Enhancing Ordering-based Causal Discovery"
_NeurIPS.cc/2025/Conference — NeurIPS 2025 poster_

### Official Review · Reviewer_uvzA · 2025-06-11

**Clarity:** 4
**Significance:** 3
**Originality:** 2
**Rating:** 5
**Confidence:** 5

**Summary:**

This paper focuses on a key challenge of ordering-based causal discovery: the computationally expensive and memory-intensive issues of score's Jacobian (Hessian) diagonal estimation. Motivated by FNO and DiffAN, this paper proposes a new time-conditioning score-based generative model, Score-informed Neural Operator (SciNO), to stably approximate the Hessian diagonal by modeling the score function with neural operators. This architecture can improve the scalability of the existing algorithms, e.g., SCORE and CaPS, by reducing their cubic computational complexity. Additionally, combining LLM with the leaf node discrimination criteria in previous works, this paper proposes a probabilistic control of autoregressive causal ordering algorithm. With the approximation power of neural operators in Theorem 3.1, the idea is theoretically sounds. Some synthetic and real-world experiments are presented.

**Questions:**

1. Could you further explain what the subgraph below in Figure 5 intends to express? What specific distributions do the prior and posterior here refer to? What does this change indicate?

2. See strengths and weaknesses. Most of my concerns are about the experimental part. I will adjust the score if the above experiments can reach a consistent conclusion.

**Ethical Concerns:**

["NO or VERY MINOR ethics concerns only"]

**Final Justification:**

See the feedback in my official comment.

**Limitations:**

yes

**Paper Formatting Concerns:**

In the template of NeurIPS, the caption of the table seems to be at the top rather than the bottom.

**Quality:**

3

**Strengths And Weaknesses:**

Pros:

1. This paper is well-motivated. Improving the estimation efficiency of Hessian diagonal can improve the scalability of many existing ordering-based causal discovery methods.

2. Theoretical motivations are clearly explained and the proof are adequately provided. Theorem 3.1 shows the approximation power of neural operators. The theorems and mathematical details appear to be correct, though I haven't checked all the details.

3. Despite some formatting issues, this paper is well-written and easy to understand.

Cons:

1. Lack of some important experiment. Although the main purpose of this paper is to reduce the computational complexity of Hessian diagonal estimation, the accuracy of the approximation is also important. Here I have two main concerns.

* About the accuracy of Hessian diagonal estimation, SciNO can have a better or comparable performance compared to the second-order Stein gradient estimator? The author can illustrate this point by adding the second-order Stein gradient estimator in figure 2 or by presenting a comparison of order divergence.

* Although SciNO can enhance the scalability of CaPS, it seems that it cannot stably improve the performance of CaPS. Does SciNO perform poorly on low-dimensional data? Providing more datasets (like Table 2) can be helpful for this analysis.

2. This paper seems to be concatenated together by two independent parts, one is SciNO and the other is the autoregressive LLM control algorithm. I suggest that at least a comparison between the LLM-based method and the previous methods, such as DiffAN and CaPS, should be provided.

---

> ### Author Rebuttal · Authors · 2025-07-31
>
> We appreciate your valuable comments and suggestions.
>
> **W1-(a). Accuracy of Hessian diagonal approximation**
>
> To assess the accuracy of Hessian diagonal estimation, we provide a comparison between SciNO and the second-order Stein gradient estimator. Since we are unable to include images, we provide the mean and standard deviation of the estimation errors for each graph used in Figure 2 in the following table. As shown, SciNO yields more accurate and stable estimates, with errors closer to zero and lower variance than the Stein-based estimator.
>
> | Graph\Method | Stein gradient estimator | SciNO  |
> | --- | --- | --- |
> | 1 | -**0.03**±1.15 | -0.05±0.72 |
> | 2 | -0.1±1.21 | -**0.04**±0.5 |
> | 3 | -0.22±1.63 | -**0.03**±0.21 |
> | 4 | -0.07±1.13 | -**0.01**±0.07 |
> | 5 | -**0.01**±1.01 | -0.02±0.08 |
> | 6 | -0.22±1.56 | -**0.04**±0.06 |
> | 7 | -0.1±0.92 | -**0.05**±1.28 |
> | 8 | -**0.03**±1.01 | -0.05±0.22 |
> | 9 | -0.18±1.56 | -**0.06**±0.11 |
>
> Furthermore, as presented in answer for **W3-(a)** from **Reviewer 2fQy,** we compared SciNO to the SCORE, which is based on the Stein gradient estimator. SciNO generally achieves a lower Order Divergence than SCORE, particularly in settings high-dimensional datasets. This combined evidence highlights SciNO’s superiority over the second-order Stein gradient estimator in accuracy.
>
> **W1-(b). Stability of CaPS w/ SciNO on low-dimensional datasets**
>
> To assess potential performance degradation, we compared CaPS and SciNO on five low-dimensional datasets.
>
> | Dataset\Method(metric) | CaPS(OD) | CaPS w/ SciNO(OD) |
> | --- | --- | --- |
> | ER(d2) | 0.3±0.46 | **0.0**±0.0 |
> | ER(d3) | 0.4±0.66 | **0.2**±0.4 |
> | ER(d5) | **0.6**±0.92 | **0.6**±0.92 |
> | Physics(d7) | **0** | **0.0**±0.0 |
> | Sachs(d8) | 5 | **4.4**±2.29 |
>
> These results confirm that SciNO does not degrade performance in low-dimensional settings. In all five cases, SciNO either improved the performance of CaPS or maintained it.
>
> **W2. Connection between causal discovery and LLM control, with comparison to previous methods**
>
> Our research is motivated by [Vashishtha et al., 2025], which demonstrated the benefits of using LLMs for ordering tasks (L27-28). We aim to improve ordering-based causal discovery by guiding it with LLM priors. Because existing pairwise-query approaches (L67–68) cannot be directly applied to SciNO due to structural differences, we introduce a novel integration method in which the LLM sequentially predicts leaf nodes given the inferred causal order (L180–181). This method depends on the accuracy of the Hessian-diagonal statistics in Equation (8). Since Figures 2 and A.3 show that SciNO provides a more stable and accurate estimate than DiffAN, we focused our main experiments exclusively on SciNO. In response to the reviewer’s suggestion, we conducted additional experiments applying our control method to existing approaches.
>
> 1. **Comparison with DiffAN-based control**
>
>     The table is provided in our response to **Reviewer 2jQc**’s **Q3** and will be added to Table 4 of the revised manuscript. DiffAN-based control yields a 49% average improvement in order divergence over uncontrolled Llama, which is lower than SciNO’s 64%. This clearly demonstrates that our superior performance arises from both the novel control method and the SciNO’s stable, accurate evidence.
>
> 2. **Comparison with SCORE-based control**
>
>     To benchmark against kernel methods (e.g. SCORE, CaPS) that cannot leverage Deep Ensembles [Lakshminarayanan et al., 2017], we used 1,000-sample bootstrapping. As noted in L186, our focus is the minimal-variance criterion eq.(3), so we conducted SCORE-based control. Although this yields large improvements on MAGIC-NIAB, it does not outperform SCORE alone on ECOLI70 and ARTH150, indicating that LLM priors lack robust synergy with kernel-based methods.
>
> - SCORE-based control (Comparison of order divergence between uncontrolled Llama and controlled Llama)
>
>     | Method\Dataset | MAGIC-NIAB(d44) | ECOLI70(d46) | MAGIC-IRRI(d64) | ARTH150(d107) |
>     | --- | --- | --- | --- | --- |
>     | Llama-3.1-8b(1-norm) | 9 | 32 | 18 | 80 |
>     | w/ control(rank) | **1** (↓88.9%) | **14** (↓56.3%) | 11 (↓38.9%) | 45 (↓43.8%) |
>     | w/ control(CI) | **1** (↓88.9%) | **14** (↓56.3%) | 11 (↓38.9%) | **43** (↓46.3%) |
>     | Llama-3.1-8b(0.5-norm) | 14 | 32 | 17 | 80 |
>     | w/ control(rank) | **1** (↓92.9%) | 16 (↓50.0%) | 10 (↓41.2%) | 45 (↓43.8%) |
>     | w/ control(CI) | 2 (↓85.7%) | 16 (↓50.0%) | **7** (↓58.8%) | **43** (↓46.3%) |
>
> - SCORE
>
>     | Method\Dataset | MAGIC-NIAB(d44) | ECOLI70(d46) | MAGIC-IRRI(d64) | ARTH150(d107) |
>     | --- | --- | --- | --- | --- |
>     | SCORE  | 8 | 7 | 17 | 38 |
>
>
> **Q1. Detailed explanation of what Figure 5's subgraph shows**
>
> As noted in L266-268, the bottom of the Figure 5 shows the probability of selecting a ground-truth leaf node at each causal ordering step on ECOLI70. When multiple ground-truth leaf nodes exist at a given step, we plot the highest probability among them.
>
> The green bars represent the initial probabilities based solely on the LLM prior, \\(\\mathbb{P}\_{\\mathrm{AR}}(x_{t+1}|x_{1:t},\\mathtt{context}) \\) from eq.(8). In high-dimensional settings, this prior often produces low, uninformative probabilities. Our control method combines these with SciNO’s evidence via eq.(8), yielding the posterior probabilities \\( \\mathbb{P}(x_{t+1}|x_{1:t},\\mathtt{stat},\\mathtt{context}) \\) (shown as pink bars).
>
> As discussed in L268-270, the dramatic rise in the ground-truth node’s probability demonstrates that, even if the LLM prior is inaccurate, the data-driven evidence compensates for this inaccuracy and enables reliable causal ordering. Crucially, during the initial steps when errors would otherwise accumulate, these probabilities are effectively boosted. We observe the same corrective effect consistently across high-dimensional datasets (Figure A.5).
>
> ---
> [Vashishtha et al., 2025] — Causal Order: The Key to Leveraging Imperfect Experts in Causal Inference. ICLR, 2025.
>
> [Lakshminarayanan et al., 2017]  — Simple and scalable predictive uncertainty estimation using deep ensembles. NeurIPS, 2017.

---

> > ### Comment · Reviewer_uvzA · 2025-08-03
> > **Thank you for your response**
> >
> > Thank you for these further clarifications. All my concerns have been address, so I am raising my score accordingly (rating: 4->5, quality: 2 -> 3, clarity: 3 -> 4). Please remember to add the detailed comparison of the second-order stein gradient estimator and other additional result to the latest version.

---

> > > ### Author Response · Authors · 2025-08-03
> > > **Re: Thank you for your response**
> > >
> > > We appreciate your valuable feedback and the revised evaluation!
> > >
> > > > Please remember to add the detailed comparison of the second-order stein gradient estimator and other additional result to the latest version.
> > >
> > > In response, we will incorporate a comparison of the second-order Stein gradient estimator and supplement our manuscript with additional experimental results to address your concerns.

---

### Official Review · Reviewer_2jQc · 2025-06-18

**Clarity:** 3
**Significance:** 2
**Originality:** 3
**Rating:** 5
**Confidence:** 3

**Summary:**

This paper studies the problem of ordering based causal discovery. Identifying that an existing method DiffAN suffers from unstable gradient estimation,  authors proposed Score-inforrmed Neural Operator (SciNO) to stably approximate the Hessian diagonal and to preserve structural information during the score modeling.  This is achieved by introducing two key modifications to the time-conditional Fourier neural operator architecture: a learnable time encoding module and that decomposition of signals in Fourier layers into real and imaginary parts. Further, the paper proposes a probabilistic control method that leverages score-informed statistics inferred by SciNO, guiding autoregressive models to generate more reliable causal reasoning. Empirical results validate that the improved stability of gradient estimation leads to better causal discovery performance compared with DiffAN.

**Questions:**

1. can you give more details on : "injecting noise into the data can destroy the functional information inherent in the original score function" (line 50-51) ？

2. line 150 :".. allows SciNO to achieve robust performance in high-dimensional causal graphs": 'robust performance' is vague here. Do you mean the trend of SHDs along number of nodes or the standard deviation? Please make it accurate.

3. regarding Section 3.3: it seems that the probabilistic control can also be applied to the original DiffAN? if so, what would be the difference?

4. **Major Concern**: The major weakness is the limited empirical comparison, lacking classic/existing benchmark methods. In particular, the paper only compares the proposed method with its base method DiffAN. It is not clear how the algorithm performs within the whole causal discovery literature. For example, the CAM method shall  perform well on the synthetic data setup of this paper. See, e.g., [1, 2] for references, where their setup is the identical or at least very similar to the one in this paper. In particular, for ER4d (d=100) case, CAM achieves  98.8$\pm$ 20.7 SHD, while the proposed method achieves 180.0$\pm$15.45. Of course, a new algorithm does not need to beat all existing methods, but this large difference needs further clarifying.

    [1] Gradient-Based Neural DAG Learning, ICLR, 2020.

   [2] Ordering based causal discovery with reinforcement learning, IJCAI, 2021.

5. minor on paper writing/organization:

  - line 150: almost half of section 3.2 is about ablation study or case study, which distracts the reading flow. Suggest to move it to experiment section.
  - tables 1-4: table caption is generally placed on top.
  - line 126: "HDM implements time-conditioned Fourier Neural Operator (FNO, [22]) by adding projected tensors from PE (Positional Embedding, [40]) ": be consistent about the format of abbreviation.

**Ethical Concerns:**

["NO or VERY MINOR ethics concerns only"]

**Final Justification:**

Authors address all my concerns and validate the empirical performance by new experimental results with CAM, which is beyond my expectation. I suggest authors add more benchmark methods, e.g., those gradient-based algorithms in the same table in future versions.

**Limitations:**

Yes.

And see above the limitation on lacking important benchmark methods. At least further clarification is needed.

**Quality:**

3

**Strengths And Weaknesses:**

Strengths:

- The paper identifies the key limitation of an existing algorithm, and the modification is sound.
- The proposed method is efficient, compared with  DiffAN.
- Writing is generally good (though can be further improved; see below.)

Weakness:

- Lacking comparison with classic algorithms.
- Novelty is a bit limited.

---

> ### Author Rebuttal · Authors · 2025-07-31
>
> We are grateful for your detailed and helpful comments.
>
> **Q1. Details on how noise injection destroys functional information**
>
> The functional information refers to the underlying functional relationships between variables, such as the derivatives of the score function. Diffusion models learn data distributions by injecting noise into the input, but this process can destroy these functional relationships. As shown in Figure 2, MLP-based diffusion models poorly estimate score derivatives, failing to preserve functional information. Furthermore, Figure 3 shows a weak correlation between MMD and OD when using a standard MLP architecture, providing additional evidence that MLP-based diffusion models fail to capture the functional information effectively.
>
> **Q2. Clarification of 'robust performance' (L150)**
>
> We agree that the phrase 'robust performance' can be vague. Our use of 'robust performance' was intended to describe the trend of performance as the number of nodes increases, rather than its standard deviation as shown in Table 1. To accurately reflect this, we will revise the sentence to state that SciNO 'achieves a performance advantage over baselines that grows with the graph's dimensionality, highlighting its superior scaling properties'.
>
> **Q3. Applicability of the control method to DiffAN**
>
> The probabilistic control method proposed in Section 3.3 uses Hessian‐diagonal statistics, therefore it can be applied to DiffAN as well. Fundamentally, its performance hinges on the quality of the evidence term in eq.(8), i.e., the Hessian‐diagonal statistics. As we shown in Figure 2 and Figure A.3, SciNO provides a more stable and accurate estimation of the Hessian than DiffAN, so we exclusively use SciNO for all control experiments in the paper.
>
> The table below reports results obtained under the same settings as Table 4, but with the evidence term extracted from DiffAN. On high-dimensional graphs, using SciNO's evidence leads to an average 64% reduction in order divergence (Table 4), versus only 49% with DiffAN. This clearly demonstrates that our superior performance arises from both the novel control method and the stable, accurate evidence provided by SciNO.
> - DiffAN-based control (Comparison of order divergence between uncontrolled Llama and controlled Llama)
>     | Method\Dataset | Physics(d7) | Sachs(d8) | MAGIC-NIAB(d44) | ECOLI70(d46) | MAGIC-IRRI(d64) | ARTH150(d107) |
>     | --- | --- | --- | --- | --- | --- | --- |
>     | Llama-3.1-8b(1-norm) | 3 | 3 | 9 | 32 | 18 | 80 |
>     | w/ control(rank) | 4 (↑33.3%) | 3 | 2 (↓77.8%) | 22 (↓31.3%) | **7** (↓61.1%) | **25** (↓68.8%) |
>     | w/ control(CI) | 3 | 4 (↑33.3%) | 6 (↓33.3%) | **16** (↓50.0%) | 13 (↓27.8%) | 52 (↓35.0%) |
>     | Llama-3.1-8b(0.5-norm) | 1  | 3 | 14 | 32 | 17 | 80 |
>     | w/ control(rank) | 4 (↑300%) | 3 | **1** (↓92.9%) | 20 (↓37.5%) | 8 (↓52.9%) | **25** (↓68.8%) |
>     | w/ control(CI) | 4 (↑300%) | 5 (↑66.7%) | 8 (↓42.9%) | 18 (↓43.8%) | 14 (↓17.6%) | 52 (↓35.0%) |
>
> - SciNO-based control (Comparison of order divergence between uncontrolled Llama and controlled Llama, Table 4)
>     | Method\Dataset | Physics(d7) | Sachs(d8) | MAGIC-NIAB(d44) | ECOLI70(d46) | MAGIC-IRRI(d64) | ARTH150(d107) |
>     | --- | --- | --- | --- | --- | --- | --- |
>     | Llama-3.1-8b(1-norm) | 3 | 3 | 9 | 32 | 18 | 80 |
>     | w/ control(rank) | 1 (↓66.7%) | 5 (↑66.7%) | 4 (↓55.6%) | **11** (↓65.6%) | **9** (↓50.0%) | 21 (↓73.8%) |
>     | w/ control(CI) | 1 (↓66.7%) | 3 | **2** (↓77.8%) | 15 (↓53.1%) | 10 (↓44.4%) | **18** (↓77.5%) |
>     | Llama-3.1-8b(0.5-norm) | 1  | 3 | 14 | 32 | 17 | 80 |
>     | w/ control(rank) | 1 | 4 (↑33.3%) | **2** (↓85.7%) | **10** (↓68.8%) | **9** (↓47.1%) | 21 (↓73.8%) |
>     | w/ control(CI) | 1 | 4 (↑33.3%) | **2** (↓85.7%) | 14 (↓56.3%) | 10 (↓41.2%) | **18** (↓77.5%) |
>
> **Q4. Performance gap in SHD between SciNO and CAM**
>
> We would like to clarify that our work concentrates on identifying and addressing the limitations of Ordering-based Causal Discovery methods that identify leaf nodes based on the Hessian diagonal.
>
> To better contextualize the performance gap, we evaluated CAM [Bühlmann et al., 2014] under the same experimental setup. Note that our datasets differ from those in [Lachapelle et al., 2020], [Wang et al., 2021] as they are newly generated. SciNO achieves consistently lower SHD on the high-dimensional ER datasets and superior performance on the semi-synthetic datasets for all datasets except ARTH150.
>
> | Dataset\Method(metric) | CAM(SHD) | CAM(SID) | DiffAN w/ SciNO(SHD) | DiffAN w/ SciNO(SID) |
> | --- | --- | --- | --- | --- |
> | ER(d2) | **0.0**±0.0 | **0.0**±0.0 | **0.0**±0.0 | **0.0**±0.0 |
> | ER(d3) | **0.0**±0.0 | **0.0**±0.0 | 0.2±0.6 | 0.4±1.2 |
> | ER(d5) | **0.0**±0.0 | **0.0**±0.0 | 1.7±1.95 | 4.8±4.94 |
> | ER(d10) | **16.6**±4.45 | **30.9**±11.96 | 20.5±3.5 | 41.6±9.31 |
> | ER(d30) | 129±14.99 | **450.2**±99.2 | **88.8**±16.5 | 492.0±91.64 |
> | ER(d50) | 238.5±17.29 | **1503.6**±201.35 | **180.0**±15.45 | 1622.0±110.92 |
> | ER(d100) | 537.2±45.37 | **6289.9**±533.87 | **445.9**±32.84 | 7259.2±653.27 |
> | Physics(d7) | 12 | 11 | **5.8**±0.40 | **7.8**±1.94 |
> | Sachs(d8) | 37 | 41 | **13.8**±1.72 | **19.4**±5.89 |
> | MAGIC-NIAB(d44) | 167 | 384 | **67.4**±2.87 | **156.6**±18.04 |
> | ECOLI70(d46) | 184 | 828 | **91.0**±7.16 | **734.0**±50.73 |
> | MAGIC-IRRI(d64) | 261 | 791 | **135.4**±5.12 | **333.2**±17.34 |
> | ARTH150(d107) | **437** | 2133 | 454.6±7.66 | **1740.4**±108.63 |
>
> Therefore, our method enhances upon existing score-matching based Causal Ordering methods and is also highly competitive with established Causal Discovery benchmarks like CAM. As detailed in our response to **Reviewer Vgze (W3)** our method also significantly outperforms other classic algorithms like PC [Spirtes et al., 2000] and GES [Chickering, 2002]. We appreciate the reviewer’s suggestion and will include these results in the paper.
>
> **Q5-(a). Placement of case study in method section**
>
> We agree with the reviewer’s point about improving the paper’s flow. Section 3.2 includes a critical ablation study and analytical findings that support the design choices and effectiveness of our proposed method. Table 1 presents an ablation study comparing our Learnable Time Encoding (LTE) module with Positional Encoding (PE), justifying its use. Figures 2 and 3 further support our approach by showing that SciNO achieves more stable Hessian approximation and learns causal relationship. Including these results directly in the Method section was intended to help readers understand why and how the method works as they encounter each design component. On other hand, we agree that the memory efficiency analysis in Figure 4 fits better in the Experiments section. We will move it to Section 4 in the final version to improve the overall organization. We appreciate the reviewer’s helpful suggestion.
>
> **Q5-(b). Table caption formatting**
>
> Thank you for pointing this out. We will move all table captions to the top to adhere to the standard format.
>
> **Q5-(c). Abbreviation formatting consistency**
>
> We appreciate the reviewer’s attention to detail in identifying this inconsistency. To correct this, we will adopt a consistent rule for the first use of any abbreviation, for example: Fourier Neural Operator (FNO) [22] and Positional Embedding (PE) [40].
>
> ---
> [Bühlmann et al., 2014] — CAM: Causal Additive Models, High-Dimensional Order Search and Penalized Regression. The Annals of Statistics, 2014.
>
> [Lachapelle et al., 2020] — Gradient-Based Neural DAG Learning. ICLR, 2020.
>
> [Wang et al., 2021] — Ordering based causal discovery with reinforcement learning. IJCAI, 2021.
>
> [Spirtes et al., 2000] — Causation, Prediction, and Search. MIT press, 2000.
>
> [Chickering, 2002] — Optimal Structure Identification with Greedy Search. JMLR, 2002.

---

> > ### Comment · Reviewer_2jQc · 2025-08-04
> >
> > Thanks for the nice response. All my concerns are addressed, and the new experimental results with CAM validate the empirical performance, which is beyond my expectation. I have increased my score to an accept.
> >
> > Please do consider to add more benchmark methods, e.g., those gradient-based algorithms, in the same table in future versions. For reference, the following package may be used: "gCastle: A Python Toolbox for Causal Discovery" https://github.com/huawei-noah/trustworthyAI/tree/master/gcastle

---

> > > ### Author Response · Authors · 2025-08-05
> > > **Re: Official Comment by Reviewer 2jQc**
> > >
> > > We deeply appreciate your considerate feedback and the revised evaluation!
> > >
> > > > Please do consider to add more benchmark methods, e.g., those gradient-based algorithms, in the same table in future versions. For reference, the following package may be used: "gCastle: A Python Toolbox for Causal Discovery"
> > >
> > > As the reviewer suggested, we will include an additional comparison of benchmark methods and supplement our manuscript to address your concerns.

---

### Official Review · Reviewer_2fQy · 2025-07-02

**Clarity:** 3
**Significance:** 3
**Originality:** 3
**Rating:** 5
**Confidence:** 3

**Summary:**

Recent methods for causal discovery rely on iteratively finding leaf nodes using the second-order derivative of the log-densities. However, current methods show a lack of stability and accuracy in the estimation of such a derivative. This method leverages and improves upon recent advances in diffusion models for more accurate estimation of derivative. These scores can be used to iteratively find leaf nodes, or to improve LLM-driven causal discovery. Experiments show the improved performance induced by the new estimation method.

**Questions:**

Questions are in the Weaknesses above; I am ready to significantly raise my score if they are mostly addressed.

**Ethical Concerns:**

["NO or VERY MINOR ethics concerns only"]

**Final Justification:**

On my side, all issues were resolved. We'll debate the paper's acceptance or not with other reviewers

**Limitations:**

Yes

**Quality:**

2

**Strengths And Weaknesses:**

Strengths: the method is original and is a welcome technical contribution, the paper is generally clear.

Weaknesses: the Theorem is a bit narrow and claims in the experiments are dubious.

Theory (Theorem 3.1) :

1a) The assumptions on the Sobolev space are strong; this means k>27 for D=50 and k>52 for D=100!

1b) Further, it is not applied to a score with a time variable, at least when the time is close to 0.

Main concerns on experiments:

2a) Only 5 independent runs are run in Sections 4.1-4.2, and even more concerning, I understand that the best of 3 runs is Section 4.3.

2b) l.230: "Table 2 shows that SciNO consistently outperforms the original DiffAN across all metrics" : I see ties in a majority of comparisons in Table 2, both for ER and "real-world" (see 3b) graphs.

2c) "SciNO achieves comparable performances across all datasets, with improved results observed on higher-dimensional graphs such as MAGIC-IRRI and ARTH150." The baseline is better with statistical signifiance in a majority of comparisons in Table 3, even most of MAGIC-IRRI.

3) More minor points in experiments

3a) There is no comparison with the original SCORE method while it is extensively discussed before.

3b) BNLearn and Physics are not "real-world" data as I understand it, I would downgrade the "real data" to "semi-synthetic data"

3c) A time complexity comparison between CaPS and the method is done in l.173-174, but Figure 4 does not show runtime.

3d) The analysis of SciNO PE vs LTE is not done on other datasets than ER.

---

> ### Author Rebuttal · Authors · 2025-07-31
>
> We would like to thank the reviewer for very helpful suggestions.
>
> **W1-(a). Strong Sobolev space assumption**
>
> We appreciate detailed questions about our theoretical results.
>
> The condition $k > 2 + \frac{D}{2}$ for Theorem 3.1 (pointwise approximation) appears strong in high dimensions. However, our method's practical success and theoretical grounding rely on a weaker condition. The key is that for our proposed causal discovery algorithms, which compute statistics like the variance eq.(3) or mean eq.(4) of the Hessian diagonal, we only need convergence in the energy norm, not the stronger pointwise sense. We provide a specific guarantee for this in Theorem A.1, which does not require the demanding $k>D/2$ condition. Therefore, Theorem A.1 provides the sufficient theoretical result for our approach. We presented Theorem 3.1 to show that under stronger conditions (which are still comparable to the infinite-differentiability assumption in the baseline SCORE method ), our operator also achieves the pointwise approximation.
>
> **W1-(b). Inapplicability near t=0 for time-conditioned scores**
>
> Thanks for your remarkable point. Our theory is designed to approximate the target score function  $\mathbf{S}\_{\ast} = \nabla\_{\mathbf{x}} \log P\_{\text{data}}$ (L830-836 and Theorem A.1) via neural operators $\widehat{\mathbf{S}}^{\theta}(\cdot)$. The connection to the marginal score model follows a fundamental procedure in diffusion models. Approximation to marginal score functions with a time variable via neural operators in Hilbert space, $\widehat{\mathbf{S}}^{\theta}(t,\cdot) \approx \mathbf{S}(t, \cdot)$ for each $t$, is proved in prior work on HDM [Lim et al., 2023]. Hence the completeness of Hilbert space implies that the distance between $\widehat{\mathbf{S}}^{\theta}(\cdot)$ and the sequence of functional diffusion models will close to zero as we train score models accurately. We will clarify this by adding a paragraph to the appendix.
>
> **W2-(a). Limited number of runs in Section 4.1-4.3**
>
> To address these concerns about statistical robustness, we performed additional experiments, and the new results will be included in our revision.
>
> 1. Experiments in Section 4.1-4.2
>
>     We increased the number of independent runs on real datasets from 5 to 10 to align with our ER setup. As shown in Table 2, SciNO consistently outperforms DiffAN across nearly all datasets and metrics (with a minor exception for SHD on Sachs). In Table 3, SciNO continues to achieve performance comparable to CaPS and exhibits improved order divergence on higher-dimensional graphs such as MAGIC-IRRI and ARTH150.
>
>
>     - **Table 2**
>
>         | Dataset\Method(metric) | DiffAN(OD) | DiffAN(SHD) | DiffAN(SID) | SciNO(OD) | SciNO(SHD) | SciNO(SID) |
>         | --- | --- | --- | --- | --- | --- | --- |
>         | Physics(d7) | 3.3±0.78 | 8.6±2.06 | 16.8±4.73 | **1.9**±0.54 | **5.8**±1.60 | **8.2**±4.62 |
>         | Sachs(d8) | 5.7±2.69 | **22.8**±4.56 | 26.1±10.34 | **5.6**±2.65 | 23.1±3.86 | **23.3**±9.10 |
>         | MAGIC-NIAB(d44) | 8.8±6.85 | 89.1±25.85 | 227.2±157.64 | **4.1**±0.70 | **72.0**±4.52 | **125.5**±28.08 |
>         | ECOLI70(d46) | 21.8±4.26 | 111.0±11.47 | 684.8±80.67 | **12.8**±3.19 | **90.3**±7.13 | **500.9**±58.16 |
>         | MAGIC-IRRI(d64) | 12.0±3.87 | 146.9±13.46 | 321.9±64.64 | **10.6**±1.69 | **144.5**±4.43 | **254.4**±39.66 |
>         | ARTH150(d107) | 43.3±15.77 | 613.3±72.49 | 2456.0±953.08 | **21.4**±2.76 | **515.6**±15.26 | **1186.7**±104.19 |
>
>     - **Table 3**
>
>         | Dataset\Method(metric) | CaPS w/ SciNO(OD) | CaPS w/ SciNO(SHD) | CaPS w/ SciNO(SID) |
>         | --- | --- | --- | --- |
>         | MAGIC-NIAB(d44) | 37.5±0.50 | 165.4±0.5 | 1185.9±5.61 |
>         | ECOLI70(d46) | 25.4±0.80 | 141.3±4.78 | 882.4±13.02 |
>         | MAGIC-IRRI(d64) | **41.6**±0.80 | 183.5±2.97 | 1280.7±36.83 |
>         | ARTH150(d107) | **48.2**±0.87 | **380.0**±4.63 | **2799.1**±73.26 |
>
> 2. Experiments in Section 4.3
>
>     For Llama-3.1 and SciNO, we evaluated performance using fixed-weight models. SciNO was trained using Deep Ensembles [Lakshminarayanan et al., 2017], which inherently account for initialization variance and thus reduce variability across trials. Accordingly, we recorded a single measurement per dataset.For GPT-4o, we initially reported the best result from 3 runs to approximate the model’s potential performance ceiling. To address concerns about generalizability, we have increased the number of runs to 10 and will report both the mean and standard deviation. Our original conclusion that GPT-4o exhibits limitations on high-dimensional causal graphs remains valid.
>
>     | Method\Dataset | Physics(d7) | Sachs(d8) | MAGIC-NIAB(d44) | ECOLI70(d46) | MAGIC-IRRI(d64) | ARTH150(d107) |
>     | --- | --- | --- | --- | --- | --- | --- |
>     | GPT-4o | 1.1±0.54 | 4.4±1.11 | 11.9±0.94 | 41.1±2.30 | 20.9±2.30 | 77.1±7.46 |
>
> **W2-(b). Overgeneralization of performance claims**
>
> To better reflect the results, where SciNO primarily excels in the Order Divergence (OD) metric (on all datasets but Sachs), we will revise the text to be more precise: "Table 2 shows that SciNO enhances the performance of DiffAN in most metrics, especially in terms of OD."
>
> **W2-(c). Overstatement of SciNO’s advantages in MAGIC-IRRI and ARTH150**
>
> We agree that your point regarding Table 3 is valid. Our original intention was to emphasize the performance gains in terms of Order Divergence (OD), which directly measures the accuracy of causal ordering. To avoid confusion, we will revise the sentence to explicitly state “in terms of OD.”
>
> **W3-(a). Missing empirical comparison with SCORE**
>
> We agree that a direct comparison with the SCORE baseline is essential, and we include the results below. As the results show, SciNO performs comparably to SCORE on low-dimensional graphs and demonstrates a significant advantage on high-dimensional datasets. This supports our paper's claim that SciNO enhances scalability and performance in higher-dimensional settings.
>
> | Dataset\Method(metric) | SCORE(OD) | SCORE(SHD) | SCORE(SID) | DiffAN w/ SciNO (OD) | DiffAN w/ SciNO (SHD) | DiffAN w/ SciNO (SID) |
> | --- | --- | --- | --- | --- | --- | --- |
> | ER(d2) | 0.1±0.3 | 0.2±0.60 | 0.2±0.50 | **0.0**±0.0 | **0.0**±0.0 | **0.0**±0.0 |
> | ER(d3) | **0.0**±0.0 | **0.0**±0.00 | **0.0**±0.00 | 0.1±0.3 | 0.2±0.6 | 0.4±1.2 |
> | ER(d5) | **0.0**±0.0 | **0.1**±0.3 | **0.1**±0.3 | 0.9±1.14 | 1.7±1.95 | 4.8±4.94 |
> | ER(d10) | 4.2±2.60 | 21.5±5.24 | **41.2**±13.24 | **2.7**±1.19 | **20.5**±3.5 | 41.6±9.31 |
> | ER(d30) | 24.8±11.97 | 92.8±16.27 | 510.8±90.77 | **16.9**±6.12 | **88.8**±16.5 | **492.0**±91.64 |
> | ER(d50) | 49.3±14.37 | 180.8±15.99 | **1527.1**±179.61 | **32.8**±6.55 | **180.0**±15.45 | 1622.0±110.92 |
> | ER(d100) | 120.0±18.49 | 455.3±26.34 | **7175.0**±435.00 | **86.6**±12.99 | **445.9**±32.84 | 7259.2±653.27 |
> | Physics(d7) | **1** | **2** | **12** | 2.0±0.0 | 5.8±0.40 | **7.8**±1.94 |
> | Sachs(d8) | 6 | 16 | 30 | **5.8**±2.93 | **13.8**±1.72 | **19.4**±5.89 |
> | MAGIC-NIAB(d44) | 8 | **61** | 189 | **3.8**±0.75 | 67.4±2.87 | **156.6**±18.04 |
> | ECOLI70(d46) | **7** | **80** | 771 | 14.4±3.61 | 91.0±7.16 | **734.0**±50.73 |
> | MAGIC-IRRI(d64) | 17 | 175 | 529 | **10.0**±0.89 | **135.4**±5.12 | **333.2**±17.34 |
> | ARTH150(d107) | 38 | 460 | 2709 | **20.8**±2.32 | **454.6**±7.66 | **1740.4**±108.63 |
>
> **W3-(b). Terminology for 'Real-World' vs. 'Semi-Synthetic' Data**
>
> In this paper, we followed the terminology used in prior works (SCORE, DiffAN, and CaPS), which refer to datasets constructed from real-world graphs as real data. However, we agree that classifying them as 'real data' may lead to confusion. We will update the paper to consistently define these datasets as 'semi-synthetic'.
>
> **W3-(c). Missing empirical runtime comparison with CaPS**
>
> While our original Figure 4 focused on memory efficiency, we have conducted the experiments comparing the runtime of CaPS against SciNO on the MAGIC NIAB(d44) dataset, varying the number of samples. Unlike CaPS, SciNO is trained in a multi-step fashion, so its total runtime depends on the number of epochs. Although GPU parallelization can reduce sample-dependent runtime, it does not mitigate the cost associated with iterative training. Therefore, we report the runtime per epoch for clarity. As demonstrated in the 100,000-sample setting, where CaPS runs out of memory while SciNO completes successfully.
>
> | Sample Size\Method | CaPS(sec) | CaPS w/ SciNO(sec/epoch) |
> | --- | --- | --- |
> | 100 | 2.04 | **0.07** |
> | 1000 | 12.08 | **6.94** |
> | 10000 | 77.07 | **62.96** |
> | 100000 | OOM | **156.71** |
>
> We will add a plot of these results to the experimental section of our revised paper. We thank the reviewer again for this valuable suggestion.
>
> **W3-(d). PE vs. LTE ablation on semi-synthetic datasets**
>
> To further validate our design choice, we performed the ablation study comparing the Order Divergence (OD) of Positional Embedding (PE) and our proposed Learnable Time Embedding (LTE) on semi-synthetic datasets. As the results show, the performance advantage of our proposed LTE is most pronounced on high-dimensional datasets, which is consistent with our findings on ER graphs.
>
> | Method(metric)\Dataset | Physics(d7) | Sachs(d8) | MAGIC-NIAB(d44) | ECOLI70(d46) | MAGIC-IRRI(d64) | ARTH150(d107) |
> | --- | --- | --- | --- | --- | --- | --- |
> | DiffAN w/ SciNO (PE) (OD) | 2.7±0.46 | **4.4**±1.5 | **4.0**±1.55 | 30.6±2.42 | 16.6±1.12 | 33.9±3.94 |
> | DiffAN w/ SciNO (LTE) (OD) | **1.9**±0.54 | 5.6±2.65 | 4.1±0.70 | **12.8**±3.19 | **10.6**±1.69 | **21.4**±2.76 |
>
> ---
> [Lakshminarayanan et al., 2017]  — Simple and scalable predictive uncertainty estimation using deep ensembles. NeurIPS, 2017.
>
> [Lim et al., 2023] — Score-based Generative Modeling through Stochastic Evolution Equations in Hilbert Spaces. NeurIPS, 2023.

---

> > ### Comment · Reviewer_2fQy · 2025-08-03
> > **Thanks, parts of Tables?**
> >
> > Many thanks for the exhaustive rebuttal! My concerns on the theory and experiments are addressed, except the following: can you add the rest of Tables 3 and 4? I guess that they should also change if you change the number of runs?

---

> ### Author Response · Authors · 2025-08-04
> **Re: Thanks, parts of Tables?**
>
> - Table 3: note that the CaPS is a kernel-based method which produces identical results when we run multiple experiments. As the reviewer suggested, we provide the complete Table 3 below with the re-evaluated results on SciNO. We will include the result in the revised manuscript.
>
>     | Dataset\Method(metric) | CaPS (OD) | CaPS (SHD) | CaPS (SID) | CaPS w/ SciNO(OD) | CaPS w/ SciNO(SHD) | CaPS w/ SciNO(SID) |
>     | --- | --- | --- | --- | --- | --- | --- |
>     | MAGIC-NIAB(d44) | 36 | 160 | 1024 | 37.5±0.50 | 165.4±0.5 | 1185.9±5.61 |
>     | ECOLI70(d46) | 25 | 134 | 786 | 25.4±0.80 | 141.3±4.78 | 882.4±13.02 |
>     | MAGIC-IRRI(d64) | 42 | 179 | 1192 | **41.6**±0.80 | 183.5±2.97 | 1280.7±36.83 |
>     | ARTH150(d107) | 49 | 406 | 3051 | **48.2**±0.87 | **380.0**±4.63 | **2799.1**±73.26 |
>
> - Table 4: note that the Llama’s pre-trained weights are unchangeable, so the source of randomness is due to the SciNO’s initialization. In the control experiment, we have used $M=30$ (L262) independently trained models for computing eq.(10) and eq.(12) via Deep Ensemble, which greatly reduces the effect of random initialization. Hence the rest of Table 4 results remain unchanged even though we run multiple experiments. We provide the complete Table 4 below with the re-evaluated results on GPT-4o. The confidence intervals of SciNO's results will also be finalized in the revised manuscript when the multiple deep ensemble training (which takes about couple of weeks) is done.
>
>     | Method\Dataset | Physics(d7) | Sachs(d8) | MAGIC-NIAB(d44) | ECOLI70(d46) | MAGIC-IRRI(d64) | ARTH150(d107) |
>     | --- | --- | --- | --- | --- | --- | --- |
>     | GPT-4o | 1.1±0.54 | 4.4±1.11 | 11.9±0.94 | 41.1±2.30 | 20.9±2.30 | 77.1±7.46 |
>     | Llama-3.1-8b(1-norm) | 3 | 3 | 9 | 32 | 18 | 80 |
>     | w/ control(rank) | 1  | 5  | 4  | **11**  | **9**  | 21  |
>     | w/ control(CI) | 1  | 3 | **2**  | 15  | 10  | **18**  |
>     | Llama-3.1-8b(0.5-norm) | 1  | 3 | 14 | 32 | 17 | 80 |
>     | w/ control(rank) | 1 | 4  | **2**  | **10**  | **9**  | 21  |
>     | w/ control(CI) | 1 | 4  | **2**  | 14  | 10 | **18**  |
>
> For clarification, as the reviewer suggested, we promise to include the mean and standard deviation over 10 independent experiments in Tables 3 and 4.

---

> ### Author Response · Authors · 2025-08-05
> **Re: Thanks, parts of Tables?**
>
> To alleviate the reviewer’s concern on multiple experiments, we have conducted supplementary multiple experiments with the reduced number of ensembling (M=3), because running the full experiment with M=30 takes a couple of weeks. We will include the complete results after the discussion period.
>
> We provide the modified experiments below with the re-evaluated results. The results confirms our method's consistency. Under rank-based control, the performance with M = 3 closely matches the results in Table 4. For the CI-based control, we observe slight drops on some datasets, an expected result of using fewer models for CI estimation. Importantly, even with M = 3, our method still yields significant improvements over the uncontrolled Llama and outperforms standalone SciNO (reported in Table 2 of our response to **W2-(a)**) on high-dimensional datasets (ARTH150 being the only exception).
>
> | Method\Dataset | Physics(d7) | Sachs(d8) | MAGIC-NIAB(d44) | ECOLI70(d46) | MAGIC-IRRI(d64) | ARTH150(d107) |
> | --- | --- | --- | --- | --- | --- | --- |
> | GPT-4o | 1.1±0.54 | 4.4±1.11 | 11.9±0.94 | 41.1±2.30 | 20.9±2.30 | 77.1±7.46 |
> | Llama-3.1-8b(1-norm) | 3 | 3 | 9 | 32 | 18 | 80 |
> | w/ control(rank) | 1.5±0.50 | 4.8±2.1 | 3.6±0.66 | **11.5**±1.63 | 9.8±0.75 | **21.5**±1.29 |
> | w/ control(CI) | 1.9±0.54 | 3.7±0.64 | **2.1**±0.94 | 15.4±1.50 | 11.8±0.87 | 23.8±2.99 |
> | Llama-3.1-8b(0.5-norm) | 1 | 3 | 14 | 32 | 17 | 80 |
> | w/ control(rank) | 1.7±0.46 | 4.4±1.86 | 3.6±0.80 | 11.9±1.45 | **9.5**±0.50 | **21.5**±1.29 |
> | w/ control(CI) | 2.8±0.87 | 4.6±0.67 | 4.0±1.00 | 16.2±2.04 | 11.9±1.14 | 23.8±2.99 |
>
> These findings suggest our main results with a full M = 30 ensemble will be even more robust. We will include the complete 10-run results for M = 30 in the final manuscript.

---

> > ### Comment · Reviewer_2fQy · 2025-08-05
> >
> > Many thanks for the new, thorough response, I am now happy to increase my score to an Accept.

---

> > > ### Author Response · Authors · 2025-08-05
> > > **Re: Official Comment by Reviewer 2fQy**
> > >
> > > We deeply appreciate your valuable comments and continuous discussion!

---

### Official Review · Reviewer_Vgze · 2025-07-02

**Clarity:** 3
**Significance:** 2
**Originality:** 2
**Rating:** 4
**Confidence:** 2

**Summary:**

This paper addresses the ordering based causal discovery methods using score matching in the ANM models.

The causal ordering is identified usually by iteratively identifying the leaf nodes, by estimating Jacobian of the score function, and in particular, its diagonal, denoted as the Hessian diagonal function. However, such estimation is not easy: current methods may suffer from numerical instability, computational complexity, high dimensional issues, etc.

Hence, the authors introduce SciNO (Score-informed Neural Operator), a functional diffusion model framework designed to provide stable and scalable approximation of the Hessian diagonal. Score functions are modeled in Sobolev spaces using neural operators.

Furthermore, the authors propose a probabilistic control framework for autoregressive models (e.g., LLMs), integrating SciNO’s score-derived statistics with LLM outputs to guide more accurate causal ordering.

**Questions:**

/

**Ethical Concerns:**

["NO or VERY MINOR ethics concerns only"]

**Final Justification:**

Most of my concerns have been addressed. I keep my positive score.

**Limitations:**

/

**Paper Formatting Concerns:**

/

**Quality:**

3

**Strengths And Weaknesses:**

**Strengths:**

1. The motivation is clear and the problem is relevant. The identified challenge of unstable second-order derivative estimation is indeed important in score matching based causal ordering identification.

2. The techniques introduced -- the use of Sobolev space approximations and neural operators -- are new to the causal discovery field (at least to my knowledge). While I haven’t verified the theoretical details in full, the framework appears sound. It would be valuable if the authors could comment on whether these theoretical insights could benefit other problems in causal discovery.

3. The section on Probabilistic Control of Autoregressive Causal Ordering is interesting and seems to bring a new perspective to causal discovery, both for evaluation and for debiasing in inference results.


**Weaknesses:**

1. While the proposed LLM control framework is interesting and new, it would help to include a deeper discussion on when and why it works. For example, how much do variable semantics and naming conventions influence the predictions? Are the LLM priors generally informative or misleading in high-dimensional settings?

2. For the techniques, could the authors clarify which parts of the proposed architecture are proposed in this paper and which are adapted from existing methods on other problems? If certain components are adopted elsewhere it would be helpful to discuss what problems they were originally designed to address.

3. Since this is an empirical causal discovery method, the experimental comparisons could be more comprehensive. It would be informative to include baselines beyond the ANM category (e.g., constraint-based or score-based methods) to better contextualize.

---

> ### Author Rebuttal · Authors · 2025-07-31
>
> We sincerely appreciate the reviewer’s valuable feedback.
>
> **W1. Deeper discussion on when and why LLM control method works**
>
> In high-dimensional settings, the LLM prior alone can be uninformative and may even assign high probability to incorrect nodes. As shown in Table 4, both GPT-4o and uncontrolled Llama struggled significantly with causal ordering when relying solely on semantic information. This is illustrated at the bottom of Figure 5, where the LLM prior probability for the ground-truth leaf node (green bars) on the ECOLI70 dataset remains very low across most steps, highlighting its lack of confidence in large variable spaces.
>
> Our probabilistic control method addresses this issue by integrating the LLM’s semantic prior with SciNO’s data-informed evidence. As described in L268–270, this integration dramatically boosts the posterior probability of the true leaf node (pink bars at the bottom of Figure 5). Crucially, our method delivers substantial improvements in the early ordering stages when the risk of error accumulation is highest.
>
> To investigate the impact of variable semantics, we conducted an ablation study that masked variable names and varied the proportion of available descriptions (Supplementary Section A.5.2). As noted in L1019–1021, Supplementary Table A.1 demonstrates a robust synergy between the LLM’s semantic information and SciNO’s data-based evidence. Even with only 10% of variable descriptions provided, our controlled model outperforms using SciNO alone in the majority of cases. This result confirms that our method can improve performance even with limited semantic information. Furthermore, by tuning the temperature parameter τ, we can adjust the balance between prior information and evidence to optimize performance across different scenarios.
>
> **W2. Clarification on architectural contributions**
>
> As stated in L133-141, our architecture incorporates two key modifications to the Neural Operator from [Lim et al., 2023], which was designed for image generation tasks. First, we decompose complex-valued Fourier features into real and imaginary parts to better capture functional structures. Second, we integrate a Learnable Time Embedding (LTE) module which is similar to [Li et al., 2021] designed for multi-dimensional spatial Positional Embedding (PE) in Vision Transformer (L145).
>
> Since our model is based on a continuous-time diffusion process, we adopt LTE as a more suitable alternative to standard PE, a choice validated by our results in Table 1 where LTE shows a significant performance advantage (L137–138). Additionally, as shown in our response to **Reviewer hiE3 (Q4-(b))**, our *multiplicative* use of LTE offers substantial gains over a standard additive approach.
>
> To further clarify the effectiveness of our modifications, we conducted a new ablation study applying LTE to DiffAN. These results demonstrate that simply applying LTE to DiffAN fails to produce the same performance gain, especially on high-dimensional datasets (e.g., 82.0 vs. 20.8 on ARTH150).
>
> | Dataset\Method(metric) | DiffAN w/ MLP (LTE) (OD) | DiffAN w/ SciNO (LTE) (OD) |
> | --- | --- | --- |
> | Physics(d7) | 5.0±1.1 | **2.0**±0.0 |
> | Sachs(d8) | 7.2±2.71 | **5.8**±2.93 |
> | MAGIC-NIAB(d44) | 10.6±12.72 | **3.8**±0.75 |
> | ECOLI70(d46) | 20.4±1.85 | **14.4**±3.61 |
> | MAGIC-IRRI(d64) | 25.0±23.06 | **10.0**±0.89 |
> | ARTH150(d107) | 82.0±23.28 | **20.8**±2.32 |
>
> To summarize, rather than naively combining existing components, we integrate architectural modifications, guided by Neural Operator theory (Section 3.1), to construct a functional diffusion model that effectively estimates the Hessian diagonal (Figure 2) and captures causal relations (Figure 3).
>
> **W3. Comparison with non-ANM baselines**
>
> We would like to clarify that our work aims to resolve key issues in ordering-based Causal Discovery methods that utilize Hessian diagonal estimation, rather than proposing a framework that outperforms all existing Causal Discovery methods.
>
> In response to the reviewer’s suggestion, we additionally conducted experiments comparing our method with the constraint-based PC algorithm [Spirtes et al., 2000] and the score-based GES algorithm [Chickering, 2002]. While our primary evaluation metric is Order Divergence, we converted the outputs of all methods to CPDAGs and measured the Structural Hamming Distance(SHD-C) to ensure a fair comparison. These results demonstrate that our method consistently outperforms both PC and GES across ER datasets.
>
> | Dataset\Method(metric) | PC(SHD-C) | GES(SHD-C) | DiffAN w/ SciNO(SHD-C) |
> | --- | --- | --- | --- |
> | ER(d2) | 0.1±0.3 | 0.2±0.4 | **0.0**±0.0 |
> | ER(d3) | 0.6±0.92 | 1.3±1.01 | **0.0**±0.0 |
> | ER(d5) | 9.03±0.64 | 8.4±1.43 | **1.3**±2.41 |
> | ER(d10) | 34.6±2.93 | 35.2±2.96 | **23.8**±5.49 |
> | ER(d30) | 108.5±15.69 | 105.7±15.3 | **88.3**±18.11 |
> | ER(d50) | 195.9±12.08 | 180.2±12.45 | **174.3**±14.48 |
>
> As we show in our response to **Reviewer 2jQc (Q4)**, our method is also highly competitive with CAM [Bühlmann et al., 2014]. This highlights the advantages of our approach against representative methods beyond the ANM family: it achieves superior accuracy while also being more scalable, overcoming the high computational complexity that limits these classic methods. We will add this comparison to our paper to better position our work within the wider causal discovery literature.
>
> ---
> [Lim et al., 2023] — Score-based Generative Modeling through Stochastic Evolution Equations in Hilbert Spaces. NeurIPS, 2023.
>
> [Li et al., 2021] — Learnable Fourier Features for Multi-Dimensional Spatial Positional Encoding. NeurIPS, 2021.
>
> [Spirtes et al., 2000] — Causation, Prediction, and Search. MIT press, 2000.
>
> [Chickering, 2002] — Optimal Structure Identification with Greedy Search. JMLR, 2002.
>
> [Bühlmann et al., 2014] — CAM: Causal Additive Models, High-Dimensional Order Search and Penalized Regression. The Annals of Statistics, 2014.

---

> > ### Comment · Reviewer_Vgze · 2025-08-05
> >
> > I thank the authors for the clear response. Most of my concerns have been addressed. I keep my positive score. Please include the clarifications on the architectural contributions to the updated version. Thanks.

---

> > > ### Author Response · Authors · 2025-08-05
> > > **Re: Official Comment by Reviewer Vgze**
> > >
> > > We greatly thank your valuable feedback!
> > >
> > > > Please include the clarifications on the architectural contributions to the updated version. Thanks.
> > >
> > > As the reviewer suggested, we will clarify those architectural contributions in the revised manuscript.

---

### Official Review · Reviewer_hiE3 · 2025-07-03

**Clarity:** 2
**Significance:** 3
**Originality:** 3
**Rating:** 3
**Confidence:** 1

**Summary:**

This paper studies the approximation of score functions using a newly proposed method, named “score-informed neural operator.” This method was applied to causal discovery score-based methods, and applied to an LLM reasoning task.

**Questions:**

1. The paper starts by introducing causal discovery. However, key related work in ANM is missing. There are ANM methods that start building the causal graph from roots from example. To claim that this method works for causal discovery, it is necessary for the authors to compare against methods beyond score-based approaches. I would actually recommend the authors introducing the method as a generic score approximation method instead,
2. The causal discovery part and the LLM part feel like two standalone topics, and it is strange to put them into the same paper.
3. Section 3.1 is disconnected from Section 3.2. The authors need to explicitly spell out 1) the input to the Fourier transform and the output , and 2) how theorem 3.1 is leveraged to improved the previous methods. In addition, it is unclear what ‘k’ means in equation (7). Is the Fourier transform used to ensure higher-order derivatives exist? How is the model operating in the function space? I found the high-level description at the beginning of Section 3.2 unclear.
4. The baseline architecture in [24] is not described in words in the paper, and the authors did not perform ablation to justify the added modifications. LTE seems like a popular heuristic in modern diffusion model trainings. If separating the Fourier layer into the real and imaginary part is the only novel part, I feel the contribution of the paper is incremental.
5. The authors use the phrase causal representation to mean causal discovery at the end of page 4. I think the authors should correct that as causal representation is often referred to as a different task.
6. I found Section 3.3 unclear: what is node set V? For the Confidence interval, the authors needs to justify why the data is I.i.d. In order to apply the central limit theorem.
7. The setup of the synthetic experiments missing key information in the main body, and the authors should consider adding these key information back. For example, the authors said “we sample node values according to an additive noise model with Gaussian noise.” However, it is important to include the information that the ANM is nonlinear, since linear Gaussian mechanisms are not identifiable.

**Ethical Concerns:**

["NO or VERY MINOR ethics concerns only"]

**Final Justification:**

While the author's rebuttal was useful in clarifying their contributions, they provide a high-level overview of what they are proposing to change in terms of writing rather than concrete changes. I think the amount of rewriting would be significant. However, I also acknowledge that the developed method in this paper improves upon prior work. Thus, I am decreasing my confidence score to 1.

**Limitations:**

see above

**Quality:**

3

**Strengths And Weaknesses:**

The paper introduces many topics, and I often find the details missing, hindering the understanding of the paper. Though, the proposed method seems to work better in approximating the score function than prior methods, the framing of the paper focuses on causal discovery, which unnecessarily complicates the problem setting (with key benchmarks in the causal discovery literature missing). The LLM reasoning is disconnected with the rest of the paper, and the setup description lacks clarity.

---

> ### Author Rebuttal · Authors · 2025-07-31
>
> We sincerely thank the reviewer for the constructive feedback.
>
> **Q1. Concern over whether the method belongs in causal discovery literature**
>
> As the reviewer pointed out, score approximation in function space is a core component of our method. We want to clarify that our primary goal is to enhance ordering-based causal discovery (e.g., DiffAN, CaPS) motivated by [Vashishtha et al., 2025], which studies that predicting causal orderings *rather than full graph structures* can improve consistency and reduce structural errors when combined with LLMs (L27–28). Therefore, SciNO ultimately serves the purpose of estimating the causal order, rather than a generic score approximation. Hence, we focus on ordering-based causal discovery algorithms that require Hessian diagonal approximation under the ANM assumption, as we clarify in Introduction (L30-46) and Supplementary Section A.1.1 (L762-764). We also included non-score matching related works in A.1.1.
>
> To address the concern on the comparison against other methods, we have conducted additional experiments, such as PC [Spirtes et al., 2000], GES [Chickering, 2002], and CAM [Bühlmann et al., 2014]. See the answers to **Reviewer Vgze**’s **W3** and **Reviewer 2jQc**’s **Q4** (We apologize that we are unable to include the table due to the length constraint). Our results demonstrate that SciNO achieves superior performance on ER datasets compared to PC and GES, and also outperforms CAM in high-dimensional settings.
>
> We will revise the related work section to better contextualize our work. We will add a discussion of the ANM methods that build the graph from the roots and contrast them with our approach, clarifying our motivation. We appreciate the reviewer’s valuable comment.
>
> **Q2. Connect between causal discovery and LLM control**
>
> As we mentioned in the answer to **Q1**, [Vashishtha et al., 2025] highlighted the benefits of using LLMs for ordering tasks, providing a key motivation for our research (L27–28).
>
> Recent causal discovery studies have shown that integrating LLM prior knowledge can boost performance. Existing LLM-based methods, however, rely on querying pairwise relationships between variables (L66–69), which suffers quadratic complexity (L73). This prompting strategy is structurally incompatible with ordering-based approaches (e.g., DiffAN, CaPS, SciNO) that sequentially predict leaf nodes. We resolve this problem by treating the LLM’s generation as sequential leaf node prediction given the inferred causal order (L180–181), thereby enabling us to seamlessly combine the LLM priors and SciNO-based control with linear complexity (L73).
>
> Therefore, our contribution is summarized as follows: enhancing ordering-based causal discovery algorithms, and bridging the gap when combining LLM prior knowledges and causal ordering algorithms. We aim to improve ordering-based causal discovery by guiding it with LLM priors, motivated by [Vashishtha et al., 2025], which concludes that SciNO and LLM control are not standalone topics. We will revise the introduction to make this clearer.
>
> **Q3. Clarification of Sections 3.1–3.2**
>
> We deeply appreciate the reviewer’s comment, that help us improve the paper’s clarity.
> Contrary to MLP-based approach (DiffAN), SciNO is founded on neural operator theory. Since MLP-based models are numerically unstable when estimating the higher-order derivatives, DiffAN has limitation in high-dimensional settings despite of its motivation. Hence Section 3.1 explains (L112-115) our theoretical foundation why score matching in function spaces is necessary for approximating the Hessian diagonal eq.(6) and Theorem 3.1 demonstrates approximation power of a family of neural operators including SciNO. Due to the page limitation, we present architectural and operational details in Section A.3. For clarity, we will move the Remark A.4 (L863-867) to the main part, which explains the theoretical connection between two sections and supplement high-level description.
>
> 1. FFT (inverse FFT) module in each Fourier layer receives (outputs) $\mathbb{R}^{H}$-valued tensor.
> 2. Theorem 3.1 presents a universal approximation of neural operator family for estimating score function and its derivatives. As approximating eq.(6) requires higher-order derivatives, $k$ in Theorem 3.1 denotes the order of differentiability when we define the Sobolev space. We will add a sentence clarifying $k$ in eq.(7).
> 3. Baseline structure of SciNO is motivated from FNO, which uses Fourier transform to formulate the operator learning in Sobolev spaces. Thus, Fourier transform is used to ensure high-order derivative approximation. We will add this explanation for clarity.
>
> **Q4-(a). Missing baseline details**
>
> We had previously omitted a detailed description of the baseline architecture from [Lim et al., 2023] since it is detailed in the original paper. Following your feedback, we will add a description of the [Lim et al., 2023] architecture and a comparison with ours in Supplementary Section A.3.
>
> **Q4-(b). Ablation study: multiplicative vs. additive LTE**
>
> We have conducted an experiment based on Order Divergence to validate our choice of a multiplicative approach over the more common additive one. As the results demonstrate, multiplicative LTE combination consistently yields better performance, providing empirical validation for our design.
>
> | Dataset\Method(metric) | SciNO (additive LTE) (OD) | SciNO (multiplicative LTE) (OD) |
> | --- | --- | --- |
> | ER(d10) | 2.8±1.74 | **2.7**±1.19 |
> | ER(d30) | 17.67±5.64 | **16.9**±6.12 |
> | ER(d50) | 38.86±11.42 | **32.8**±6.55 |
> | ER(d100) | 99.2±14.27 | **86.6**±12.99 |
>
> **Q4-(c). Clarification on the novelty of our contribution**
>
> To address the concern that our contribution may be incremental, we want to clarify that our main novelty does not lie in the architecture itself. Our key contribution is the proposal of SciNO as a functional diffusion framework which can accurately estimate the Hessian diagonal (L76-78). This capability is non-trivial, since standard diffusion models inherently fail at this task as we demonstrate in Figure 2. Furthermore, as shown in Figure 3, SciNO can learn the underlying causal relationships in the data.
>
> Our proposed framework is not simply the application of a known heuristic. As discussed in our response to **Reviewer Vgze (W2)**, simply applying the LTE to a standard diffusion model does not guarantee the performance improvements we achieved. Our work shows that the beneficial combination of our Neural Operator architecture and the LTE leads to our effectiveness on this task.
>
> We validate this framework through a theoretical proof (Section 3.1) and empirical results in our experiments, which we believe constitutes a significant contribution.
>
> **Q5. Misuse of the term 'causal representation'**
>
> We agree that our use of 'causal representation' could create confusion with the distinct field of causal representation learning. Our intention was to emphasize that SciNO learns the underlying functional information of the data distribution. To improve clarity, we will replace 'causal representation' with 'causal relationship' throughout the manuscript.
>
> **Q6. Clarification of Section 3.3: Node Set V and i.i.d. Assumption**
>
> 1. Definition of Node Set \\(\mathcal{V}\\)
>
>     The node set \\(\\mathcal{V}\\) is defined in Section 2 (L90). To enhance clarity, we will restate this definition in Section 3.3 of the revised manuscript.
>
> 2. Justification for the i.i.d. Assumption
>
>     The Central Limit Theorem applies to a set of independent and identically distributed (i.i.d.) random variables. In our method, we consider each variance estimate \\( (\sigma_i^{(m)})^2 \\) from the M models as a random variable.
>
>     As stated in L190–191 and L193, each of the M models is trained and evaluated independently, so their predictions are independent. Moreover, since all models share the same architecture and are trained on the same data with identical hyperparameters, their outputs are drawn from the same distribution.
>
>     Therefore, the variance estimates \\( (\sigma_i^{(m)})^2 \\)  satisfy the i.i.d. condition.
>
> **Q7. Missing nonlinear ANM details in Synthetic Experiments**
>
> The reviewer is correct that nonlinearity is crucial for identifiability in Gaussian ANMs. As stated on L226, we generate the data using a Gaussian Process with an RBF kernel, which ensures that the functional relationships are nonlinear. To improve clarity, we will revise the text to state: '...we sample node values according to a nonlinear additive noise model (ANM) with Gaussian noise, where the functions are generated by a Gaussian Process with an RBF kernel.'
>
> ---
> [Spirtes et al., 2000] — Causation, Prediction, and Search. MIT press, 2000.
>
> [Chickering, 2002] — Optimal Structure Identification with Greedy Search. JMLR, 2002.
>
> [Bühlmann et al., 2014] — CAM: Causal Additive Models, High-Dimensional Order Search and Penalized Regression. The Annals of Statistics, 2014.
>
> [Vashishtha et al., 2025] — Causal Order: The Key to Leveraging Imperfect Experts in Causal Inference. ICLR, 2025.
>
> [Lim et al., 2023] — Score-based Generative Modeling through Stochastic Evolution Equations in Hilbert Spaces. NeurIPS, 2023.

---

> ### Author Response · Authors · 2025-08-01
> **Re: Thanks for Clarifying but the Paper Needs to be Rewritten and New Baselines are Needed**
>
> > For example, the authors claim that the main contribution of the paper is to propose a novel causal discovery algorithm.
>
> For clarification, we did **not** claim that SciNO is a novel causal discovery algorithm. I wonder why the reviewer thinks so, since we didn't claim that in the paper or in our rebuttal. One of our paper's contributions is ***enhancing*** the performance of Hessian diagonal-based causal discovery algorithms. I wonder what the reviewer's rationale is for claiming that CaPS, which is a recent algorithm among Hessian diagonal-based causal discovery algorithms, is not suitable as a baseline.
>
> > authors did not provide the details of the outcome of the experiments
>
> We copy the answer to **Reviewers** **2jQc** and **Vgze** here. To better contextualize the performance gap, we evaluated CAM [Bühlmann et al., 2014] under the same experimental setup. Note that our datasets differ from those in [Lachapelle et al., 2020], [Wang et al., 2021] as they are newly generated. SciNO achieves consistently lower SHD on the high-dimensional ER datasets and superior performance on the semi-synthetic datasets for all datasets except ARTH150. Therefore, our method enhances upon existing score-matching based Causal Ordering methods and is also highly competitive with established Causal Discovery benchmarks like CAM.
>
> | Dataset\Method(metric) | CAM(SHD) | CAM(SID) | DiffAN w/ SciNO(SHD) | DiffAN w/ SciNO(SID) |
> | --- | --- | --- | --- | --- |
> | ER(d2) | **0.0**±0.0 | **0.0**±0.0 | **0.0**±0.0 | **0.0**±0.0 |
> | ER(d3) | **0.0**±0.0 | **0.0**±0.0 | 0.2±0.6 | 0.4±1.2 |
> | ER(d5) | **0.0**±0.0 | **0.0**±0.0 | 1.7±1.95 | 4.8±4.94 |
> | ER(d10) | **16.6**±4.45 | **30.9**±11.96 | 20.5±3.5 | 41.6±9.31 |
> | ER(d30) | 129±14.99 | **450.2**±99.2 | **88.8**±16.5 | 492.0±91.64 |
> | ER(d50) | 238.5±17.29 | **1503.6**±201.35 | **180.0**±15.45 | 1622.0±110.92 |
> | ER(d100) | 537.2±45.37 | **6289.9**±533.87 | **445.9**±32.84 | 7259.2±653.27 |
> | Physics(d7) | 12 | 11 | **5.8**±0.40 | **7.8**±1.94 |
> | Sachs(d8) | 37 | 41 | **13.8**±1.72 | **19.4**±5.89 |
> | MAGIC-NIAB(d44) | 167 | 384 | **67.4**±2.87 | **156.6**±18.04 |
> | ECOLI70(d46) | 184 | 828 | **91.0**±7.16 | **734.0**±50.73 |
> | MAGIC-IRRI(d64) | 261 | 791 | **135.4**±5.12 | **333.2**±17.34 |
> | ARTH150(d107) | **437** | 2133 | 454.6±7.66 | **1740.4**±108.63 |
>
> In response to the reviewer’s suggestion, we additionally conducted experiments comparing our method with the constraint-based PC algorithm [Spirtes et al., 2000] and the score-based GES algorithm [Chickering, 2002]. While our primary evaluation metric is Order Divergence, we converted the outputs of all methods to CPDAGs and measured the Structural Hamming Distance(SHD-C) to ensure a fair comparison. These results demonstrate that our method consistently outperforms both PC and GES across ER datasets.
>
> | Dataset\Method(metric) | PC(SHD-C) | GES(SHD-C) | DiffAN w/ SciNO(SHD-C) |
> | --- | --- | --- | --- |
> | ER(d2) | 0.1±0.3 | 0.2±0.4 | **0.0**±0.0 |
> | ER(d3) | 0.6±0.92 | 1.3±1.01 | **0.0**±0.0 |
> | ER(d5) | 9.03±0.64 | 8.4±1.43 | **1.3**±2.41 |
> | ER(d10) | 34.6±2.93 | 35.2±2.96 | **23.8**±5.49 |
> | ER(d30) | 108.5±15.69 | 105.7±15.3 | **88.3**±18.11 |
> | ER(d50) | 195.9±12.08 | 180.2±12.45 | **174.3**±14.48 |
>
> > I find the current write-up in the paper on LLM extremely unclear. I am not sure how the authors are considering the uncertainty coming from the LLM prediction.
>
> If you could point out which parts are "extremely unclear" in the LLM experiments presented in this paper, we would be able to clarify them. The LLM control part proposed in this paper is designed to manage the possibility that LLM predictions could be uncertain or incorrect, and this is explained in Section 3.3. While the reviewer claimed that comparison with existing techniques is necessary, the appropriate comparison target for the LLM control technique proposed in this paper is the LLM prediction results without the control applied.
>
> We would appreciate considering the feedback if the reviewer can provide the detailed reason for claiming that our experimental results distract readers.

---

> > ### Comment · Reviewer_hiE3 · 2025-08-04
> > **update**
> >
> > I thank the authors for the further clarification, and apologize for my misreading of the paper. Here are a few suggestions that might help better clarify the contribution of the paper.
> >
> > The beginning of the abstract and introduction heavily focuses on the description of ordering-based causal discovery methods without quickly diving into the contribution of the paper. This might lead to my initial confusion. I would suggest that the authors change the third sentence of the abstract to directly say that this paper aims to improve the approximation of the Hessian Diagonal of the log-densities, thus improving the performance of existing ordering-based causal discovery algorithms. The authors can then contrast with existing Hessian Diagonal approximation methods. The introduction should follow a similar logic. Under this clarification, my prior comments on the baseline no longer apply.
> >
> > Regarding the LLM part, I still don't understand how having a better Hessian Diagonal approximator alone will lead to better LLM reasoning. Given your current response, I think your contribution on the LLM part relies on proposing a new procedure for integrating the knowledge generated by LLM into the discovery procedure. From the current description of the paper, I feel that the two contributions of the paper are disconnected. If the authors insist on including LLM as a supplement procedure of their proposed method in this paper, then L65 should start with "We supplement our method with a novel procedure that enables better LLM causal reasoning capabilities. Explain what their procedure is doing, how its performance relies on a better HD approximator, and then contrast with existing approaches." Section 3.3 similarly feels disconnected from the rest of the paper due to the lack of transition paragraphs.

---

> > > ### Author Response · Authors · 2025-08-05
> > > **Re: update**
> > >
> > > We appreciate the reviewer for their considerate follow-up and constructive suggestions. We apologize for the initial lack of clarity that led to the misreading.
> > >
> > > ### Regarding the framing of the abstract and introduction
> > >
> > > > I would suggest that the authors change the third sentence of the abstract to directly say that this paper aims to improve the approximation of the Hessian Diagonal of the log-densities, thus improving the performance of existing ordering-based causal discovery algorithms. The authors can then contrast with existing Hessian Diagonal approximation methods. The introduction should follow a similar logic. Under this clarification, my prior comments on the baseline no longer apply.
> > >
> > > We agree that these parts will be clearer if they highlight improvement in the approximation of the Hessian diagonal of the log-densities in the earlier sentence. Following your advice, we will revise these parts to state in front that our primary contribution is a more stable and accurate method for approximating the Hessian Diagonal of log-densities using our proposed Score-informed Neural Operator (SciNO), which are currently at L51-55. We thank you for this comment and for retracting your prior comments on baselines in light of this clarification.
> > >
> > > ### Regarding the connection to the LLM controller
> > >
> > > > From the current description of the paper, I feel that the two contributions of the paper are disconnected. If the authors insist on including LLM as a supplement procedure of their proposed method in this paper, then L65 should start with "We supplement our method with a novel procedure that enables better LLM causal reasoning capabilities. Explain what their procedure is doing, how its performance relies on a better HD approximator, and then contrast with existing approaches." Section 3.3 similarly feels disconnected from the rest of the paper due to the lack of transition paragraphs.
> > >
> > > Thank you for guiding us to make the link between SciNO and the LLM control algorithm more explicit. As the reviewer suggested, we will revise the sentence in the introduction (L65) and add a transition paragraph before Section 3.3 to ensure a smooth logical flow from our core method to Section 3.3.
> > >
> > > Our probabilistic control algorithm (Algorithm 1) works by fusing an **LLM's prior probabilities** with a data-driven **evidence (likelihood)** term. This evidence term is computed directly from statistics of the Hessian diagonal estimates that DiffAN or SciNO provide (eq. 8). To compare SciNO with the existing methods, we also apply the control method to DiffAN (we copy the answers to the Reviewer 2jQc). The table below reports results obtained under the same settings as Table 4, but with the evidence term extracted from DiffAN. On high-dimensional graphs, using SciNO's evidence leads to an average 64% reduction in order divergence (Table 4), versus only 49% with DiffAN. This demonstrates that performance arises from both the novel control method and the stable, accurate evidence provided by SciNO. We will include this result in the revised version.
> > >
> > > | Method\Dataset | Physics(d7) | Sachs(d8) | MAGIC-NIAB(d44) | ECOLI70(d46) | MAGIC-IRRI(d64) | ARTH150(d107) |
> > > | --- | --- | --- | --- | --- | --- | --- |
> > > |GPT-4o| 1 | 2 | 12 | 37 | 22 | 73 |
> > > | **DiffAN (Sanchez et al., 2023)** |  | | | | | |
> > > | Llama-3.1-8b(1-norm) | 3 | 3 | 9 | 32 | 18 | 80 |
> > > | w/ control(rank) | 4  | **3** | **2**  | 22 | **7**  | 25  |
> > > | w/ control(CI) | 3 | 4  | 6  | 16  | 13 | 52 |
> > > | Llama-3.1-8b(0.5-norm) | 1  | 3 | 14 | 32 | 17 | 80 |
> > > | w/ control(rank) | 4 | **3** | **1** | 20 | **8** | 25 |
> > > | w/ control(CI) | 4 | 5 | 8 | 18 | 14 | 52 |
> > > | **SciNO (Ours)** |  | | | | | |
> > > | Llama-3.1-8b(1-norm) | 3 | 3 | 9 | 32 | 18 | 80 |
> > > | w/ control(rank) | **1**  | 5 | 4  | **11** | 9  | **21**  |
> > > | w/ control(CI) | **1** | **3**  | **2**  | **15**  | **10** | **18** |
> > > | Llama-3.1-8b(0.5-norm) | 1  | 3 | 14 | 32 | 17 | 80 |
> > > | w/ control(rank) | **1** | 4 | 2 | **10** | 9 | **21** |
> > > | w/ control(CI) | **1** | 4 | **2** | **14** | **10** | **18** |
> > >
> > > Again, we are very thankful for these actionable suggestions. We are confident that by reflecting them, the two main contributions of the paper will feel cohesive and the overall quality of the manuscript will be substantially improved.

---

### Note · Authors · 2025-08-14

We sincerely thank all reviewers for constructive and active feedbacks, which have helped us clarify the core contributions and improve the completeness of our work. We categorize below the discussion points raised by the reviewers and our corresponding responses and revision plan.

**Clarifying the Contribution**

We clarified that our primary contribution is the proposal of SciNO, a functional diffusion model framework that enhances ordering-based causal discovery algorithms. Accordingly, we will revise the Introduction and Abstract to highlight this. We will also clarify the theoretical motivation for using Neural Operators over MLPs for higher-order derivative estimation and explicitly state that Theorem A.1 offers sufficient conditions for practical use to address the concerns on Theorem 3.1’s assumptions. We will add a detailed comparison of SciNO and FNO architectures and incorporate the ablation results presented during the rebuttal to validate each design choice.

**Strengthening the Connection between SciNO and LLM Control**

To address concerns that SciNO and the LLM control method seem disconnected, we clarified their connection by explaining that our work is motivated by the potential of LLMs for ordering tasks [Vashishtha et al., 2025]. Additional experiments confirm that SciNO-based control outperforms approaches relying on DiffAN or SCORE, highlighting their synergy. We will add a transition paragraph before Section 3.3 and include these comparative results to make the connection explicit.

**Strengthening Experimental Results**

Following the feedback, we provided our baseline comparisons including PC, GES, CAM, and SCORE, and will include gradient-based baselines, confirming our method’s competitiveness. We also increased the number of experiments in Sections 4.1-4.3 for statistical robustness. We also verified that the proposed method’s Hessian diagonal estimates consistently achieve lower error than Stein gradient estimator and yield strong results on low-dimensional data. We will also add a runtime analysis alongside the memory usage comparison (Figure 4).

**Minor Revisions**

We will address all remaining feedback, such as terminology changes and consistency in table captions and abbreviations, to further improve the paper’s clarity and quality.

---

[Vashishtha et al., 2025] — Causal Order: The Key to Leveraging Imperfect Experts in Causal Inference. ICLR, 2025.

---

### Decision · Program_Chairs · 2025-09-17

**Decision:**

Accept (poster)

**Comment:**

The paper contribution is SciNO, a functional diffusion model framework that enhances ordering-based causal discovery algorithms,
building upon [Vashishtha et al., 2025], who show how LLMs can contribute to better ranking performances.

The authors aim to address the limitations of Ordering-based Causal Discovery methods that identify leaf nodes based on the Hessian diagonal (empirically and convincingly showing that Stein gradient-based approaches suffer from similar or worse limitations as Hessian based approaches).

Most concerns of the reviewers were related with insufficient empirical studies. These have been extensively addressed by the authors, considering more baselines, more datasets and more runs, and presenting lesion studies.

Other concerns were voiced, regarding the organization of the paper, as the parts related to SciNO and LLM seemed disconnected at first, but the authors did an excellent job in their rebuttal to explain this organization, and slightly changing the focus/presentation of the paper.

The proposed revision plan comprehensively addresses the concerns.